# DiffSDA: Unsupervised Diffusion Sequential Disentanglement Across Modalities

**Hedi Zisling**\*
Ben-Gurion University

**Ilan Naiman**\*
Ben-Gurion University

**Nimrod Berman**
Ben-Gurion University

{hediz,naimani,bermann}@post.bgu.ac.il

**Supasorn Suwajanakorn**
VISTEC
supasorn.s@vistec.ac.th

**Omri Azencot**
Ben-Gurion University
azencot@bgu.ac.il

## ABSTRACT

Unsupervised representation learning, particularly sequential disentanglement, aims to separate static and dynamic factors of variation in data without relying on labels. This remains a challenging problem, as existing approaches based on variational autoencoders and generative adversarial networks often rely on multiple loss terms, complicating the optimization process. Furthermore, sequential disentanglement methods face challenges when applied to real-world data, and there is currently no established evaluation protocol for assessing their performance in such settings. Recently, diffusion models have emerged as state-of-the-art generative models, but no theoretical formalization exists for their application to sequential disentanglement. In this work, we introduce the Diffusion Sequential Disentanglement Autoencoder (DiffSDA), a novel, modal-agnostic framework effective across diverse real-world data modalities, including time series, video, and audio. DiffSDA leverages a new probabilistic modeling, latent diffusion, and efficient samplers, while incorporating a challenging evaluation protocol for rigorous testing. Our experiments on diverse real-world benchmarks demonstrate that DiffSDA outperforms recent state-of-the-art methods in sequential disentanglement. Code is available at https://github.com/azencot-group/DiffSDA.

## 1 INTRODUCTION

Unconditional generation (Ho et al., 2020; Dhariwal & Nichol, 2021; Rombach et al., 2022), and more broadly, unsupervised learning (Bengio et al., 2012), play a central role in todays machine learning research, as it enables leveraging large-scale data without requiring expensive annotations. Within unsupervised learning, *disentangled representation learning* has become particularly significant (Bengio et al., 2013). This approach seeks to decompose latent representations into distinct factors, where each factor captures a specific variation in the data. Such representations improve interpretability (Liu et al., 2020), mitigate biases (Creager et al., 2019), and improve generalization (Zhang et al., 2022). A prominent challenge is to develop a modal-agnostic approach for *sequential* data such as video, audio, and time series. In particular, the goal is to decompose the sequential signal into separate static and dynamic latent components in an unsupervised manner. For example, in a video of a person speaking, the static factors could represent the person's facial appearance, while the dynamic factors encode facial movements. In audio recordings, static factors may correspond to the speaker's identity, while dynamic factors capture content of the speech.

Despite recent advancements, most sequential disentanglement methods (Tulyakov et al., 2018; Yingzhen & Mandt, 2018; Bai et al., 2021; Han et al., 2021; Naiman et al., 2023; Berman et al., 2024) rely on VAEs and GANs, which often require complex optimization with extensive hyperparameter tuning. For instance, C-DSVAE (Bai et al., 2021) requires *five* hyperparameters solely to balance its various loss terms. Moreover, these models are often evaluated on toy datasets and struggle to produce high-quality samples in real-world scenarios. The reliance on VAEs and GANs is directly related to

---

\*Equal contribution

the absence of a modeling framework for sequential disentanglement within diffusion-based modeling. Further, existing diffusion architectures do not produce disentangled representations (Preechakul et al., 2022; Wang et al., 2023). We hypothesize that a diffusion-based framework can reduce hyperparameter tuning and improve sample quality, paving the way for more robust and scalable approaches to unsupervised sequential disentanglement.

In this work, we introduce *Diffusion Sequential Disentanglement Autoencoder (DiffSDA)*, a novel probabilistic framework for sequential disentanglement. Unlike prior tools (Bai et al., 2021; Naiman et al., 2023), our method models static and dynamic factors as interdependent, enhancing the expressivity of their marginal distributions. Notably, our approach is based on **a single** standard diffusion loss term, while producing high-quality results. Furthermore, DiffSDA is **modal-agnostic**, allowing it to disentangle data across diverse modalities, such as video, audio, and time series, with only minor adjustments to the network. This stands in contrast to modal-dependent methods, such as animation-based approaches for video, which rely on temporal and spatial consistency properties inherent to visual data (Siarohin et al., 2019), or methods designed specifically for audio that depend on spectral or temporal cues (Xu et al., 2024a).

Practically, we implement a **sequential semantic encoder** and adopt the efficient sampling framework EDM (Karras et al., 2022). Moreover, we incorporate a latent diffusion module (LDM) (Rombach et al., 2022) into our architecture, which enables robust handling of high-dimensional, real-world data, outperforming prior sequential disentanglement methods. Finally, using our method, we demonstrate that applying principal component analysis (PCA) to the latent static and dynamic representations reveals a further disentanglement into multiple interpretable factors, showcasing the richness of the learned representations.

We perform a comprehensive evaluation of our model on standard benchmarks for sequential disentanglement (Naiman et al., 2023) across three diverse data domains: audio, time series, and video. To further advance the field, we introduce a novel *evaluation protocol for high-quality visual sequential disentanglement*, incorporating three high-resolution video datasets and multiple quantitative metrics. Additionally, we propose a new post-training approach for disentangling representations into multiple factors. For the first time, our work presents a zero-shot task to demonstrate the generalizability of the factorization framework. Through these extensive evaluations, we show that DiffSDA not only effectively disentangles real-world data but also outperforms recent state-of-the-art methods. Our key contributions are summarized as follows:

1. We propose a novel modal-agnostic probabilistic framework for sequential disentanglement grounded in diffusion processes. Unlike most existing approaches, our formulation accommodates dependent static and dynamic factors of variation. The model is optimized using a single, unified score estimation loss.

2. Our design enables the effective disentanglement of high-dimensional, real-world data and supports zero-shot disentanglement tasks. Moreover, we demonstrate DiffSDA's capability to disentangle static and dynamic information into multiple interpretable factors.

3. We provide a comprehensive evaluation demonstrating our model's superiority in both qualitative and quantitative tasks, outperforming state-of-the-art methods. Additionally, we introduce a novel evaluation protocol specifically designed for video-based disentanglement.

## 2 RELATED WORK

**Generative modeling** is a fundamental methodology for effectively sampling from numerical approximations of data distributions. Prominent approaches include variational autoencoders (VAEs) and generative adversarial networks (GANs) (Kingma, 2013; Goodfellow et al., 2014). More recently, diffusion models (Sohl-Dickstein et al., 2015) and score matching (Hyvärinen & Dayan, 2005; Vincent, 2011) have emerged as powerful alternatives, outperforming VAEs and GANs in generating high-quality samples through iterative denoising of latent variables (Ho et al., 2020; Dhariwal & Nichol, 2021). These methods are unified under a score-based modeling framework (Song et al., 2021). A critical challenge in generative modeling lies in representation learning, where semantic encodings of inputs are derived in an unsupervised manner. A related topic, center to this work, is the study of modal-agnostic disentangled representations, aiming to decompose data of various modalities into distinct factors of variation (Bengio et al., 2013).

**Disentangled Representation Learning.** Most existing works on disentangled learning leverage VAEs and GANs to decompose non-sequential (Higgins et al., 2017; Chen et al., 2018; Kim & Mnih, 2018; Tran et al., 2017; Karras et al., 2020; Ren et al., 2021) and sequential (Hsu et al., 2017; Yingzhen & Mandt, 2018; Zhu et al., 2020; Bai et al., 2021; Han et al., 2021; Naiman et al., 2023; Berman et al., 2024; Simon et al., 2025; Villegas et al., 2017; Tulyakov et al., 2018) data. A key limitation of these approaches lies in their reliance on complex loss formulations, which typically involve multiple regularizers alongside the standard VAEs and GANs losses. While significant progress has been made in enhancing the generative capabilities of VAEs and GANs (Vahdat & Kautz, 2020; Karras et al., 2020), state-of-the-art methods for sequential disentanglement largely focus on simple datasets, far from real-world scenarios, with few exceptions like SPYL's preliminary results (Naiman et al., 2023). In contrast, works in animation (Siarohin et al., 2019; Hu, 2024; Xu et al., 2024b) have shown strong results on real-world data by leveraging video priors for disentangling objects and motion. However, these modal-dependent approaches can exploit relaxed assumptions and specialized tools, whereas our modal-agnostic method can adapt to diverse modalities, including video, audio, and time series.

Table 1: A comparison between animation, diffusion, and sequential disentanglement methods.

| | Method | Modal Agnostic | Efficient | Real-World | Latent Factorization | Latents Prior | Loss Terms |
|---|---|---|---|---|---|---|---|
| ani-mation | FOM Siarohin et al. (2019) | ✗ | ✓ | ✓ | ✗ | N/A | 2 |
| | AA Hu (2024) | ✗ | ✓ | ✓ | ✗ | N/A | 1 |
| | MA Xu et al. (2024b) | ✗ | ✓ | ✓ | ✗ | N/A | 2 |
| non seq. | DiffAE Preechakul et al. (2022) | ✗ | ✗ | ✓ | ✗ | N/A | 1 |
| | InfoDiff Wang et al. (2021) | ✗ | ✗ | ✓ | ✗ | N/A | 2 |
| sequen-tial | SPYL Naiman et al. (2023) | ✓ | ✓ | ✗ | ✓ | independent | 5 |
| | DBSE Berman et al. (2024) | ✓ | ✓ | ✗ | ✓ | independent | 2 |
| | Ours | ✓ | ✓ | ✓ | ✓ | dependent | 1 |

**Diffusion-Based Disentanglement.** The emergence of diffusion models has recently enabled novel approaches for non-sequential disentanglement (Kwon et al., 2022; Yang et al., 2023; Wang et al., 2023; Yang et al., 2024; Zhu et al., 2024; Baumann et al., 2024), achieving high-resolution image generation with disentangled factors. Moreover, other efforts have concentrated on structuring their latent representations. For instance, DiffAE (Preechakul et al., 2022) introduces an autoencoder to facilitate the manipulation of visual features, while InfoDiffusion (Wang et al., 2023) adds a loss regularizer to enhance disentanglement. Despite these advances, to the best of our knowledge, no theoretical formalization, and specifically, probabilistic modeling, has yet been proposed for diffusion-based disentanglement in sequential settings. Furthermore, practical approaches for this domain remain unexplored.

To contextualize our work within the landscape of existing tools, we present a comparative summary in Tab. 1, highlighting how our approach either advances or maintains all key aspects of representation learning. Specifically, while animation methods (FOM, AA, MA) and non-sequential diffusion tools (DiffAE, InfoDiff) handle real-world data, they are modal-dependent and do not provide a latent factorization. Within sequential disentanglement approaches (SPYL, DBSE), only our work supports real-world data via a single loss optimization.

## 3 METHOD

In this section, we introduce a novel probabilistic framework for unsupervised sequential disentanglement based on diffusion models. Currently, none of the existing approaches leverage diffusion models for unsupervised sequential disentanglement, leaving a significant gap in the field. Our framework addresses this gap by establishing a probabilistic modeling formalization and providing an efficient implementation for disentangling static and dynamic factors in sequential data. Background on diffusion models, diffusion autoencoders, and additional details about the method can be found in App. A and App. B. Throughout this section, and the subsequent ones, the subscripts represent time in the diffusion process, and superscripts indicate time in the sequence, e.g., a sequence state of the diffusion process is denoted by $x_t^\tau$, $t \in [0, T]$ and $\tau \in \{1, \ldots, V\}$. $T$ and $V$ represent the maximum diffusion and sequence times, respectively. We consider discrete time sequences of continuous time diffusion processes; however, our modeling can be extended to additional settings.

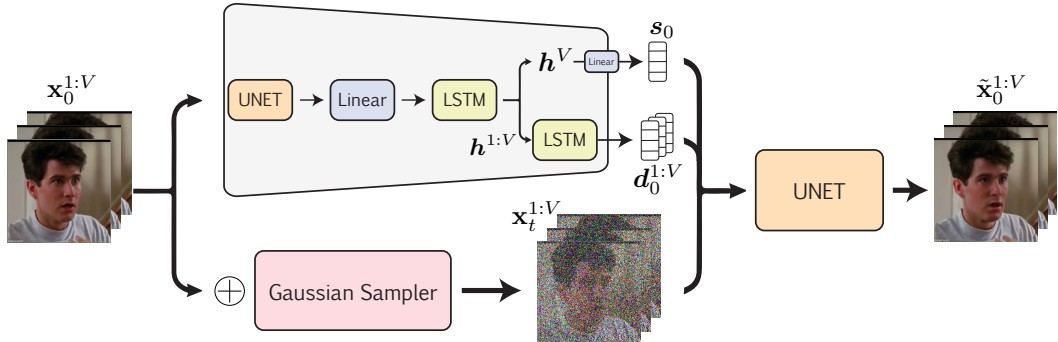

Figure 1: *DiffSDA* processes sequences $\mathbf{x}_0^{1:V}$ via semantic and stochastic encoders (top and bottom). Their outputs $(\boldsymbol{s}_0, \boldsymbol{d}_0^{1:V}, \mathbf{x}_t^{1:V})$ are fed to a stochastic decoder yielding a denoised $\tilde{\mathbf{x}}_0^{1:V}$ (right).

## 3.1 PROBABILISTIC MODELING

Existing frameworks for sequential disentanglement lack a probabilistic modeling foundation for diffusion-based modeling. To address this gap, we propose a novel probabilistic approach based on two diffusion models. The first model details the latent-independent distribution density of the static (time-invariant) and dynamic (time-variant) factors, $\boldsymbol{s}_0$ and $\boldsymbol{d}_0^{1:V}$, respectively. The second model specifies the observed distribution and its dependence on the disentangled factors. Formally, the joint distribution is given by

$$p(\mathbf{x}_0^{1:V}, \mathbf{x}_T^{1:V}, \boldsymbol{s}_0, \boldsymbol{s}_T, \boldsymbol{d}_0^{1:V}, \boldsymbol{d}_T^{1:V}) = p_{T0}(\boldsymbol{s}_0, \boldsymbol{d}_0^{1:V} \mid \boldsymbol{s}_T, \boldsymbol{d}_T^{1:V}) \prod_{\tau=1}^{V} p_{T0}(\mathbf{x}_0^\tau \mid \mathbf{x}_T^\tau, \boldsymbol{s}_0, \boldsymbol{d}_0^\tau) \quad (1)$$

where $p_{T0}(\boldsymbol{s}_0, \boldsymbol{d}_0^{1:V} \mid \boldsymbol{s}_T, \boldsymbol{d}_T^{1:V})$ is a standard diffusion process with $p_{T0}(\cdot)$ being the transition distribution from time $T$ to time 0. The state distribution of $p_{T0}(\mathbf{x}_0^\tau \mid \mathbf{x}_T^\tau, \boldsymbol{s}_0, \boldsymbol{d}_0^\tau)$ is conditioned on the latent $\mathbf{x}_T^\tau$ and the factors $\boldsymbol{s}_0$ and $\boldsymbol{d}_0^\tau$.

Importantly, our probabilistic approach differs from existing work (Bai et al., 2021; Naiman et al., 2023) in that our static and dynamic factors are interdependent. We motivate our model by three main reasons: i) expressiveness—the overall dependence facilitates learning of different state trajectories, leading to higher expressivity in the marginals $p_{t0}(\cdot)$; and ii) efficiency—our sampler is not autoregressive, allowing for fast and parallelized sampling; and iii) causality—our model has the ability to learn intricate relationships between the static and dynamic factors, if needed. We evaluate both the dependent and independent approaches on our model to highlight the effectiveness of our approach. In summary, adopting dependent modeling improves generation quality by 13%. Further details can be found in App. G.1.

Given a sequence $\mathbf{x}_0^{1:V} \sim p_0(\mathbf{x}_0^{1:V})$, the posterior distribution of the latent variables $\mathbf{x}_t^{1:V}$ and latent factors $\boldsymbol{s}_0$ and $\boldsymbol{d}_0^{1:V}$ is composed of three independent distributions. Further, unlike the non-autoregressive prior in Eq. 1, here, we explicitly assume temporal dependence. The posterior distribution reads

$$p(\mathbf{x}_t^{1:V}, \boldsymbol{s}_0, \boldsymbol{d}_0^{1:V} \mid \mathbf{x}_0^{1:V}) = p_{0t}(\mathbf{x}_t^{1:V} \mid \mathbf{x}_0^{1:V}) p(\boldsymbol{s}_0 \mid \mathbf{x}_0^{1:V}) \prod_{\tau=1}^{V} p(\boldsymbol{d}_0^\tau \mid \boldsymbol{d}_0^{<\tau}, \mathbf{x}_0^{\leq\tau}) \quad (2)$$

where $\mathbf{x}_t^{1:V}$ and $\boldsymbol{s}_0$ are conditioned on the entire input $\mathbf{x}_0^{1:V}$, and the dynamic factors only depend on previous dynamic factors and current and previous data elements. We employ score matching (Hyvärinen & Dayan, 2005; Song et al., 2021), to optimize for the denoising parametric map $\boldsymbol{D}_\theta$. The map $\boldsymbol{D}_\theta$ takes the noisy latent $\mathbf{x}_t^\tau$, time $t$, and disentangled factors $\mathbf{z}_0^\tau := (\boldsymbol{s}_0, \boldsymbol{d}_0^\tau)$, and it returns an estimate of the score function $\nabla_{\mathbf{x}} \log p_{0t}(\mathbf{x}_t^\tau \mid \mathbf{x}_0^\tau)$. Overall, the optimization objective reads

$$\boldsymbol{\theta}^* = \arg\min_{\boldsymbol{\theta}} \mathbb{E}_t \left\{ \lambda_t \mathbb{E} \left[ \|\boldsymbol{D}_\theta - \nabla_{\mathbf{x}} \log p_{0t}\|_2^2 \right] \right\}, \quad (3)$$

where $\lambda_t \in \mathbb{R}^+$ is a positive weight, $t \sim \mathcal{U}[0, T]$ is uniformly sampled over $[0, T]$, the variables $\mathbf{x}_t^\tau$, $\mathbf{x}_0^\tau$ are sampled from their respective distributions, $p_{0t}(\cdot), p_0(\cdot)$, and $\mathbf{z}_0^\tau$ via the densities in Eq. 2.

The inner expectation is taken over $\mathbf{x}_t^\tau$, $\mathbf{z}_0^\tau$, and $\mathbf{x}_0^\tau$. Importantly, $p_{T0}$ of $\boldsymbol{s}_0, \boldsymbol{d}_0^{1:V}$ is not used in Eq. 3, and thus its optimization can be separated.

Notably, we make no assumptions about the given data $\mathbf{x}_0^{1:V}$, ensuring that our framework remains modal-free and independent of specific properties of video, audio, or time series data. This theoretical compatibility with any type of sequence makes it highly adaptable to diverse applications.

## 3.2 DIFFUSION SEQUENTIAL DISENTANGLEMENT AUTOENCODER

Our architecture, shown in Fig. 1, comprises three main components: (1) a sequential semantic encoder, (2) a stochastic encoder, and (3) a stochastic decoder. At a high level, the sequential semantic encoder factorizes data into separate static and dynamic components, while the stochastic decoder denoises the noisy latent representation produced by the stochastic encoder, conditioned on the disentangled factors. Notably, unlike prior works, our implementation achieves disentanglement with a single, simple loss term.

**Encoders.** Inspired by prior work in sequential disentanglement (Yingzhen & Mandt, 2018), we design a novel *sequential semantic encoder* to extract $\boldsymbol{s}_0$ and $\boldsymbol{d}_0^{1:V}$. Particularly, it consists of a U-Net (Ronneberger et al., 2015) for video data and an MLP for other modalities, coupled with linear layers that independently process each sequence element. Then, an LSTM module summarizes the sequence into a latent representation $\boldsymbol{h}^{1:V}$. The last hidden, $\boldsymbol{h}^V$, is passed to a linear layer to produce $\boldsymbol{s}_0$, whereas $\boldsymbol{h}^{1:V}$ are processed with another LSTM and a linear layer to produce $\boldsymbol{d}_0^{1:V}$. Our stochastic encoder follows the EDM framework (Karras et al., 2022), adding noise $\epsilon \sim \mathcal{N}(0, \sigma_t^2 I)$ to each element $\mathbf{x}_0^\tau$, yielding $\mathbf{x}_t^\tau = \mathbf{x}_0^\tau + \epsilon$. These encoders realize in practice the posterior in Eq. 2.

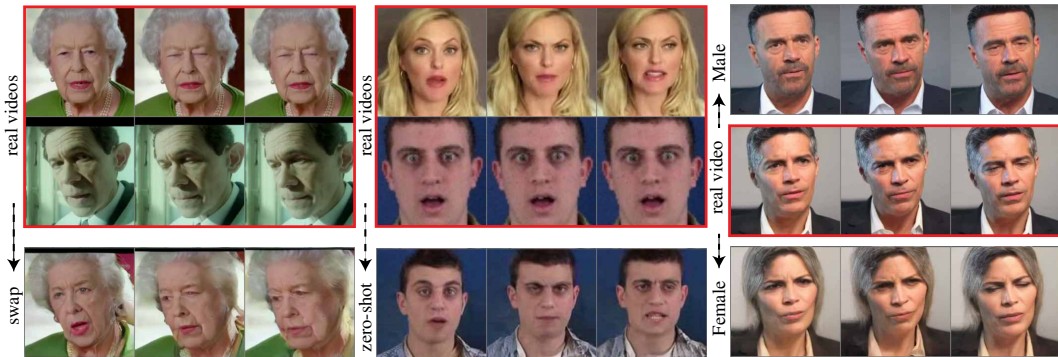

Figure 2: We present swap (left), zero-shot (middle), and multifactor disentanglement (right) results on multiple real-world and high-resolution visual datasets. See Sec. 4 for further details.

**Decoder.** To efficiently handle real-world sequential information, we follow the decoding in EDM (Karras et al., 2022), featuring only 63 neural function evaluations (NFEs) during inference. Our decoder $\boldsymbol{D}_\theta$ takes as inputs the noisy input $\mathbf{x}_t^\tau$ and disentangled factors $\mathbf{z}_0^\tau := (\boldsymbol{s}_0, \boldsymbol{d}_0^\tau)$, and it returns a denoised version of $\mathbf{x}_t^\tau$, denoted by $\tilde{\mathbf{x}}_0^\tau$. Given any $t \in [0, T]$ and $\tau \in \{1, \ldots, V\}$, the decoder is parameterized independently from other times $t', \tau'$ as follows

$$\tilde{\mathbf{x}}_0^\tau := \boldsymbol{D}_\theta(\mathbf{x}_t^\tau, t, \mathbf{z}_0^\tau) = c_t^{\text{skip}} \mathbf{x}_t^\tau + c_t^{\text{out}} \boldsymbol{F}_\theta \left( c_t^{\text{in}} \mathbf{x}_t^\tau, \mathbf{z}_0^\tau, c_t^{\text{noise}} \right) , \qquad (4)$$

where $c_t^{\text{skip}}$ modulates the skip connection, $c_t^{\text{in}}, c_t^{\text{out}}$ scale the input/output magnitudes, and $c_t^{\text{noise}}$ maps noise at time $t$ into a conditioning input for the neural network $\boldsymbol{F}_\theta$, conditioned on $\mathbf{z}_0^\tau$ through AdaGN.

**Loss.** While prior sequential disentanglement works depend on intricate prior modeling, regularization terms, and mutual information losses, leading to many hyper-parameters and challenging training, we opt for a simpler objective containing a single loss term that is based on Eq. 3,

$$\mathbb{E}_{t,\mathbf{x}_t^\tau,\mathbf{z}_0^\tau,\mathbf{x}_0^\tau} \left[ \lambda_t (c_t^{\text{out}})^2 \|\boldsymbol{F}_\theta - \frac{1}{c_t^{\text{out}}} (\mathbf{x}_0^\tau - c_t^{\text{skip}} \cdot \mathbf{x}_t^\tau) \|_2^2 \right] , \qquad (5)$$

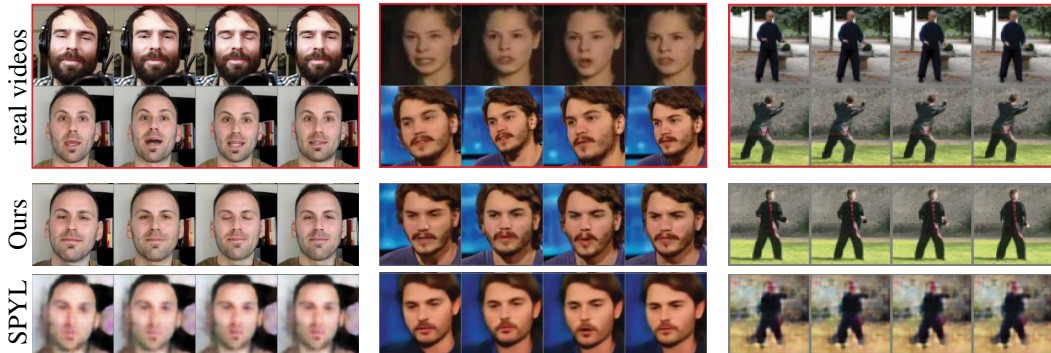

Figure 3: We present dynamic swap results of our approach (third row) and SPYL (fourth row) on CelebV-HQ (left), VoxCeleb (middle), and TaiChi-HD (right).

where $\boldsymbol{F}_\theta$ takes as inputs $c_t^{\text{in}}\mathbf{x}_t^\tau$, $\mathbf{z}_0^\tau$, and $c_t^{\text{noise}}$. While our loss in Eq. 5 does not include auxiliary terms, it promotes disentanglement due to two main reasons: i) the static factor $\boldsymbol{s}_0$ is shared across $\tau$, and thus it will not hold dynamic information, and ii) the dynamic factors $\boldsymbol{d}_0^\tau \in \mathbb{R}^k$ are low-dimensional (i.e., $k$ is small), making it difficult for $\boldsymbol{d}_0^\tau$ to store static features. We empirically validate these assumptions through experiments presented in App. G.2. Finally, we briefly mention that to support high-resolution sequences, we incorporate latent diffusion models (LDM) (Rombach et al., 2022), using a pre-trained VQ-VAE autoencoder to reduce the high-dimensionality of input frames. Instead of factorizing all the equations above with new symbols for the features VQ-VAE produces, we denote by $\mathrm{x}_0^{1:V}$ the input sequence, and we abuse the notation $\mathbf{x}_0^{1:V}$ to denote the latent features, i.e., $\mathbf{x}_0^{1:V} = \mathcal{E}(\mathrm{x}_0^{1:V})$ and $\mathrm{x}_0^{1:V} = \mathcal{D}(\mathbf{x}_0^{1:V})$, where $\mathcal{E}$ and $\mathcal{D}$ are the VQ-VAE encoder and decoder, respectively.

## 4 RESULTS

Below, we empirically evaluate the modeling capabilities of DiffSDA in comparison to recent *modal-agnostic* state-of-the-art methods (see Tab. 1), SPYL (Naiman et al., 2023) and DBSE (Berman et al., 2024). In general, we consider quantitative and qualitative experiments. For video, we include three high-resolution, real-world visual datasets that have not been previously used for sequential disentanglement: VoxCeleb (Nagrani et al., 2017), CelebV-HQ (Zhu et al., 2022), and TaiChi-HD (Siarohin et al., 2019), along with the popular MUG dataset (Aifanti et al., 2010). For audio, we consider TIMIT (Garofolo, 1993) and a new dataset, Libri Speech (Panayotov et al., 2015). The time series datasets are PhysioNet, ETTh1, and Air Quality (Tonekaboni et al., 2022). Detailed descriptions of the datasets and their pre-processing can be found in App. D, while extended baseline comparisons are provided in App. H.1. For brevity, we omit below the subscript indicating the diffusion step for clean samples (corresponding to time step 0).

### 4.1 CONDITIONAL SWAP IN VIDEOS

We begin our tests with the conditional swap task (Yingzhen & Mandt, 2018). Given two sample videos $\mathbf{x}, \hat{\mathbf{x}} \sim p_0$, the goal in this experiment is to create a new sample $\bar{\mathbf{x}}$, conditioned on the static factor of $\mathbf{x}$ and dynamic features of $\hat{\mathbf{x}}$. This is done by extracting the latent factors $\mathbf{z} = (\boldsymbol{s}, \boldsymbol{d}^{1:V})$ and $\hat{\mathbf{z}} = (\hat{\boldsymbol{s}}, \hat{\boldsymbol{d}}^{1:V})$ for $\mathbf{x}$ and $\hat{\mathbf{x}}$, respectively. The new sample $\bar{\mathbf{x}}$ is defined to be the reconstruction of $\bar{\mathbf{z}} = (\boldsymbol{s}, \hat{\boldsymbol{d}}^{1:V})$ through sampling, see Alg. 1. In an ideal swap, $\bar{\mathbf{x}}$ preserves the static characteristics of $\mathbf{x}$ while presenting the dynamics of $\hat{\mathbf{x}}$, thus demonstrating strong disentanglement capabilities of the swapping method. We show in Fig. 2 (left) a swap example of DiffSDA, where the top two rows are real videos, and the third row shows the new sample obtained by preserving the static features of the first row and using the dynamics of the second row. Remarkably, while the people in these sequences are very different, many fine details are transferred, including head angle and orientation, as well as mouth and eyes orientation and openness. In Fig. 3, we present additional swap results on CelebV-HQ (left), VoxCeleb (middle), and TaiChi-HD (right), comparing DiffSDA (third row) to SPYL (fourth row). Our approach produces high-quality samples, while swapping the dynamics of

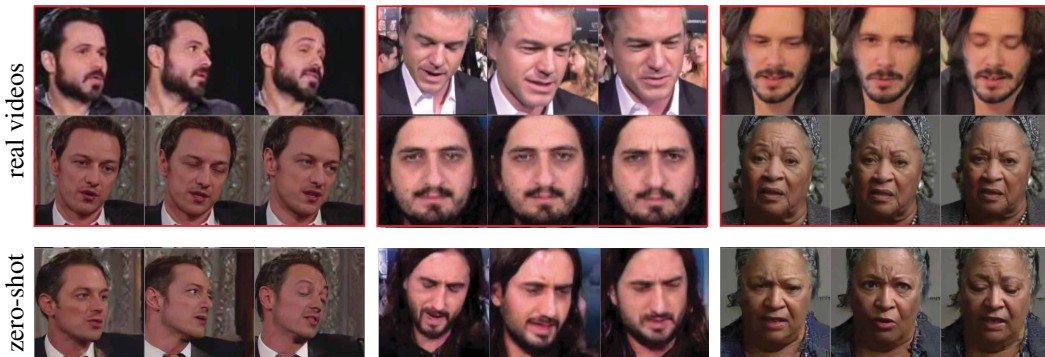

Figure 4: Zero-shot swap results, training on VoxCeleb and tested on CelebV-HQ or MUG.

the second row into the first row, whereas SPYL struggles both with the reconstruction and swap. Additional conditional and unconditional swap results appear in App. H.3 and App. H.4, respectively.

In addition to the above qualitative evaluation, we also want to quantitatively assess DiffSDA's effectiveness. We report in App. F results from the traditional quantitative benchmark, where a pre-trained judge (classifier) is used to determine if swapped content is correct (Bai et al., 2021). However, there are two main issues with the benchmark: i) it depends on labeled data, making it relevant to only a small number of datasets; and ii) results are sensitive to the expressivity and generalizability of the judge. For instance, swapping a smiling expression from person A to person B, may result in person B having a smile, different from the one in the data. In these cases, the judge may wrongly classify a different expression to the smiling person B, see App. F for further discussion.

Towards addressing these issues, we propose new *unsupervised* swapping metrics to quantitatively measure the model's disentanglement abilities. We adopt estimators commonly used in animation for assessing if objects and motions are preserved (Siarohin et al., 2019). Specifically, we utilize the *average Euclidean distance* (AED) that is based on the distances between the latent representations of images. Further, we also employ the *average keypoint distance* (AKD) which computes the distances between selected keypoints in images. Intuitively, AED and AKD have been designed to identify the preservation of objects and motions in images, respectively. See App. E for definitions.

Equipped with these new metrics, we perform conditional swapping over a pre-defined random list of sample pairs, $\mathbf{x}, \hat{\mathbf{x}}$. Particularly, we reconstruct new samples of the form $\mathbf{z}^s := (\boldsymbol{s}, \hat{\boldsymbol{d}}^{1:V})$ and $\mathbf{z}^d := (\hat{\boldsymbol{s}}, \boldsymbol{d}^{1:V})$, encoding dynamic and static swaps, respectively. We compute the AED of $\mathbf{z}^s$ with respect to $\mathbf{z}$ (arising from $\mathbf{x}$), expecting their static features to be similar. Following the same logic, we compute the AKD of $\mathbf{x}^d$ (reconstructed from $\mathbf{z}^d$) and $\mathbf{x}$, as they share the dynamic factors. Our findings are presented in Tab. 2, where DiffSDA outperforms SOTA previous (SPYL, DBSE) approaches across all datasets, except for AED on TaiChi-HD, where we attain the second best error. Notably, our AKD errors are significantly lower than SPYL and DBSE. Further, we apply these metrics to assess reconstruction performance, as well as the mean squared error (MSE), with the results shown in Tab. 3. Again, DiffSDA is superior to current SOTA methods. Additionally, we include a generative evaluation in App. G.4, comparing our approach to previous methods.

Table 2: Preservation of objects (AED) and motions (AKD) is estimated across several datasets and methods. The labels 'static frozen' and 'dynamics frozen' correspond to samples $\mathbf{z}^s$ and $\mathbf{z}^d$.

|  | AED↓ (static frozen) | | | AKD↓ (dynamics frozen) | | |
|---|---|---|---|---|---|---|
|  | SPYL | DBSE | Ours | SPYL | DBSE | Ours |
| MUG ($64 \times 64$) | 0.766 | 0.773 | **0.751** | 1.132 | 1.118 | **0.802** |
| VoxCeleb ($256 \times 256$) | 1.058 | 1.026 | **0.846** | 4.705 | 10.96 | **2.793** |
| CelebV-HQ ($256 \times 256$) | 0.631 | 0.751 | **0.540** | 39.16 | 28.69 | **6.932** |
| TaiChi-HD ($64 \times 64$) | 0.443 | **0.325** | 0.326 | 7.681 | 6.312 | **2.143** |

## 4.2 ZERO-SHOT VIDEO DISENTANGLEMENT

In the previous sub-section, the conditional swap was performed on the held-out test set of each dataset on which we trained on. In contrast to previous work, for the first time, we perform the same task on a dataset unseen during training. We show an example in Fig. 2 (middle) of zero-shot swap, where our model was trained on the VoxCeleb dataset (1st row) and the inferred sequence was taken from MUG (2nd row). Particularly, we froze the static features of the MUG sample and swapped the dynamic factors with those of VoxCeleb (3rd row). Remarkably, in addition to changing the facial expression of the person, DiffSDA also adds the necessary details to mimic the body pose. We emphasize that the MUG dataset does not include sequences similar to the third row in Fig. 2, but rather zoomed-in facial videos as shown in the second row, thus, our zero-shot results present a significant adaptation to the new data. Additionally, we include in Fig. 4 zero-shot examples where DiffSDA is trained on VoxCeleb and evaluated on CelebV-HQ or MUG. These results further highlight the effectivity of our approach in transferring dynamic features across different datasets. Finally, we provide more zero-shot examples in App. H.5.

## 4.3 TOWARD MULTIFACTOR VIDEO DISENTANGLEMENT

Multifactor sequential disentanglement is a challenging problem, where the objective is to produce several static factors and several dynamic factors per frame (Berman et al., 2023; Barami et al., 2025). Here, we show that our model has the potential to further disentangle the static and dynamic features into additional factors of variation. Inspired by DiffAE (Preechakul et al., 2022), we explore the learned latent space in an unsupervised linear fashion, particularly, using principal component analysis (PCA). Namely, to obtain fine-grained semantic static factors of variation, we sample a large batch of static vectors $\hat{s}_j \in \mathbb{R}^h$, with $h$ the static latent size, $j = 1, \ldots, b = 2^{15}$. Then, we compute PCA on the matrix formed by arranging $\{\hat{s}_j\}$ in its columns, yielding the principal components $\{v_i\}_{i=1}^h$, given that $b \geq h$. We can utilize the latter pool of static variability by exploring the latent space from a static code $s$ of a real example $\mathbf{x}$ in the test set, i.e.,

$$\bar{s} = \left( \frac{s - \mu_{\hat{s}}}{\sigma_{\hat{s}}} + \alpha v_i \cdot \sqrt{h} \right) \cdot \sigma_{\hat{s}} + \mu_{\hat{s}} \,, \tag{6}$$

where $\mu_{\hat{s}}$ and $\sigma_{\hat{s}}^2$ are the mean and variance of the sampled static features, $\{\hat{s}_j\}_{j=1}^b$, and $\alpha \in [-\kappa, \kappa]$, notice that $\alpha = 0$ recovers the original sequence. The new sample $\bar{\mathbf{x}}$ is obtained by reconstructing the new static features $\bar{s}$ with the original dynamic factors $d^{1:V}$ of $\mathbf{x}$.

We demonstrate a static PCA exploration in Fig. 2 (right) on VoxCeleb. The middle row is the real video, whereas the top and bottom rows use positive and negative $\alpha$ values, respectively. Our results show that traversing in the positive direction yields more masculine appearances, and in contrast, going in the negative direction produces more feminine characters. Importantly, we highlight that other static features and the dynamics are fully preserved across the sequence. In App. H.6, we present further results on full sequences using multiple $\alpha$ values to demonstrate the gradual transition in the latent space. Notably, we find in our exploration principal components that control other features such as skin tone, image blurriness, and more.

Table 3: Reconstruction errors are measured in terms of AED, AKD, and MSE across several datasets and models. We find DiffSDA to be orders-of-magnitude better than other methods.

|  | AED↓ | | | AKD↓ | | | MSE↓ | | |
| --- | --- | --- | --- | --- | --- | --- | --- | --- | --- |
|  | SPYL | DBSE | Ours | SPYL | DBSE | Ours | SPYL | DBSE | Ours |
| MUG | 0.49 | 0.49 | **0.11** | 0.47 | 0.48 | **0.06** | 0.001 | 0.001 | **3e−7** |
| VoxCeleb | 0.99 | 1.03 | **0.37** | 2.27 | 2.43 | **1.09** | 0.005 | 0.003 | **5e−4** |
| CelebV-HQ | 0.70 | 0.78 | **0.29** | 15.0 | 13.8 | **1.26** | 0.012 | 0.006 | **6e−4** |
| TaiChi-HD | 0.32 | 0.29 | **0.001** | 4.31 | 3.83 | **0.10** | 0.018 | 0.007 | **2e−7** |

## 4.4 SPEAKER IDENTIFICATION IN AUDIO

Our approach is inherently modal-agnostic and extends beyond the video domain. Unlike methods tailored specifically for video or audio, which often require extensive modifications when applied to new modalities, our method is versatile and can adapt to different modalities with minimal adjustments to the backbone architecture. For example, to process audio data, we simply replace the U-Net with an MLP. In Tab. 4, we demonstrate the adaptability of our model by successfully disentangling audio data from the TIMIT dataset and Libri Speech, where TIMIT is a widely used benchmark for speech-related tasks and Libri Speech is an additional dataset we add for this benchmark. Following the speaker identification benchmarks (Yingzhen & Mandt, 2018), we evaluate disentanglement quality using the Equal Error Rate (EER), a standard metric in speech tasks. Specifically, the Static EER measures how effectively the static latent representations capture speaker identity, and similarly, the Dynamic EER assesses the dynamic latent representations. Notably, a well-disentangled model should yield a low Static EER (capturing speaker identity in static representations) and a high Dynamic EER (capturing content-related dynamics without speaker identity). The overall goal is to maximize the gap between these two metrics (Dis. Gap). Our model, achieves in TIMIT a disentanglement gap improvement of over 11%, with a 42.29% compared to 31.11% achieved by DBSE, thereby surpassing current state-of-the-art methods. Similar strong performance is achieved on Libri Speech as well. These results highlight the efficacy of our approach in the audio domain. Additional details regarding the dataset, evaluation metrics, and implementation are provided in the appendix. Furthermore, we report speech quality and reconstruction results in App. G.3, further validating our model's effectiveness in the audio domain.

Table 4: Disentanglement metrics on TIMIT and LibriSpeech

| Method | TIMIT | | | LibriSpeech | | |
|---|---|---|---|---|---|---|
| | Static EER↓ | Dynamic EER↑ | Dis. Gap↑ | Static EER↓ | Dynamic EER↑ | Dis. Gap↑ |
| DSVAE | 5.64% | 19.20% | 13.56% | 15.06% | 28.94% | 13.87% |
| SPYL | **3.41**% | 33.22% | 29.81% | 24.87% | **49.76**% | 24.89% |
| DBSE | 3.50% | 34.62% | 31.11% | 16.75% | 22.61% | 5.58% |
| Ours | 4.43% | **46.72**% | **42.29**% | **11.02**% | 45.94% | **34.93**% |

## 4.5 DOWNSTREAM PREDICTION AND CLASSIFICATION TASKS ON TIME SERIES INFORMATION

Generative modeling of time series data has advanced rapidly in recent years (Naiman et al., 2024; Fadlon et al., 2025; Gonen et al., 2025). In the context of disentanglement, we evaluate our approach on time series data, following the evaluation protocol in (Berman et al., 2024). The evaluation is carried out in two main independent setups: 1) We assess the quality of the learned latent representations using a predictive task. The model is trained on a dataset, and at test time, the static and dynamic factors are extracted and used as input features for a predictive model. Two tasks are considered: (i) predicting mortality risk with the PhysioNet dataset (Goldberger et al., 2000), and (ii) predicting oil temperature using the ETTh1 dataset (Zhang et al., 2017). Performance is evaluated using AUPRC and AUROC for PhysioNet, and Mean Absolute Error (MAE) for ETTh1. 2) We investigate the model's ability to capture global patterns within its disentangled static latent representations, which have been shown to enhance performance (Trivedi et al., 2015). Following a similar procedure, the model is trained, and now only the static representations are extracted. These representations are then used as input features for a classifier. For the PhysioNet dataset, Intensive Care Unit (ICU) unit types are used as global labels, while for the Air Quality dataset, the month of the year serves as the target variable. Further details regarding datasets, metrics, and implementation can be found in App. D and App. E. We compare our method vs. state-of-the-art baselines, including DBSE, SPYL, and GLR (Tonekaboni et al., 2022). Results for predictive and classification tasks are given in Tab. 5. Notably, our model outperforms across all tasks.

Table 5: Time series prediction and classification benchmarks.

| | Task | GLR | SPYL | DBSE | Supervised | Ours |
|---|---|---|---|---|---|---|
| pred. | AUPRC↑ | $0.37 \pm 0.09$ | $0.37 \pm 0.02$ | $0.47 \pm 0.02$ | $0.44 \pm 0.02$ | $\mathbf{0.50 \pm 0.006}$ |
| | AUROC↑ | $0.75 \pm 0.01$ | $0.76 \pm 0.04$ | $0.86 \pm 0.01$ | $0.80 \pm 0.04$ | $\mathbf{0.87 \pm 0.004}$ |
| | MAE↓ (ETTh1) | $12.3 \pm 0.03$ | $12.2 \pm 0.03$ | $11.2 \pm 0.01$ | $10.19 \pm 0.20$ | $\mathbf{9.89 \pm 0.280}$ |
| cls. | PhysioNet↑ | $38.9 \pm 2.48$ | $47.0 \pm 3.04$ | $56.9 \pm 0.34$ | $62.00 \pm 2.10$ | $\mathbf{64.6 \pm 0.35}$ |
| | Air Quality↑ | $50.3 \pm 3.87$ | $57.9 \pm 3.53$ | $65.9 \pm 0.01$ | $62.43 \pm 0.54$ | $\mathbf{69.2 \pm 1.50}$ |

## 5 CONCLUSIONS

The analysis and results of this study underscore the potential of the proposed DiffSDA model to address key limitations in sequential disentanglement, specifically in the context of complex real-world visual data, speech audio, and time series. By leveraging a novel probabilistic framework, diffusion autoencoders, efficient samplers, and latent diffusion models, DiffSDA provides a robust solution for disentangling both static and dynamic factors in sequences, outperforming existing state-of-the-art methods. Moreover, the introduction of a new real-world visual evaluation protocol marks a significant step towards standardizing the assessment of sequential disentanglement models. Nevertheless, while DiffSDA shows promise in handling high-resolution videos and varied datasets, future research should focus on optimizing its computational efficiency and extending its applicability to more diverse sequence modalities, such as sensor data. Such modalities present unique challenges, as varying temporal characteristics and distinct data patterns, which may require adapting the model architecture and training strategies. In addition, given that our current video generation process operates frame-by-frame, potentially limiting spatio-temporal coherence, an interesting direction for future work is to integrate DiffSDA with latent video diffusion models, e.g., LVDM (He et al., 2022), or related architectures to further strengthen its generative fidelity. Finally, a key challenge ahead lies in fully extending our multifactor exploration procedure to effectively disentangle and represent multiple interacting factors (Berman et al., 2023). We leave these considerations and further explorations for future work.

### ACKNOWLEDGEMENTS

This research was partially supported by the Lynn and William Frankel Center of The Stein Faculty of Computer and Information Science, Ben-Gurion University of the Negev, ISF grants 668/21 and 1299/25, an ISF equipment grant, and by the Israeli Council for Higher Education (CHE) via the Data Science Research Center, Ben-Gurion University of the Negev, Israel.

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

# A BACKGROUND

## A.1 DIFFUSION MODELS

Diffusion models (Sohl-Dickstein et al., 2015) are a family of SOTA generative models, that were recently described using stochastic differential equations (SDEs), diffusion processes, and score-based modeling (Song et al., 2021). We will use diffusion models and score-based models interchangeably. These models include two processes: the forward process and the reverse process. The forward process (often not learnable) is an iterative procedure that corrupts the data by progressively adding noise to it. Specifically, the change to the state $\mathbf{x}_t$ can be formally described by

$$\mathrm{d}\mathbf{x}_t = \mathbf{f}(\mathbf{x}_t, t)\mathrm{d}t + g(t)\mathrm{d}\mathbf{w} , \tag{7}$$

where $\mathbf{w}$ is the standard Wiener process, $\mathbf{f}(\cdot, t)$ is a vector-valued function called the drift coefficient, and $g(\cdot)$ is a scalar function known as the diffusion coefficient. From a probabilistic viewpoint, Eq. 7 is associated with modeling the transition from the given data distribution, $\mathbf{x}_0 \sim p_0$, to $p_t$, the probability density of $\mathbf{x}_t$, $t \in [0, T]$. Typically, the prior distribution $p_T$ is a simple Gaussian distribution with fixed mean and variance that contains no information of $p_0$. The reverse process, which is learnable, de-noises the data iteratively. The reverse of a diffusion process is also a diffusion process, depending on the score function $\nabla_{\mathbf{x}} \log p_t(\mathbf{x})$ and operating in reverse time (Anderson, 1982). In our approach, we utilize the conditioned reverse process

$$\mathrm{d}\mathbf{x}_t = [\mathbf{f}(\mathbf{x}_t, t) - g(t)^2 \nabla_{\mathbf{x}} \log p_t(\mathbf{x}_t \mid \mathbf{u})]\mathrm{d}\bar{t} + g(t)\mathrm{d}\bar{\mathbf{w}}, \tag{8}$$

where $\bar{\mathbf{w}}$ is a standard Wiener process as time progresses backward from $T$ to 0, $\mathrm{d}\bar{t}$ is an negative timestep, and $\mathbf{u}$ is a condition variable. Diffusion models are generative by sampling from $p_T$ and use $\nabla_{\mathbf{x}} \log p_t(\mathbf{x}_t \mid \mathbf{u})$ to iteratively solve Eq. 8 until samples from $p_0$ are recovered.

## A.2 DIFFUSION AUTOENCODERS

Although diffusion models are powerful generative tools, they are not inherently designed to learn meaningful representations of the data. To address this limitation, several works (Preechakul et al., 2022; Wang et al., 2023) have adapted diffusion models into autoencoders, resulting in diffusion autoencoders (DiffAEs). These models have demonstrated the ability to learn semantic representations of the data, allowing certain modifications of the resulting samples by altering their latent vectors. To this end, DiffAEs introduce a semantic encoder, taking a data sample $x_0$ and returning its semantic latent encoding $z_{\text{sem}}$. Then, the latter vector conditions the reverse process, enhancing the model's ability to reconstruct and manipulate data samples. In practice, the denoiser is also conditioned on a feature map $h$ and the time $t$, combined using an adaptive group normalization (AdaGN) layer (Dhariwal & Nichol, 2021). The AdaGN block is defined as

$$\text{AdaGN}(h, t, z_{\text{sem}}) = z_s \left( t_s \, \text{GroupNorm}(h) + t_b \right) , \tag{9}$$

where $z_s$ is the output of a linear layer applied to $z_{\text{sem}}$, $t_s$ and $t_b$ are the outputs of a multi-layer perceptron (MLP) applied to the time $t$, and multiplications are done element-wise.

# B DIFFSDA MODELING

## B.1 UNSUPERVISED SEQUENTIAL DISENTANGLEMENT

Unsupervised sequential disentanglement is a challenging problem in representation learning, aiming to decompose a given dataset to its static (time-independent) and dynamic (time-dependent) factors of variation. Let $\mathcal{D} = \{\mathbf{x}_j^{1:V}\}_{j=1}^N$ be a dataset with $N$ sequences $\mathbf{x}_j^{1:V} := \{\mathbf{x}_j^1, \ldots, \mathbf{x}_j^V\}$, where $\mathbf{x}_j^\tau \in \mathbb{R}^d$. We omit the subscript $j$ for brevity, unless noted otherwise. The goal of sequential disentanglement is to extract an alternative representation of $\mathbf{x}^{1:V}$ via a single static factor $\mathbf{s}$ and multiple dynamic factors $\mathbf{d}^{1:V}$. Note that $\mathbf{s}$ is shared across the sequence.

We can formalize the sequential disentanglement problem as a *generative task*, where every sequence $\mathbf{x}^{1:V}$ from the data space $\mathcal{X}$ is conditioned on some $\mathbf{z}^{1:V}$ from a latent space $\mathcal{Z}$. We aim to maximize the probability of each sequence under the entire generative process

$$p(\mathbf{x}^{1:V}) = \int_{\mathcal{Z}} p(\mathbf{x}^{1:V} \mid \mathbf{z}^{1:V})p(\mathbf{z}^{1:V}) \, \mathrm{d} \, \mathbf{z}^{1:V} , \tag{10}$$

where $\mathbf{z}^{1:V} := (\mathbf{s}, \mathbf{d}^{1:V})$. One of the main challenges with directly maximizing Eq. (10) is that the latent space $\mathcal{Z}$ is too large to practically integrate over. Instead, a separate distribution, denoted here as $q(\mathbf{z}^{1:V} \mid \mathbf{x}^{1:V})$, is used to narrow search to be only over $\mathbf{z}^{1:V}$ associated with sequences from the dataset $\mathcal{D}$. Importantly, the distributions $p(\mathbf{x}^{1:V} \mid \mathbf{z}^{1:V})$ and $q(\mathbf{z}^{1:V} \mid \mathbf{x}^{1:V})$ take the form of a decoder and an encoder in practice, suggesting the development of *autoencoder* sequential disentanglement models (Yingzhen & Mandt, 2018). The above $p(\mathbf{x}^{1:V} \mid \mathbf{z}^{1:V})$ and $q(\mathbf{z}^{1:V} \mid \mathbf{x}^{1:V})$ are denoted by $p_{T0}(\mathbf{x}_0^\tau \mid \mathbf{x}_T^\tau, \mathbf{s}_0, \mathbf{d}_0^\tau)$ and $p(\mathbf{x}_t^{1:V}, \mathbf{s}_0, \mathbf{d}_0^{1:V} \mid \mathbf{x}_0^{1:V})$, respectively, in Eq. 1 and Eq. 2.

## B.2 High-Resolution Disentangled Sequential Diffusion Autoencoder

In addition to transitioning to real-world data, our goal is to manage high-resolution data for unsupervised sequential disentanglement, for the first time. Drawing inspiration from Rombach et al. (2022), we incorporate perceptual image compression, which combines an autoencoder with a perceptual loss (Zhang et al., 2018) and a patch-based adversarial objective (Dosovitskiy & Brox, 2016; Esser et al., 2021; Isola et al., 2017). Specifically, we explore two main variants of the autoencoder. The first variant applies a small Kullback–Leibler penalty to encourage the learned latent space to approximate a standard normal distribution, similar to a VAE (Kingma, 2013; Rezende et al., 2014). The second variant integrates a vector quantization layer (Van Den Oord et al., 2017; Razavi et al., 2019) within the decoder. Empirically, we find that the VQ-VAE-based model performs better when combined with our method. Given a pre-trained encoder $\mathcal{E}$ and decoder $\mathcal{D}$, we can extract $\mathbf{x}_0^\tau = \mathcal{E}(\mathrm{x}_0^\tau)$, which represents a low-dimensional latent space where high-frequency, imperceptible details are abstracted away. Finally, $\mathrm{x}_0^\tau$ can be reconstructed from the latent $\mathbf{x}_0^\tau$ by applying the decoder $\mathrm{x}_0^\tau = \mathcal{D}(\mathbf{x}_0^\tau)$. The EDM formulation in Eq. 4 makes relatively strong assumptions about the mean and standard deviation of the training data. To meet these assumptions, we opt to normalize the training data globally rather than adjusting the value of $\sigma_{\text{data}}$, which could significantly affect other hyperparameters (Karras et al., 2024). Therefore, we keep $\sigma_{\text{data}}$ at its default value of 0.5 and ensure that the latents have a zero mean during dataset preprocessing. When generating sequence elements, we reverse this normalization before applying $\mathcal{D}$.

## B.3 Prior Modeling

We model the prior static and dynamic distribution with $p_{T0}(\boldsymbol{s}_0, \boldsymbol{d}_0^{1:V} \mid \boldsymbol{s}_T, \boldsymbol{d}_T^{1:V})$. To sample static and dynamic factors, we train a separate latent DDIM model (Song et al., 2020). Then, we can extract the factors by sampling noise, and reversing the trained model. Specifically, we learn $p_{\Delta t}(\mathbf{z}_{t-1}^{1:V} \mid \mathbf{z}_t^{1:V})$ where $\mathbf{z}_0 = (\boldsymbol{s}_0, \boldsymbol{d}_0^{1:V})$ are the outputs of our sequential semantic encoder. The training is done by simply optimizing the $\mathcal{L}_{\text{latent}}$ with respect to DDIM's output $\varepsilon_\phi(\cdot)$:

$$\mathcal{L}_{\text{latent}} = \sum_{t=1}^{T} \mathbb{E}_{\mathbf{z}^{1:V}, \varepsilon_t} \left[ \|\varepsilon_\phi(\mathbf{z}_t^{1:V}, t) - \varepsilon_t\| \right] \tag{11}$$

where $\varepsilon_t \in \mathbb{R}^{dV+s} \sim \mathcal{N}(\mathbf{0}, \mathbf{I})$, $V$ is the sequence length, $s, d$ are the static and dynamic factors dimensions respectively. Additionally, $\mathbf{z}_t^{1:V}$ is the noise version of $\mathbf{z}_t$ as described in Song et al. (2020). For designing the architecture of our latent model, we follow Preechakul et al. (2022) and it is based on 10 MLP layers. Our network architecture and hyperparamters are provided in Tab. 8.

## B.4 Reverse processes

The detailed reverse sampling algorithm is provided in Alg. 1. We follow Karras et al. (2022) sampling techniques, however, each step in our reverse process is conditioned on the latent static and dynamic factors extracted by our sequential semantic encoder. As in Preechakul et al. (2022), we observe that auto-encoding is improved significantly when using the stochastic encoding technique. Since we have a different reverse process, we provide the algorithm for stochastic encoding for our modeling in Alg. 2. Finally, when performing conditional swapping, we observe that performing stochastic encoding on the sample from which we borrow the dynamics and using it as an input to Alg. 1, improves the results empirically. That is, given two sample videos $\mathbf{x}, \hat{\mathbf{x}} \sim p_0$, to create a new sample $\bar{\mathbf{x}}$, conditioned on the static factor of $\mathbf{x}$ and dynamic features of $\hat{\mathbf{x}}$, we use the stochastic encoding of $\hat{\mathbf{x}}$ in Alg. 1.

---

**Algorithm 1** Conditioned Stochastic Sampler with $\sigma(t) = t$ and $s(t) = 1$.

---

1: **procedure** CONDITIONEDSTOCHASTICSAMPLER($D_\theta$, $t_{i \in \{0,\dots,N\}}$, $\gamma_{i \in \{0,\dots,N-1\}}$, $\mathbf{z}_0^{1:V}$, $\mathbf{x}_0^{1:V}$, $S_{\text{noise}}^2$)
2:      **if** $\mathbf{x}_0^{1:V} \neq$ None **then**
3:          $\mathbf{x}_N^{1:V} \leftarrow$ **Algorithm 2 output**
4:      **else**
5:          **sample** $\mathbf{x}_N^{1:V} \sim \mathcal{N}(\mathbf{0}, \ t_N^2 \mathbf{I})$
6:      **for** $i \in \{N, \dots, 1\}$ **do**        $\triangleright \gamma_i = \begin{cases} \min\left(\frac{S_{\text{churn}}}{N}, \sqrt{2}-1\right) & \text{if } t_i \in [S_{\text{tmin}}, S_{\text{tmax}}] \\ 0 & \text{otherwise} \end{cases}$
7:          **sample** $\boldsymbol{\epsilon}_i \sim \mathcal{N}(\mathbf{0}, \ S_{\text{noise}}^2 \mathbf{I})$
8:          $\hat{t}_i \leftarrow t_i + \gamma_i t_i$        $\triangleright$ Select temporarily increased noise level $\hat{t}_i$
9:          $\hat{\mathbf{x}}_i^\tau \leftarrow \mathbf{x}_i^\tau + \sqrt{\hat{t}_i^2 - t_i^2}\, \boldsymbol{\epsilon}_i$        $\triangleright$ Add new noise to move from $t_i$ to $t_i$
10:          $\boldsymbol{d}_i \leftarrow (\mathbf{x}_i^\tau - D_\theta(\mathbf{x}_i^\tau, \mathbf{z}_0^\tau; \hat{t}_i))/\hat{t}_i$        $\triangleright$ Evaluate $\mathrm{d}\mathbf{x}/\mathrm{d}t$ at $t_i$
11:          $\mathbf{x}_{i-1}^\tau \leftarrow \mathbf{x}_i^\tau + (t_{i-1} - \hat{t}_i)\boldsymbol{d}_i$        $\triangleright$ Take Euler step from $t_i$ to $t_{i-1}$
12:          **if** $t_{i-1} \neq 0$ **then**
13:             $\boldsymbol{d}_i' \leftarrow (\mathbf{x}_{i-1}^\tau - D_\theta(\mathbf{x}_{i-1}^\tau, \mathbf{z}_0^\tau; t_{i-1}))/t_{i-1}$        $\triangleright$ Apply 2$^{\text{nd}}$ order correction
14:             $\mathbf{x}_{i-1}^\tau \leftarrow \hat{\mathbf{x}}_i^\tau + (t_{i-1} - \hat{t}_i)\left(\frac{1}{2}\boldsymbol{d}_i + \frac{1}{2}\boldsymbol{d}_i'\right)$
15:      **return** $\mathbf{x}_0$

---

**Algorithm 2** Stochastic Encoding with $\sigma(t) = t$ and $s(t) = 1$.

---

1: **procedure** STOCHASTICENCODER($D_\theta$, $t_{i \in \{0,\dots,N\}}$, $\gamma_{i \in \{0,\dots,N-1\}}$, $\mathbf{x}_0^{1:V}$, $\mathbf{z}_0^{1:V}$)
2:      **sample** $\boldsymbol{x}_0 \sim \mathcal{N}(\mathbf{0}, \ t_0^2 \mathbf{I})$
3:      **for** $i \in \{0, \dots, N-1\}$ **do**        $\triangleright \gamma_i = \begin{cases} \min\left(\frac{S_{\text{churn}}}{N}, \sqrt{2}-1\right) & \text{if } t_i \in [S_{\text{tmin}}, S_{\text{tmax}}] \\ 0 & \text{otherwise} \end{cases}$
4:          **sample** $\boldsymbol{\epsilon}_i \sim \mathcal{N}(\mathbf{0}, \ S_{\text{noise}}^2 \mathbf{I})$
5:          $\hat{t}_i \leftarrow t_i + \gamma_i t_i$        $\triangleright$ Select temporarily increased noise level $\hat{t}_i$
6:          $\hat{\boldsymbol{x}}_i \leftarrow \boldsymbol{x}_i + \sqrt{\hat{t}_i^2 - t_i^2}\, \boldsymbol{\epsilon}_i$        $\triangleright$ Add new noise to move from $t_i$ to $t_i$
7:          $\boldsymbol{d}_i \leftarrow (\mathbf{x}_i^\tau - D_\theta(\mathbf{x}_i^\tau, \mathbf{z}_0^\tau; t_i))/t_i$        $\triangleright$ Evaluate $\mathrm{d}\mathbf{x}^\tau/\mathrm{d}t$ at $t_i$
8:          $\mathbf{x}_{i+1}^\tau \leftarrow \mathbf{x}_i^\tau + (t_{i+1} - t_i)\boldsymbol{d}_i$        $\triangleright$ Take Euler step from $t_i$ to $t_{i+1}$
9:          **if** $t_{i+1} \neq \sigma_{\text{max}}$ **then**
10:             $\boldsymbol{d}_i' \leftarrow (\mathbf{x}_{i+1}^\tau - D_\theta(\mathbf{x}_{i+1}^\tau, \mathbf{z}_0^\tau; t_{i+1}))/t_{i+1}$        $\triangleright$ Apply 2$^{\text{nd}}$ order correction
11:             $\mathbf{x}_{i+1}^\tau \leftarrow \mathbf{x}_i^\tau + (t_{i+1} - t_i)\left(\frac{1}{2}\boldsymbol{d}_i + \frac{1}{2}\boldsymbol{d}_i'\right)$
12:      **return** $\mathbf{x}_N^{1:V}$

---

## C   HYPER-PARAMETERS

The hyperparameters used in our autoencoder are listed in Tab. 6 and Tab. 7, detailing the configurations for each dataset: MUG, TaiChi-HD, VoxCeleb, CelebV-HQ, TIMIT, LibriSpeech, PhysioNet, Air Quality and ETTh1. We provide the values of essential parameters such as sequence lengths, batch sizes, learning rates, and the use of $P_{\text{mean}}$ and $P_{\text{std}}$ to manage noise disturbance during training. In addition, the table specifies whether VQ-VAE was employed. Tab. 8 outlines the architecture of our latent DDIM model, including batch size, number of epochs, MLP layers, hidden sizes, and the $\beta$ scheduler. These details are essential for understanding the model's structure and its training process. For the VQ-VAE model, we utilized the pre-trained model from (Rombach et al., 2022) with hyperparameters $f = 8$, $Z = 256$, and $d = 4$, which encodes a frame of size $3 \times 256 \times 256$ into a latent representation of size $4 \times 32 \times 32$.

## D   DATASETS

**MUG.** The MUG facial expression dataset, introduced by Aifanti et al. (2010), contains image sequences from 52 subjects, each displaying six distinct facial expressions: anger, fear, disgust, happiness, sadness, and surprise. Each video sequence in the dataset ranges from 50 to 160 frames. To create sequences of length 15, as done in prior work (Bai et al., 2021), we randomly select 15 frames from the original sequences. We then apply Haar Cascade face detection to crop the faces and resize them to $64 \times 64$ pixels, resulting in sequences of $x \in \mathbb{R}^{15 \times 3 \times 64 \times 64}$. The final dataset comprises 3,429 samples. In the case of of the zero shot experiments we resize the images to $256 \times 256$ pixels.

Table 6: Hyperparameters for Video datasets.

| Dataset | MUG | TaiChi-HD | VoxCeleb | CelebV-HQ |
|---|---|---|---|---|
| $P_{\text{maen}}$ | $-1.2$ | $-1.2$ | $-0.4$ | $-0.4$ |
| $P_{\text{std}}$ | 1.2 | 1.2 | 1.0 | 1.0 |
| NFE | 71 | 63 | 63 | 63 |
| lr | 1e$-$4 | 1e$-$4 | 1e$-$4 | 1e$-$4 |
| bsz | 8 | 16 | 16 | 16 |
| #Epoch | 1600 | 40 | 100 | 450 |
| Dataset repeats | 1 | 150 | 1 | 1 |
| $s$ dim | 256 | 512 | 512 | 1024 |
| $d$ dim | 64 | 64 | 12 | 16 |
| hidden dim | 128 | 1024 | 1024 | 1024 |
| Base channels | 64 | 64 | 192 | 192 |
| Channel multipliers | $[1, 2, 2, 2]$ | | | |
| Attention placement | $[2]$ | | | |
| Encoder base ch | 64 | 64 | 192 | 192 |
| Encoder ch. mult. | $[1, 2, 2, 2]$ | | | |
| Enc. attn. placement | $[2]$ | | | |
| Input size | $3 \times 64 \times 64$ | $3 \times 64 \times 64$ | $3 \times 256 \times 256$ | $3 \times 256 \times 256$ |
| Seq len | 15 | 10 | 10 | 10 |
| Optimizer | AdamW (weight decay$= 1$e$-5$) | | | |
| Backbone | Unet | | | |
| GPU | 1 RTX 4090 | | 3 RTX 4090 | |

Table 7: Hyperparameters for audio and TS.

| Dataset | TIMIT | LibriSpeech | Physionet | Airq | ETTH |
|---|---|---|---|---|---|
| $P_{\text{maen}}$ | $-0.4$ | $-0.4$ | $-0.4$ | $-0.4$ | $-0.4$ |
| $P_{\text{std}}$ | 1.0 | 1.0 | 1.0 | 1.0 | 1.0 |
| NFE | 63 | 63 | 63 | 63 | 63 |
| lr | 1e$-$4 | 1e$-$3 | 5e$-$5 | 1e$-$4 | 1e$-$4 |
| bsz | 128 | 128 | 30 | 10 | 10 |
| #Epoch | 750 | 200 | 200 | 200 | 200 |
| $s$ dim | 32 | 32 | 24 | 16 | 16 |
| $d$ dim | 4 | 2 | 2 | 4 | 4 |
| hidden dim | 128 | 256 | 96 | 512 | 512 |
| Base channels | 256 | 64 | 256 | 256 | 128 |
| Channel multipliers | $[4, 4, 4, 4]$ | | | | |
| Attention placement | None | | | | |
| Encoder base ch | 128 | 128 | 96 | 128 | 256 |
| Encoder ch. mult. | $[4, 4, 4, 4]$ | | | | |
| Enc. attn. placement | None | | | | |
| Input size | 80 | 80 | 10 | 10 | 6 |
| Seq len | 68 | 68 | 80 | 672 | 672 |
| Optimizer | AdamW (weight decay$= 1$e$-5$) | | | | |
| Backbone | MLP | | | | |
| GPU | 1 RTX 4090 | | | | |

**TaiChi-HD.** The TaiChi-HD dataset, introduced by Siarohin et al. (2019), contains videos of full human bodies performing Tai Chi actions. We follow the original preprocessing steps from FOMM (Siarohin et al., 2019) and use a $64 \times 64$ version of the dataset. The dataset comprises 3,081 video chunks with varying lengths, ranging from 128 to 1,024 frames. We split the data into 90% for training and 10% for testing. To create sequences of length 10, similar to the approach used for the MUG dataset, we randomly select 10 frames from the original sequences. The resulting sequences are resized to $64 \times 64$ pixels, forming $x \in \mathbb{R}^{10 \times 3 \times 64 \times 64}$.

Table 8: Network architecture of our latent DDIM.

| Parameter | MUG | TaiChi-HD | VoxCeleb | Celebv-HQ |
|---|---|---|---|---|
| Batch size | 128 | 128 | 128 | 128 |
| #Epoch | 500 | 500 | 200 | 1000 |
| MLP layers ($N$) | | | 10 | |
| MLP hidden size | 1216 | 5008 | 2528 | 4736 |
| $\beta$ scheduler | | | Linear | |
| Learning rate | | | 1e$-$4 | |
| Optimizer | | AdamW (weight decay= 1e$-$5) | | |
| Train Diff T | | | 1000 | |
| Diffusion loss | | L2 loss with noise prediction $\epsilon$ | | |
| GPU | | | 1 RTX 4090 | |

**VoxCeleb.** The VoxCeleb dataset (Nagrani et al., 2017) is a collection of face videos extracted from YouTube. We used the preprocessing steps from Albanie et al. (2018), where faces are extracted, and the videos are processed at 25/6 fps. The dataset comprises 22,496 videos and 153,516 video chunks. We used the verification split, which includes 1,211 speakers in the training set and 40 different speakers in the test set, resulting in 148,642 video chunks for training and 4,874 for testing. To create sequences of length 10, we randomly select 10 frames from the original sequences. The videos are processed at a resolution of $256 \times 256$ resulting in sequences represented as $x \in \mathbb{R}^{10 \times 3 \times 256 \times 256}$.

**CelebV-HQ.** The CelebV-HQ dataset (Zhu et al., 2022) is a large-scale collection of high-quality video clips featuring faces, extracted from various online sources. The dataset consists of 35,666 video clips involving 15,653 identities, with each clip manually labeled with 83 facial attributes, including 40 appearance attributes, 35 action attributes, and 8 emotion attributes. The videos were initially processed at a resolution of $512 \times 512$. We then used Wang et al. (2021) to crop the facial regions, resulting in videos at a $256 \times 256$ resolution. To create sequences of length 10, we randomly selected 10 frames from the original sequences, producing sequences represented as $x \in \mathbb{R}^{10 \times 3 \times 256 \times 256}$.

**TIMIT.** The TIMIT dataset, introduced by Garofolo (1993), is a collection of read speech designed for acoustic-phonetic research and other speech-related tasks. It contains 6300 utterances, totaling approximately 5.4 hours of audio recordings, from 630 speakers (both men and women). Each speaker contributes 10 sentences, providing a diverse and comprehensive pool of speech data. To pre-process the data we use mel-spectogram feature extraction with 8.5ms frame shift applied to the audio. Subsequently, segments of 580ms duration, equivalent to 68 frames, are sampled from the audio and treated as independent samples.

**LibriSpeech.** The LibriSpeech dataset Panayotov et al. (2015) is a corpus of read English speech derived from audiobooks, containing 1,000 hours of speech sampled at 16 kHz. For our training, we used the `train-clean-360` subset, which consists of 363.6 hours of speech from 921 speakers. As validation and test sets, we use `dev-clean` and `test-clean`, each containing 5.4 hours of speech from 40 unique speakers, where there is no identity overlap across all subsets. For pre-processing, we extract mel-spectrogram features with an 8.5 ms frame shift applied to the audio. We then sample segments of 580 ms duration (equivalent to 68 frames) from the audio, treating them as independent samples.

**PhysioNet.** The PhysioNet ICU dataset (Goldberger et al., 2000) consists of medical time series data collected from 12,000 adult patients admitted to the Intensive Care Unit (ICU). This dataset includes time-dependent measurements such as physiological signals, laboratory results, and relevant patient demographics like age and reasons for ICU admission. Additionally, labels indicating in-hospital mortality events are included. Our preprocessing procedures follow the guidelines provided in (Tonekaboni et al., 2022).

**Air Quality.** The UCI Beijing Multi-site Air Quality dataset (Zhang et al., 2017) comprises hourly records of air pollution levels, collected over a four-year period from March 1, 2013, to February 28,

2017, across 12 monitoring sites in Beijing. Meteorological data from nearby weather stations of the China Meteorological Administration is also included. Our approach to data preprocessing, as described in (Tonekaboni et al., 2022), involves segmenting the data based on different monitoring locations and months of the year.

**ETTh1.** The ETTh1 dataset is a subset of the Electricity Transformer Temperature (ETT) dataset, containing hourly data over a two-year period from two counties in China. The dataset is focused on Long Sequence time series Forecasting (LSTF) of transformer oil temperatures. Each data point consists of the target value (oil temperature) and six power load features. The dataset is divided into training, validation, and test sets, with a 12/4/4-month split.

# E  METRICS

**Average Keypoint Distance (AKD).**    To evaluate whether the motion in the reconstructed video is preserved, we utilize pre-trained third-party keypoint detectors on the TaiChi-HD, VoxCeleb, CelebV-HQ, and MUG datasets. For the VoxCeleb, CelebV-HQ and MUG datasets, we employ the facial landmark detector from Bulat & Tzimiropoulos (2017), whereas for the TaiChi-HD dataset, we use the human-pose estimator from Cao et al. (2017). Keypoints are computed independently for each frame. AKD is calculated by averaging the $L_1$ distance between the detected keypoints in the ground truth and the generated video. The TaiChi-HD and MUG datasets are evaluated at a resolution of $64 \times 64$ pixels, and the VoxCeleb and CelebV-HQ datasets at $256 \times 256$ pixels. If the model output is at a lower resolution, it is interpolated to $256 \times 256$ pixels for evaluation.

**Average Euclidean Distance (AED).**    To assess whether the identity in the reconstructed video is preserved, we use the Average Euclidean Distance (AED) metric. AED is calculated by measuring the Euclidean distance between the feature representations of the ground truth and the generated video frames. We selected the feature embedding following the example set in Siarohin et al. (2019). For the VoxCeleb, CelebV-HQ, and MUG datasets, we use a VGG-FACE for facial identification using the framework of Serengil & Ozpinar (2020), whereas for TaiChi-HD, we use a network trained for person re-identification (Hermans et al., 2017). TaiChi-HD and MUG are evaluated at a resolution of $64 \times 64$ pixels, and VoxCeleb and CelebV-HQ at $256 \times 256$ pixels.

To ensure fairness when measuring AED and AKD, we created a predefined dataset of example pairs, ensuring that all models are evaluated on the exact same set of pairs. This is important because when measuring quantitative metrics, the results may vary depending on the dynamics swapped between two subjects, as e.g., the key points in AKD in the original video are influenced by the identity of the person. To address this issue, we establish a fixed set of pairs for a consistent comparison across all methods.

**Accuracy (Acc).**    As in Naiman et al. (2023), we used this metric for the MUG dataset to evaluate a model's ability to preserve fixed features while generating others. For example, dynamic features are frozen while static features are sampled. Accuracy is computed using a pre-trained classifier, referred to as the "judge", which is trained on the same training set as the model and tested on the same test set. For the MUG dataset, the classifier checks that the facial expression remains unchanged during the sampling of static features.

**Inception Score (IS).**    The Inception Score is a metric used to evaluate the performance of the model generation. First, we apply the judge, to all generated videos $x_0^{1:V}$, obtaining the conditional predicted label distribution $p\left(y|x^{1:V}\right)$. Next, we compute $p(y)$, the marginal predicted label distribution, and calculate the KL-divergence $\mathrm{KL}\left[p\left(y|x_0^{1:V}\right)\|p(y)\right]$. Finally, the Inception Score is computed as $\mathrm{IS} = \exp\left(\mathbb{E}_x\mathrm{KL}\left[p\left(y|x_0^{1:V}\right)\|p(y)\right]\right)$. We use this metric evaluate our results on MUG dataset.

**Inter-Entropy** $(H(y|x_0^{1:V}))$**.**    This metric reflects the confidence of the judge in its label predictions, with lower inter-entropy indicating higher confidence. It is calculated by passing $k$ generated sequences $\{x_0^{1:V}\}^{1:k}$ into the judge and computing the average entropy of the predicted label distributions: $\frac{1}{k}\sum_{i=1}^{k} H(p(y|\{x_0^{1:V}\}^i))$. We use this metric evaluate our results on MUG dataset.

**Intra-Entropy** ($H(y)$). This metric measures the diversity of the generated sequences, where a higher intra-entropy score indicates greater diversity. It is computed by sampling from the learned prior distribution $p(y)$ and then applying the judge to the predicted labels $y$. We use this metric to evaluate our results on the MUG dataset.

**EER.** Equal Error Rate (EER) metric is widely employed in speaker verification tasks. The EER represents the point at which the false positive rate equals the false negative rate, offering a balanced measure of performance in speaker recognition. This metric, commonly applied to the TIMIT dataset, provides a robust evaluation of the model's ability to disentangle features relevant to speaker identity.

**AUPRC.** The Area Under the Precision-Recall Curve (AUPRC) is a metric that evaluates the balance between precision and recall by measuring the area beneath their curve. A higher AUPRC reflects superior model performance, with values nearing 1 being optimal, indicating both high precision and recall.

**AUROC.** The Area Under the Receiver Operating Characteristic Curve (AUROC) measures the trade-off between true positive rate (TPR) and false positive rate (FPR), quantifying the area under the curve of these rates. A higher AUROC signifies better performance, with values close to 1 being desirable, representing a model that distinguishes well between positive and negative classes.

**MAE.** Mean Absolute Error (MAE) calculates the average magnitude of errors between predicted and observed values, offering a simple and intuitive measure of model accuracy. As it computes the average absolute difference between predicted and actual values, MAE is resistant to outliers and provides a clear indication of the model's prediction precision.

**DNSMOS.** Deep Noise Suppression Mean Opinion Score (DNSMOS (Reddy et al., 2021)) is a neural network-based metric introduced to estimate the perceptual quality of speech processed by noise suppression algorithms. Trained to predict human Mean Opinion Scores (MOS), DNSMOS provides a no-reference quality assessment that correlates strongly with subjective human judgments. It evaluates both the speech quality and the effectiveness of noise reduction, offering a comprehensive measure of audio clarity and intelligibility. This metric is especially useful in evaluating real-world performance of speech enhancement systems without the need for costly and time-consuming human listening tests.

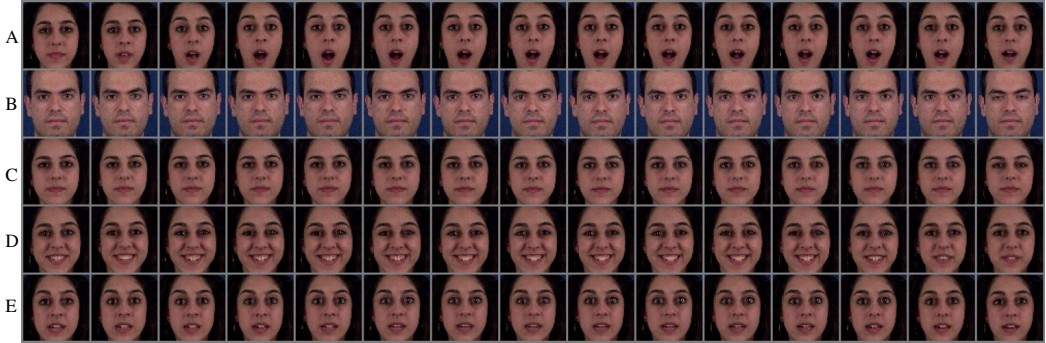

Figure 5: Rows A and B are two inputs from the test set. Row C shows a dynamic swap example, using the static of A and dynamics of B. In row D we extract the same person from A, but with the dynamics as labeled in B. Finally, in row E, we extract the same person from A with the dynamics that are predicted by the classifier.

## F   MUG AND JUDGE METRIC ANALYSIS

While our results show significant improvement over previous methods on VoxCeleb (Nagrani et al., 2017), CelebV-HQ (Zhu et al., 2022), and TaiChi-HD (Siarohin et al., 2019), both in terms of disentanglement and reconstruction, our performance on MUG (Aifanti et al., 2010) is only on par

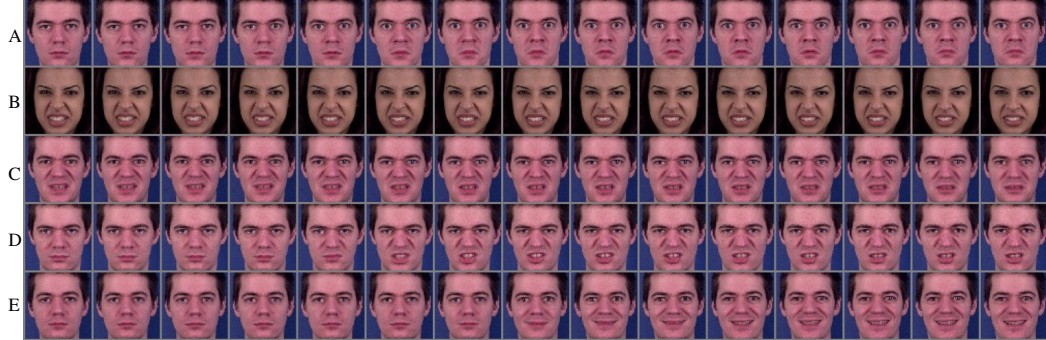

Figure 6: Rows A and B are two inputs from the test set. Row C shows a dynamic swap example, using the static of A and dynamics of B. In row D we extract the same person from A, but with the dynamics as labeled in B. Finally, in row E, we extract the same person from A with the dynamics that are predicted by the classifier.

Table 9: Judge benchmark disentanglement metrics on MUG.

| Method | Acc↑ | IS↑ | $H(y|x)\downarrow$ | $H(y)\uparrow$ | Reconstruction (MSE)↓ |
|--------|------|-----|--------|--------|-----------------------|
| | | | | MUG | |
| MoCoGAN | 63.12% | 4.332 | 0.183 | 1.721 | – |
| DSVAE | 54.29% | 3.608 | 0.374 | 1.657 | – |
| R-WAE | 71.25% | 5.149 | 0.131 | 1.771 | – |
| S3VAE | 70.51% | 5.136 | 0.135 | 1.760 | – |
| SKD | 77.45% | 5.569 | 0.052 | 1.769 | – |
| C-DSVAE | 81.16% | 5.341 | 0.092 | 1.775 | – |
| SPYL | 85.71% | 5.548 | 0.066 | 1.779 | 1.311e−3 |
| DBSE | **86.90%** | **5.598** | **0.041** | **1.782** | 1.286e−3 |
| Ours | 81.15% | 5.382 | 0.090 | 1.773 | **2.669e−7** |

with the state-of-the-art methods. Since MUG is a labeled dataset, the traditional evaluation task involves the unconditional generation of static factors while freezing the dynamics, resulting in altering the appearance of the person. The generated samples are then evaluated using an off-the-shelf judge model (See App. E), which is a neural network trained to classify both static and dynamic factors. If the disentanglement method disentangles these factors effectively, we expect the judge to correctly identify the dynamics while outputting different predictions for the static features, since the latter were randomly sampled and should differ from the original static factor.

Surprised by our results on MUG, we investigated the failure cases to understand the limitations of our model. In particular, we examined scenarios where we freeze the dynamics and swap the static features between two samples, and then we generate the corresponding output. In Fig. 5, we show an example where the static features of the second row are swapped with those of the first row, and the resulting generation is displayed in the third row. We observe that while the dynamics from the second row are well-preserved, the generated person retains the identity of the first row. However, the classifier incorrectly predicts the dynamics for the sequence. To further investigate this, we extracted a ground-truth example of the person from the first row in the dataset expressing the expected emotion and the predicted one. In the last two rows of Fig. 5, we show the same person with predicted dynamics (fourth row) and the same person with the dynamics that the classifier predicted (fifth row). We provide another example of the same phenomenon in Fig. 6.

We observe that while the judge predicts the wrong label for our generated samples in rows C, the facial expressions of the people there align better with the actual dynamics in rows B. This suggests that the classifier is biased towards the identity when predicting dynamics, potentially forming a discrete latent space where generalization to nearby related expressions is not possible. Importantly, the judge attains > 99% accuracy on the test set. We conclude that utilizing a judge can be problematic for measuring new and unseen variations in the data. This analysis motivates us to present the AKD and AED, as detailed above in App. E.

## G  ADDITIONAL EXPERIMENTS

### G.1  DEPENDENT VS. INDEPENDENT PRIOR MODELING

In Sec. 3, we describe our approach to prior modeling, highlighting our decision to generate latent factors dependently rather than independently, as done in previous state-of-the-art methods. Beyond being a parameter- and time-efficient choice, we empirically validate the advantages of our approach in the following experiment.

In this experiment, we compare two setups: (1) dependent generation of static and dynamic latent vectors, and (2) independent generation of these latent vectors using two latent DDIM models: one for the static vector and another for the dynamic vectors. To quantitatively assess the effectiveness of both approaches, we measure the FrÃ©chet Video Distance (FVD) Blattmann et al. (2023), a metric derived from the well-established FID score for videos. This metric evaluates how well a generative model captures the observed data distribution, where lower scores indicate better performance.

We conduct our evaluation on the VoxCeleb dataset, training two latent models. The independent model achieves an FVD score of 75.03, whereas our dependent approach achieves a significantly lower score of **65.23**, representing a $\approx 13\%$ improvement. This result underscores the expressive advantage of modeling latent factors dependently.

### G.2  ADDITIONAL ANALYSIS OF DIFFSDA DISENTANGLEMENT COMPONENTS

This section explores the impact of two key components of our method on disentanglement quality: i) the static latent factor $s_0$ shared across all time steps $\tau$, and ii) the dimensionality of the dynamic latent factor $d_0^\tau$.

To analyze these effects, we trained four models on the VoxCeleb dataset for 100 epochs, maintaining a static latent dimension of 128 while varying the size of the dynamic latent factor and whether the static latent factor was shared or not. The models were evaluated using our conditional swapping protocol and a verification metric based on the VGG-FACE framework proposed in Serengil & Ozpinar (2020). Specifically, we assessed identity consistency by freezing the static factor and swapping the dynamic factor, with the verification score representing the percentage of cases where identity was correctly preserved across frames.

As shown in Tab.10, our results indicate that the optimal performance (first row of the table) is achieved when $d_0^\tau$ has a smaller dimensionality, and the static factor is shared. Other configurations reveal significant trade-offs: increasing $d_0^\tau$ dimensionality results in higher AED scores but reduced verification accuracy, indicating weaker disentanglement of the static factor. Similarly, when $s_0$ is not shared, the AKD score degrades significantly, suggesting ineffective disentanglement of the dynamic factor. These findings underscore the importance of both (i) and (ii) in achieving robust sequential disentanglement.

Table 10: Disentanglement effect of VoxCeleb dataset

| $d_0^\tau$ size | $s$ shared? | Verification ACC ↑ (Static Frozen) | AED ↓ (static frozen) | AKD ↓ (dynamics frozen) |
|---|---|---|---|---|
| 16 | ✓ | **64.36%** | 0.925 | 2.882 |
| 128 | ✓ | 18.03% | 1.054 | **2.077** |
| 16 | ✗ | 56.75% | **0.898** | 12.64 |
| 128 | ✗ | 48.41% | 0.980 | 12.28 |

To strengthen these observations, we repeated the study on MUG, where ground-truth static (identity) and dynamic (action) labels allow a clean swap-task evaluation (Tab. 11). The trends mirror VoxCeleb: (i) sharing the static representation $s$ is critical when $s$ is not shared, the dynamic pathway collapses. In this case, identity remains stably preserved under static swap (as it should), but dynamic recognition deteriorates toward chance because temporal variation is absorbed into or suppressed by the static channel, indicating failed sequential disentanglement; and (ii) constraining the dimensionality of the dynamic latent $d_0^\tau$ provides a helpful bottleneck that limits identity leakage and sharpens the separation between static and dynamic factors. Although the absolute gaps on MUG are more modest than in Tab. 10, the qualitative agreement across datasets reinforces that both sharing $s$ and limiting the capacity of $d_0^\tau$ are key for robust sequential disentanglement.

Table 11: Disentanglement effect of MUG dataset

| $d_0^\tau$ size | $s$ shared? | Verification ACC ↑ (Static Frozen) | Action ACC ↓ (dynamics frozen) |
|---|---|---|---|
| 64 | ✓ | 95.59% | 80.08% |
| 256 | ✓ | 92.12% | 81.28% |
| 64 | ✗ | 99.69% | 16.71% |
| 256 | ✗ | 99.83% | 18.18% |

### G.3 SPEECH QUALITY AND RECONSTRUCTION COMPARISON

This section discusses the results of speech reconstruction and quality evaluation presented in table 12 on the LibriSpeech dataset. We compare the reconstruction performance using the Mean Squared Error (MSE) on the spectrograms and assess speech quality using the Deep Noise Suppression Mean Opinion Score (DNSMOS) (Reddy et al., 2021). The DNSMOS metric has a maximum score of 5, but the original (reference) dataset achieves a score of 3.9, as shown in the REF row of the table. As can be seen in the table, our model outperforms all comparable methods, achieving the lowest MSE and the highest DNSMOS among the evaluated approaches.

Table 12: Disentanglement and generation quality metrics on Libri Speech. For generation quality, we report MSE on the spectogram and Deep Noise Suppression Mean Opinion Score (DNSMOS).

| | Method | MSE↓ | DNSMOS↑ |
|---|---|---|---|
| | REF | –– | 3.9 |
| Libri Speech | DSVAE | 5.53e−2 | 3.13 |
| | SPYL | 4.40e−1 | 2.21 |
| | DBSE | 6.72e−3 | 2.88 |
| | Ours | **1.83e−4** | **3.41** |

### G.4 GENERATIVE QUALITY COMPRESSION

This section discusses the generative quality results shown in Table 13, evaluated using the Fréchet Video Distance (FVD) on the VoxCeleb dataset. We generated the same number of samples as in the test set and computed the FVD score against the test set. This process was repeated five times for each model using different five diffrent seeds to obtain a robust estimate. We report the mean FVD along with the standard deviation. The results demonstrate that our model outperforms existing state-of-the-art sequential disentanglement models in the video generation task.

Table 13: Fréchet Video Distance (FVD) results on VoxCeleb dataset to assess video generation quality. All experiments were conducted across five different random seeds to ensure robustness and account for variability in generation.

| Model | FVD↓ |
|---|---|
| SPYL | $582.28 \pm 1.15$ |
| DBSE | $1076.44 \pm 2.22$ |
| Ours | $\mathbf{65.23 \pm 0.81}$ |

### G.5 EFFECT OF VQ-VAE ON ZERO-SHOT SWAPS

In this subsection, we provide further details regarding the experiment aimed at evaluating the role of the VQ-VAE in zero-shot cross-dataset transfer. The experiment was designed to isolate the contribution of the VQ-VAE by removing it entirely and training DiffSDA directly in pixel space on

downsampled VoxCeleb. This setup allows us to examine how well the model maintains disentanglement when the unified latent space provided by VQ-VAE is absent. The observed deterioration in generalization, particularly in the stability and consistency of identity and expression transfer, indicates that the VQ-VAE plays a crucial role in producing coherent cross-dataset representations that support strong zero-shot disentanglement as seen in the examples of Fig. 7.

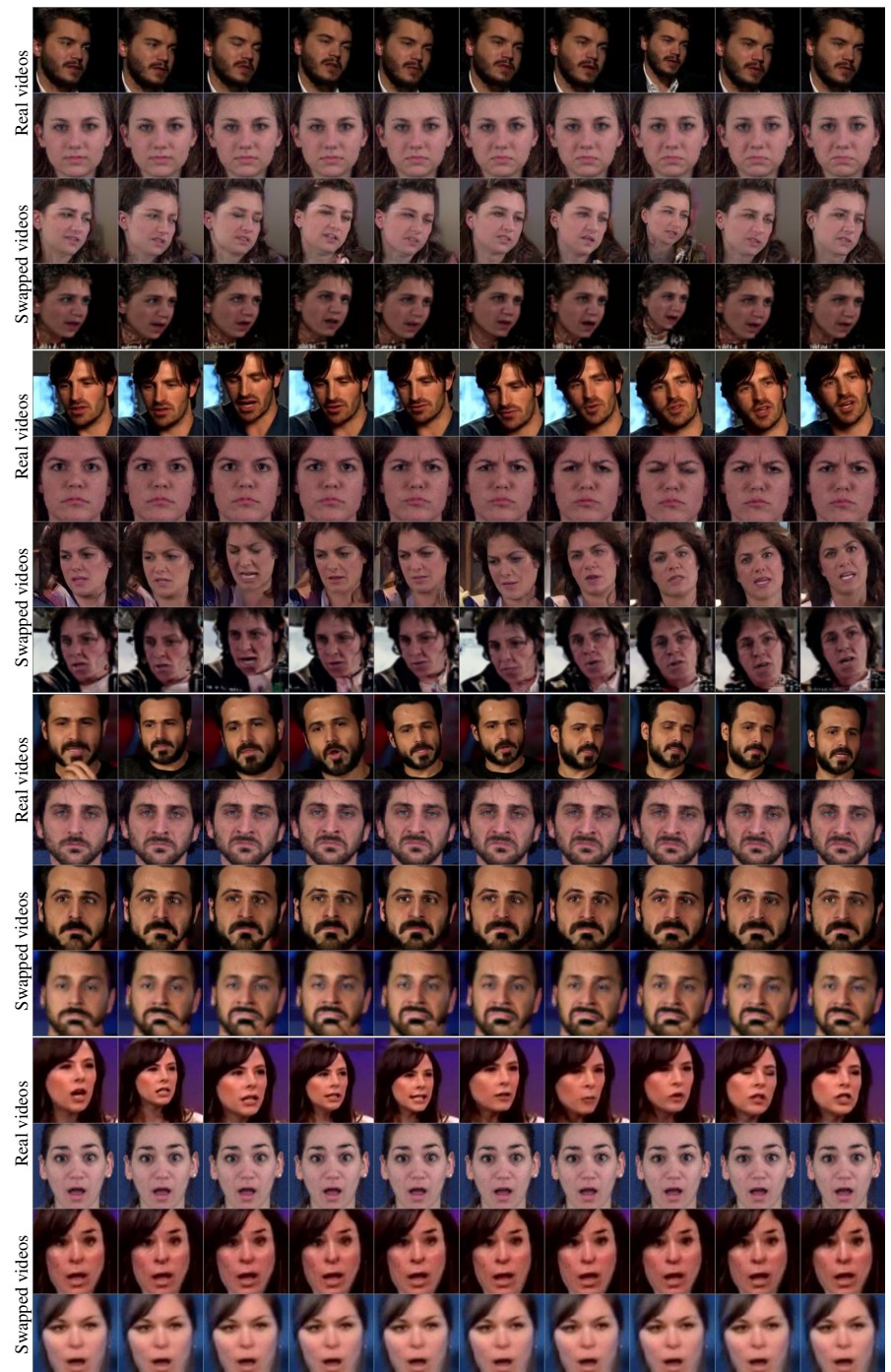

Figure 7: Each panel shows, in the first and second rows, real video pairs from the VoxCeleb and MUG datasets, respectively. We perform conditional swapping using two models: one trained on VoxCeleb with VQ-VAE and another trained on VoxCeleb without VQ-VAE at a resolution of $64 \times 64$. The resulting swaps are shown in the final two rows. In the first two examples, the dynamics are taken from the VoxCeleb video, while in the last two examples, the dynamics come from the MUG video.

# H ADDITIONAL RESULTS

## H.1 EXTENDED BENCHMARK RESULTS

In this section, we expand the comparisons from the main paper by adding results for additional baselines on the same tasks and datasets. Specifically, we include time-series prediction on PhysioNet and ETTh1 (Tab. 14), time-series classification on PhysioNet and Air Quality using only the static latents (Tab. 15), and disentanglement on TIMIT reported as Static EER, Dynamic EER, and Disentanglement Gap (Tab. 16). Across these additions, our method remains competitive or superior to the added baselines. We also add CDSVAE results on MUG and provide a MUG-only summary (Tab. 17): under swaps, our model better preserves identity (with $\mathbf{z}^s$ frozen) and motion (with $\mathbf{z}^d$ frozen), and in reconstruction achieves uniformly lower AED/AKD/MSE–supporting stronger disentanglement and higher-fidelity reconstructions.

Table 14: Time series prediction benchmark.

| | PhysioNet | | ETTh1 |
| Method | AUPRC ↑ | AUROC ↑ | MAE ↓ |
| --- | --- | --- | --- |
| VAE | $0.157 \pm 0.05$ | $0.564 \pm 0.04$ | $13.66 \pm 0.20$ |
| GP-VAE | $0.282 \pm 0.09$ | $0.699 \pm 0.02$ | $14.98 \pm 0.41$ |
| C-DSVAE | $0.158 \pm 0.01$ | $0.565 \pm 0.01$ | $12.53 \pm 0.88$ |
| GLR | $0.365 \pm 0.09$ | $0.752 \pm 0.01$ | $12.27 \pm 0.03$ |
| SPYL | $0.367 \pm 0.02$ | $0.764 \pm 0.04$ | $12.22 \pm 0.03$ |
| DBSE | $0.473 \pm 0.02$ | $0.858 \pm 0.01$ | $11.21 \pm 0.01$ |
| Ours | $\mathbf{0.50 \pm 0.006}$ | $\mathbf{0.87 \pm 0.004}$ | $\mathbf{9.89 \pm 0.280}$ |
| RF | $0.446 \pm 0.04$ | $0.802 \pm 0.04$ | $10.19 \pm 0.20$ |

Table 15: Time series classification benchmark.

| Method | PhysioNet ↑ | Air Quality ↑ |
| --- | --- | --- |
| VAE | $34.71 \pm 0.23$ | $27.17 \pm 0.03$ |
| GP-VAE | $42.47 \pm 2.02$ | $36.73 \pm 1.40$ |
| C-DSVAE | $32.54 \pm 0.00$ | $47.07 \pm 1.20$ |
| GLR | $38.93 \pm 2.48$ | $50.32 \pm 3.87$ |
| SPYL | $46.98 \pm 3.04$ | $57.93 \pm 3.53$ |
| DBSE | $56.87 \pm 0.34$ | $65.87 \pm 0.01$ |
| OUR | $\mathbf{64.6 \pm 0.35}$ | $\mathbf{69.2 \pm 1.50}$ |
| RF | $62.00 \pm 2.10$ | $62.43 \pm 0.54$ |

Table 16: Disentanglement metrics on TIMIT

| Method | Static EER↓ | Dynamic EER↑ | Dis. Gap↑ |
| --- | --- | --- | --- |
| FHVAE | $5.06\%$ | $22.77\%$ | $17.71\%$ |
| DSVAE | $5.64\%$ | $19.20\%$ | $13.56\%$ |
| R-WAE | $4.73\%$ | $23.41\%$ | $18.68\%$ |
| S3VAE | $5.02\%$ | $25.51\%$ | $20.49\%$ |
| SKD | $4.46\%$ | $26.78\%$ | $22.32\%$ |
| C-DSVAE | $4.03\%$ | $31.81\%$ | $27.78\%$ |
| SPYL | $\mathbf{3.41}\%$ | $33.22\%$ | $29.81\%$ |
| DBSE | $3.50\%$ | $34.62\%$ | $31.11\%$ |
| Ours | $4.43\%$ | $\mathbf{46.72}\%$ | $\mathbf{42.29}\%$ |

Table 17: MUG results only. Preservation of objects (AED) and motions (AKD) under conditional swapping, and reconstruction errors (AED/AKD/MSE). Labels 'static frozen' and 'dynamics frozen' correspond to samples $\mathbf{z}^s$ and $\mathbf{z}^d$.

|  | CDSVAE | SPYL | DBSE | Ours |
| --- | --- | --- | --- | --- |
| Swap |  |  |  |  |
| AED $\downarrow$ (static frozen) | 0.774 | 0.766 | 0.773 | **0.751** |
| AKD $\downarrow$ (dynamics frozen) | 1.170 | 1.132 | 1.118 | **0.802** |
| Reconstruction |  |  |  |  |
| AED $\downarrow$ | 0.56 | 0.49 | 0.49 | **0.11** |
| AKD $\downarrow$ | 0.50 | 0.47 | 0.48 | **0.06** |
| MSE $\downarrow$ | 0.001 | 0.001 | 0.001 | **3e−7** |

## H.2 RECONSTRUCTION RESULTS

In Figs. 8 to 11, we present several qualitative reconstruction examples across all datasets.

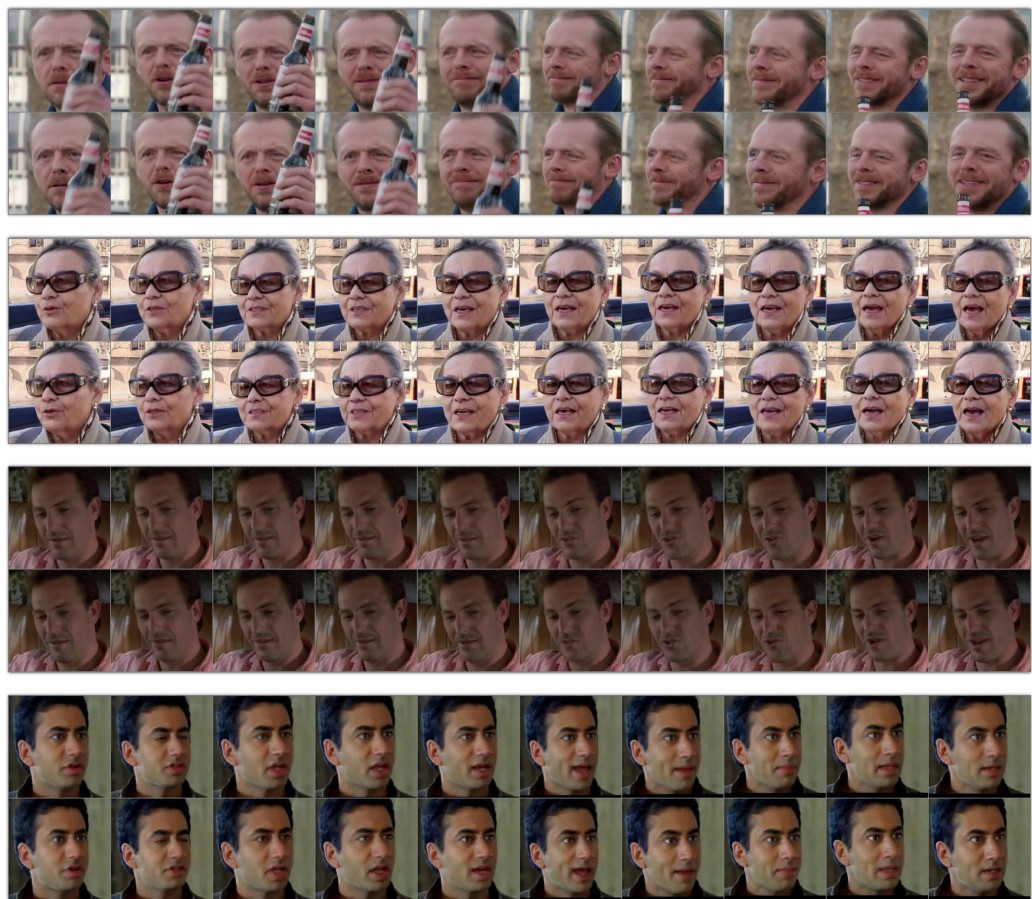

Figure 8: Reconstruction results of CelebV-HQ ($256 \times 256$). The first row for each pair is the original video and the second row is its reconstruction.

## H.3 ADDITIONAL RESULTS: CONDITIONAL SWAP

In what follows, we present more results for the conditional swapping experiment from the main text (Sec. 4.1). In each figure, the first two rows show the original sequences (real videos). The third and

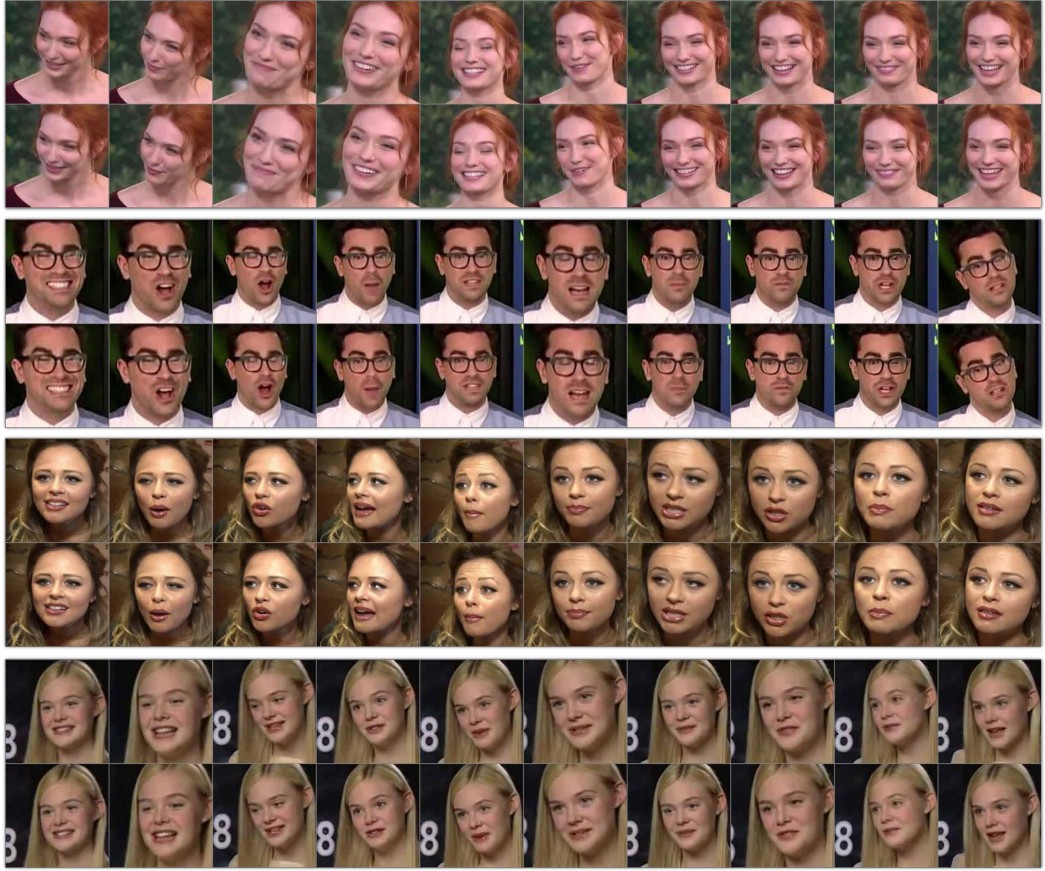

Figure 9: Reconstruction results of VoxCeleb ($256 \times 256$). The first row for each pair is the original video and the second row is its reconstruction.

fourth rows are the results of the conditional swap where we change the dynamic and static factors, respectively. We show our results for all datasets in Figs. 12 to 15.

## H.4 ADDITIONAL RESULTS: UNCONDITIONAL SWAP

In addition to the conditional and zero-shot shot tasks considered above, we can also perform such tasks in an unconditional manner. Specifically, given a real sequence $\mathbf{x}^{1:V}$ with its factors $(s, d^{1:V})$, we can unconditionally sample new $(\hat{s}, \hat{d}^{1:V})$ using our separate DDIM model (see Sec. 3). We then reconstruct the static swap $(\hat{s}, d^{1:V})$ and the dynamic swap $(s, \hat{d}^{1:V})$ similarly as described above. In Fig. 16, we present unconditional swap results on CelebV-HQ (left), VoxCeleb (middle), and TaiChi-HD (right). The middle rows represent the original sequences, whereas the top and bottom rows demonstrate dynamic and static swaps, respectively. Across all datasets and swap settings, our approach succeeds in modifying the swapped features while preserving the frozen factors, either in the static or in the dynamic examples. In addition, we also present more results where each figure is composed of separate panels. In each panel, the middle row represents the original sequence. In the top row, we sample new dynamic factors and freeze the static factor. In the bottom row below, we sample a new static factor and freeze the dynamics. We show our results on all datasets in Figs. 17 to 20.

## H.5 ADDITIONAL RESULTS: ZERO-SHOT DISENTANGLEMENT

Here we extend the results from Sec. 4.2. We provide additional examples of conditional swapping when the model is trained on one dataset and evaluated on another dataset, unseen during training.

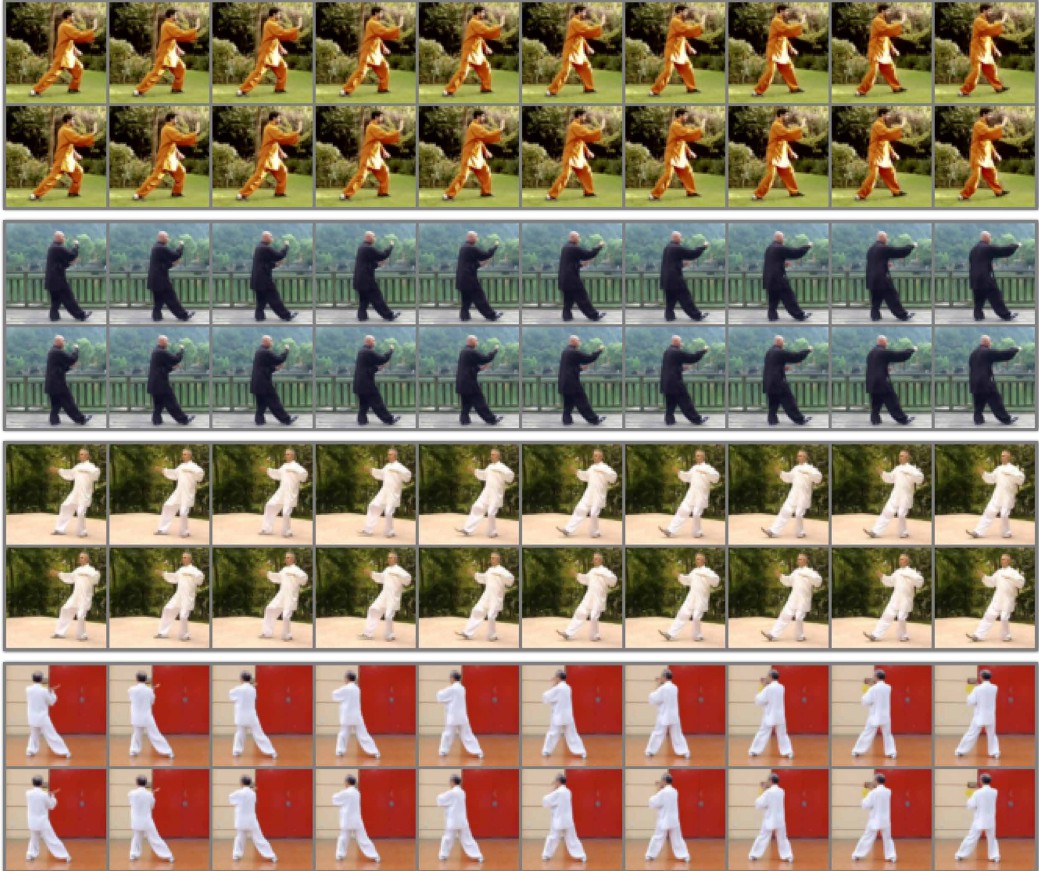

Figure 10: Reconstruction results of TaiChi-HD. The first row for each pair is the original video and the second row is its reconstruction.

Specifically, in Fig. 21, we show examples where the model is trained on VoxCeleb and tested on MUG. Additionally, in Fig. 22, the model is trained on VoxCeleb and tested on CelebV-HQ. Finally, in Fig. 23, the model is trained on CelebV-HQ and tested on VoxCeleb.

### H.6 ADDITIONAL RESULTS: MULTIFACTOR DISENTANGLEMENT

In this section, we present more examples for traversing the latent space, separately for the static and dynamic factors. For static factors, we show in Figs. 25 to 36. There, we find different factors of variation such as Male to Female, younger to older, brighter and darker hair color, and more. Each row in the figure is a video, and the different columns represent the traversal in $\alpha$ values (see Eq. 6). In addition, we present full examples of dynamic factor traversal in Figs. 37 to 48, demonstrating various factors of variation. Among the factors are facial expressions, camera angles, head rotations, eyes and mouth control, etc.

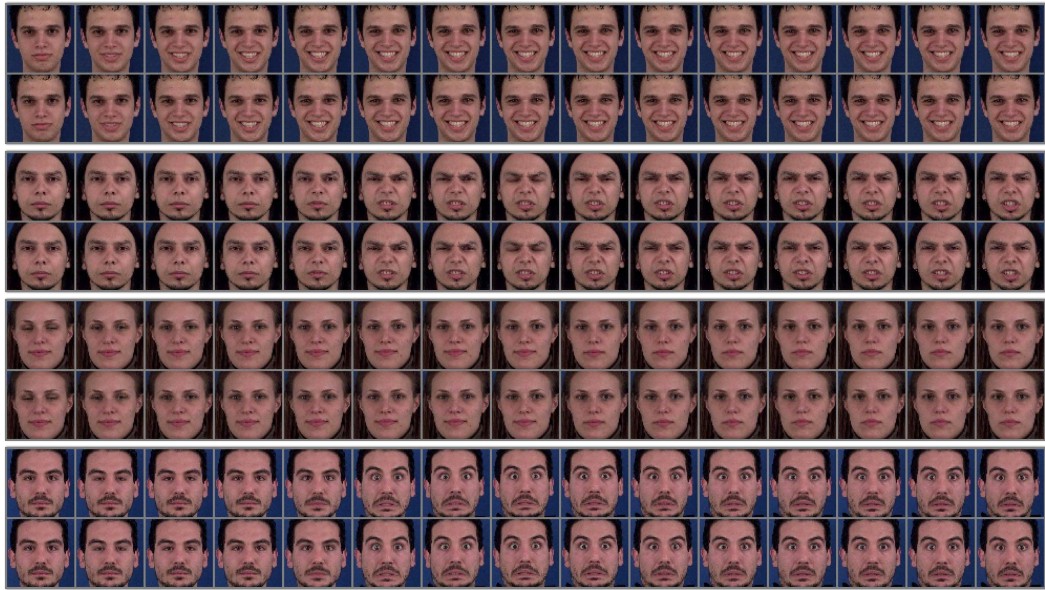

Figure 11: Reconstruction results of MUG. The first row for each pair is the original video and the second row is its reconstruction.

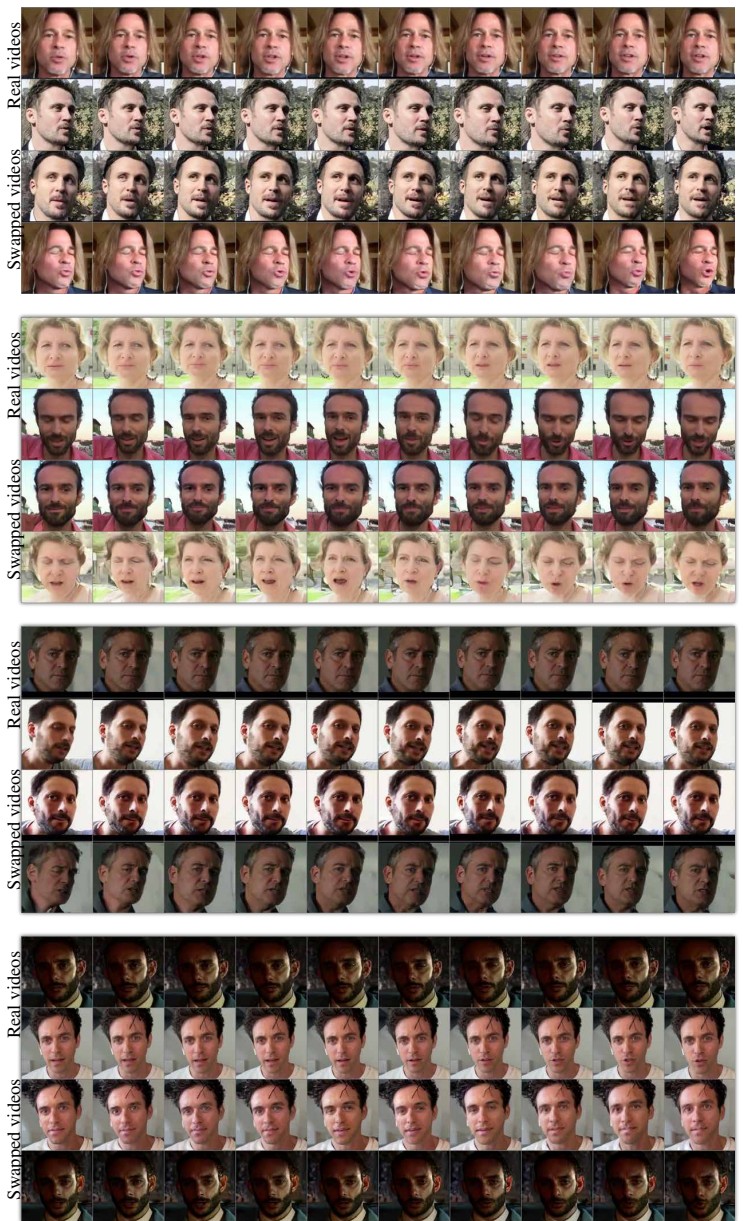

Figure 12: Each panel contains a pair of original videos from CelebV-HQ (Real videos), and a pair of conditional swapping of the dynamic and static factors (Swapped videos).

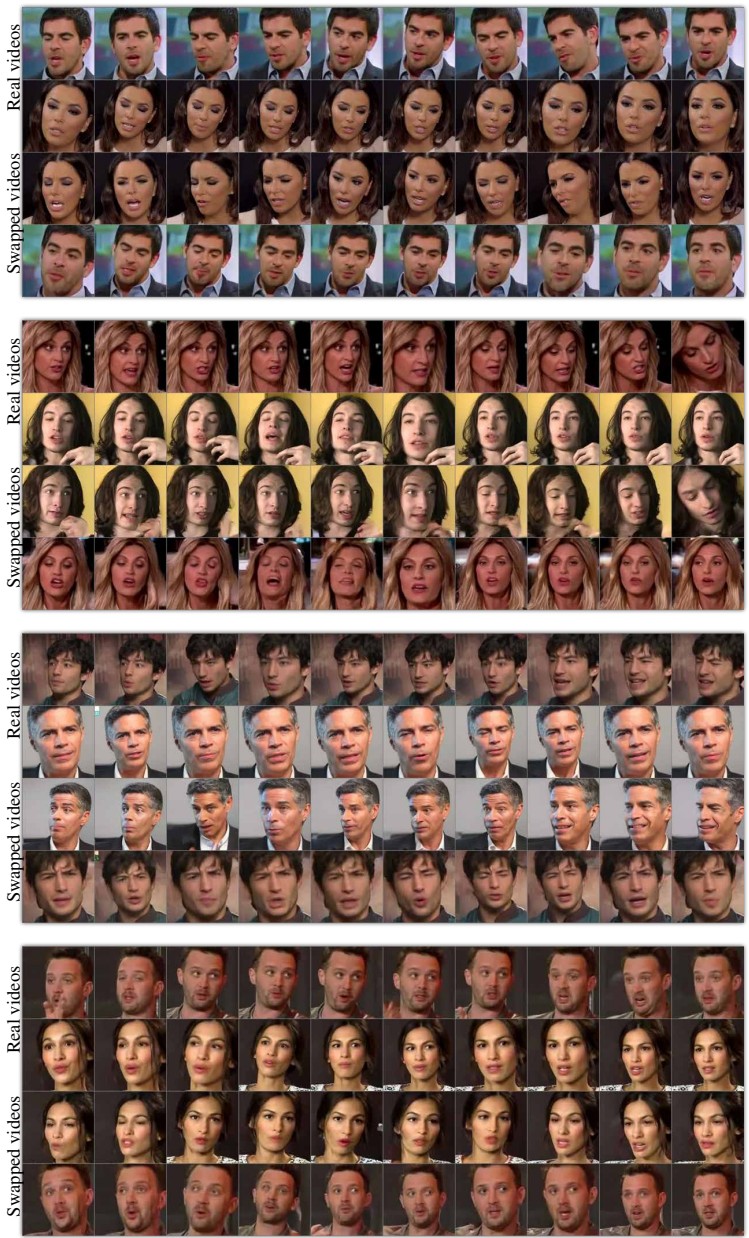

Figure 13: Each panel contains a pair of original videos from VoxCeleb (Real videos), and a pair of conditional swapping of the dynamic and static factors (Swapped videos).

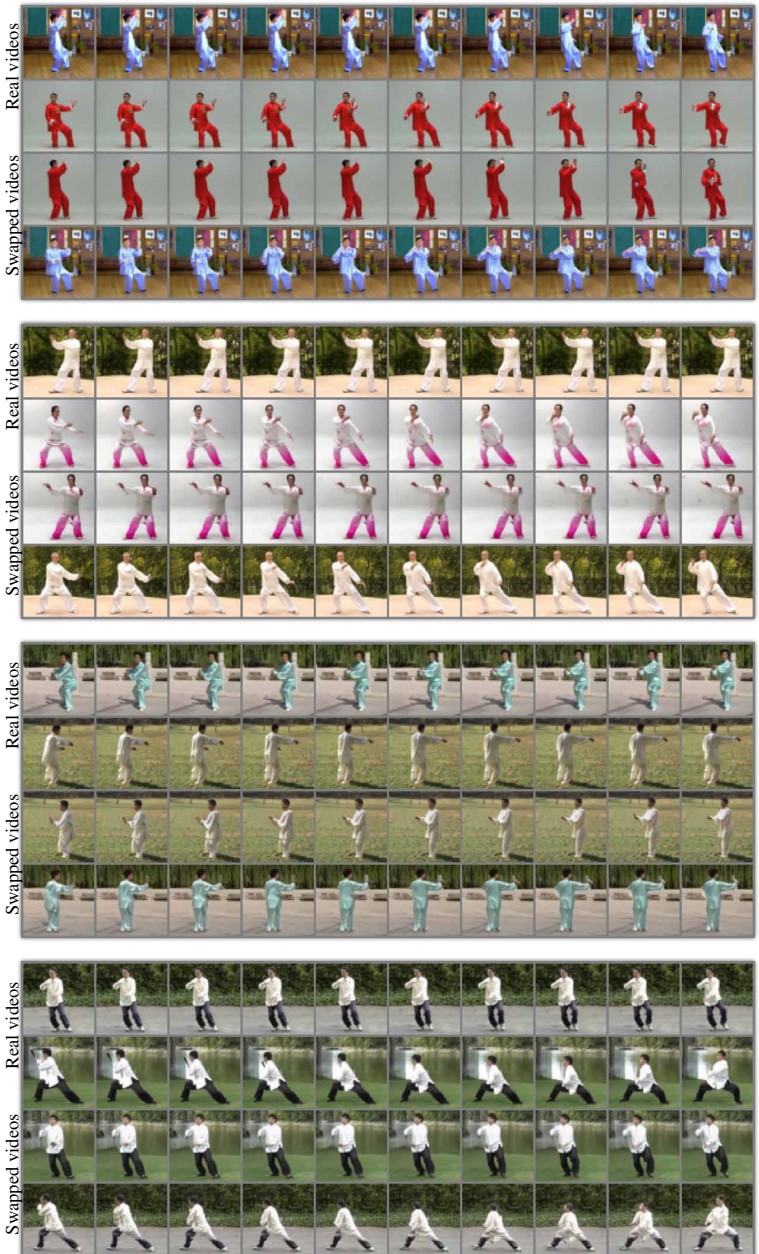

Figure 14: Each panel contains a pair of original videos from TaiChi-HD (Real videos), and a pair of conditional swapping of the dynamic and static factors (Swapped videos).

Figure 15: Each panel contains a pair of original videos from MUG (Real videos), and a pair of conditional swapping of the dynamic and static factors (Swapped videos).

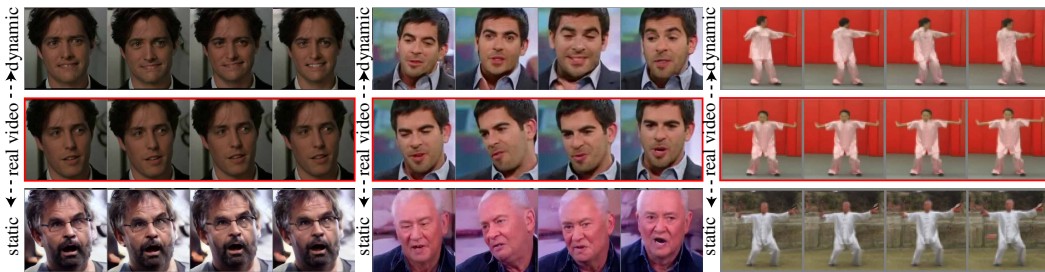

Figure 16: Unconditional dynamic (top) and static (bottom) swap results on CelebV-HQ (left), VoxCeleb (middle), and TaiChi-HD (right).

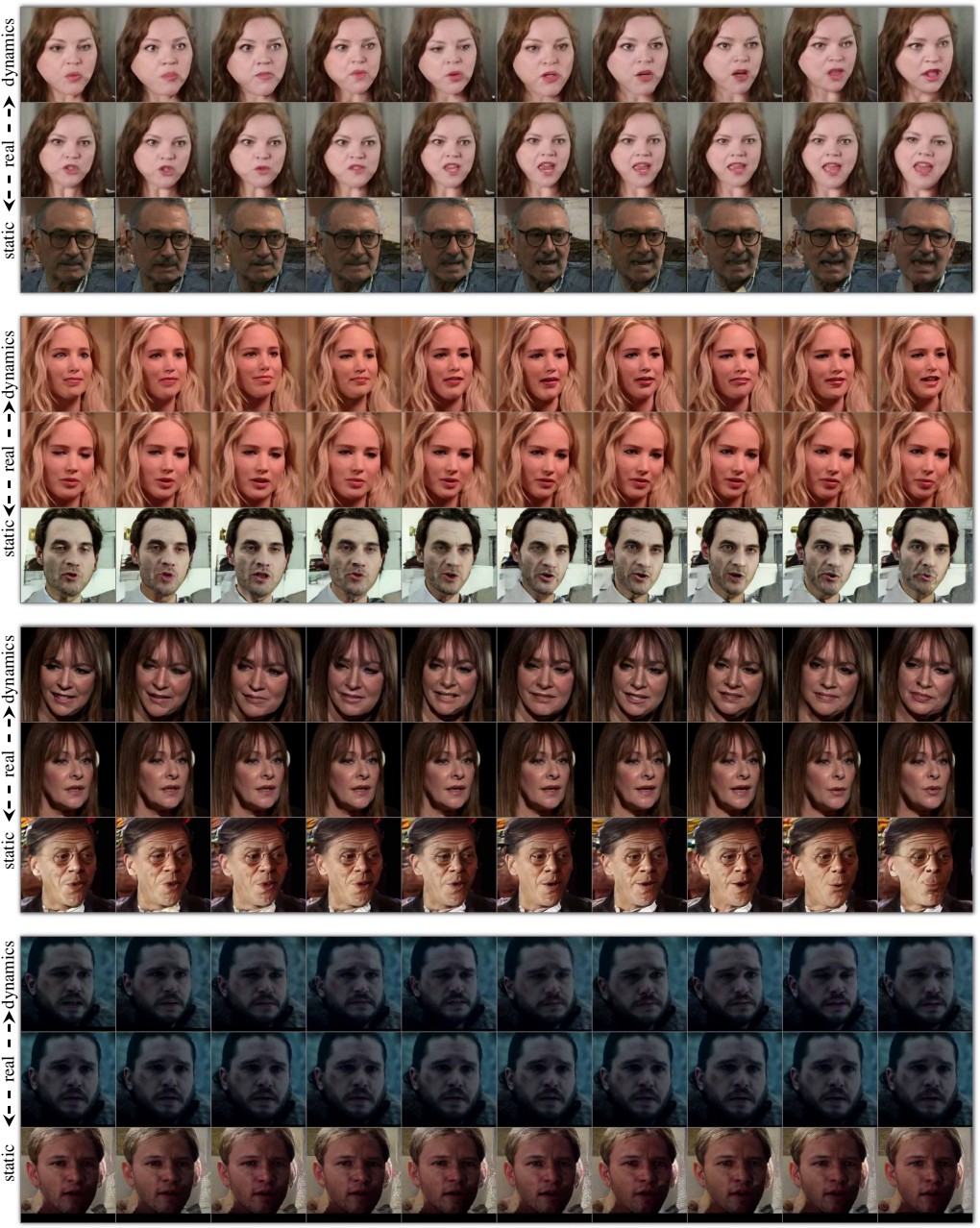

Figure 17: CelebV-HQ unconditional swapping. The middle row represents the original video (real), the row above shows a dynamic swap (dynamics), and the row below shows a static swap (static).

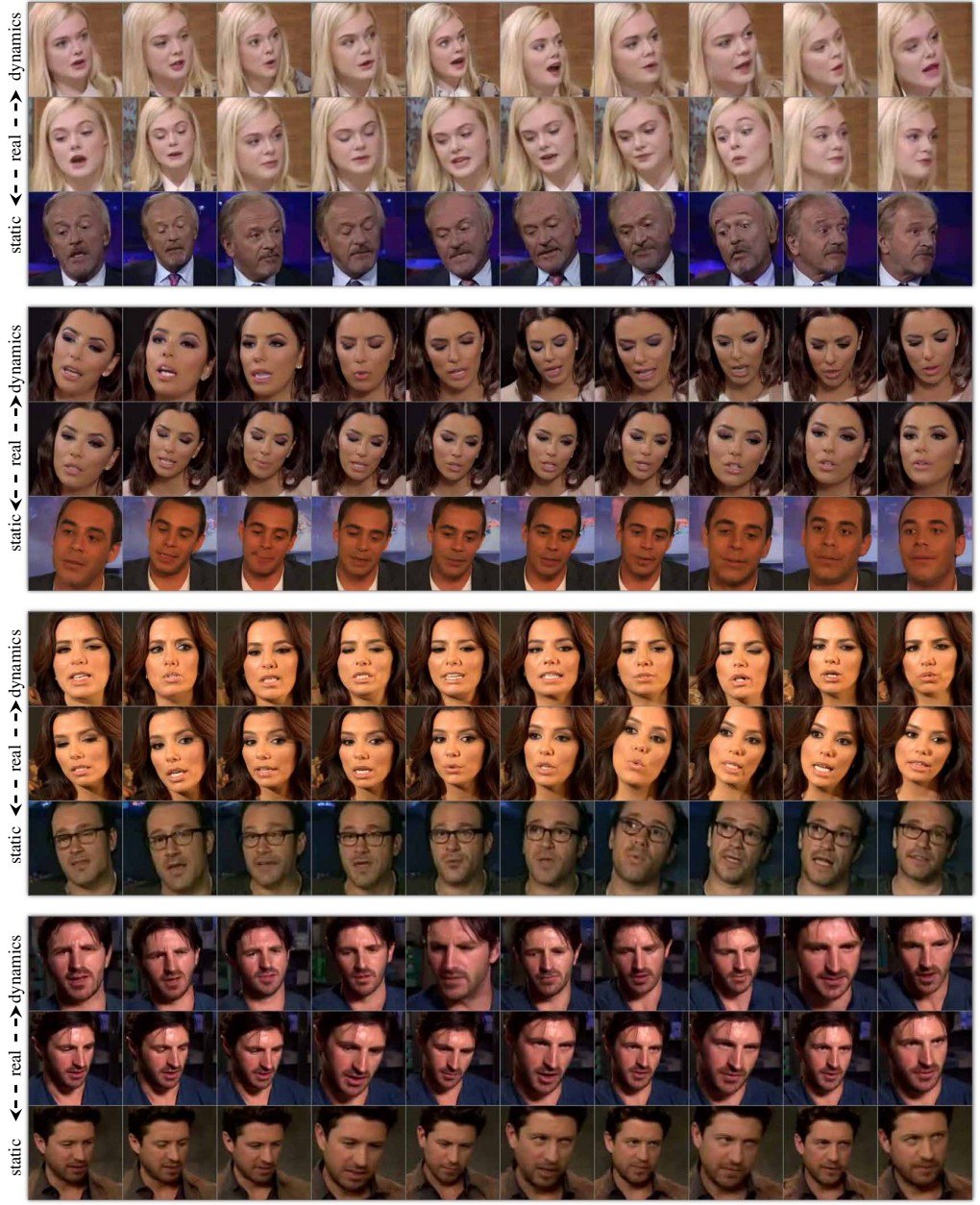

Figure 18: VoxCeleb unconditional swapping. The middle row represents the original video (real), the row above shows a dynamic swap (dynamics), and the row below shows a static swap (static).

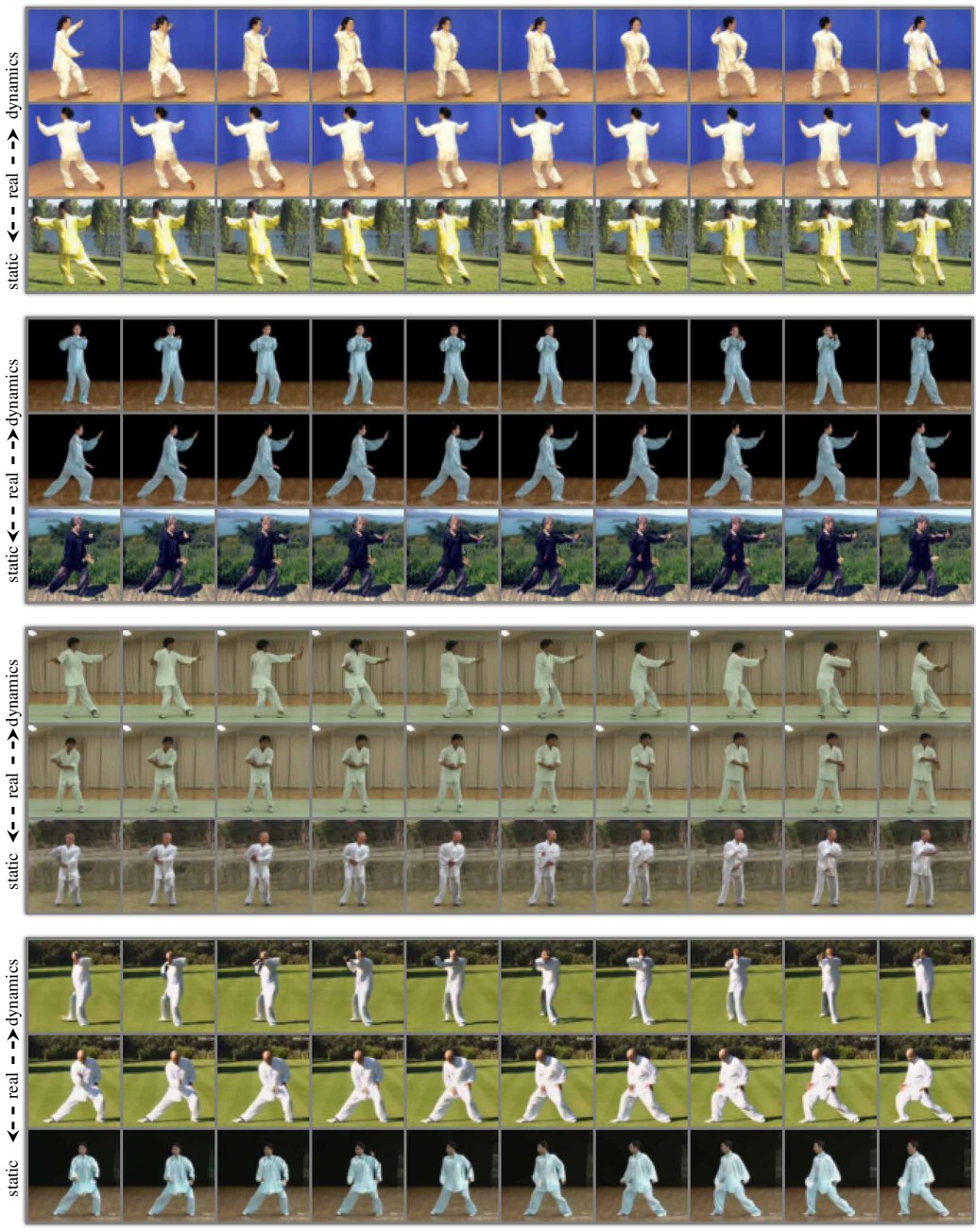

Figure 19: TaiChi-HD unconditional swapping. The middle row represents the original video (real), the row above shows a dynamic swap (dynamics), and the row below shows a static swap (static).

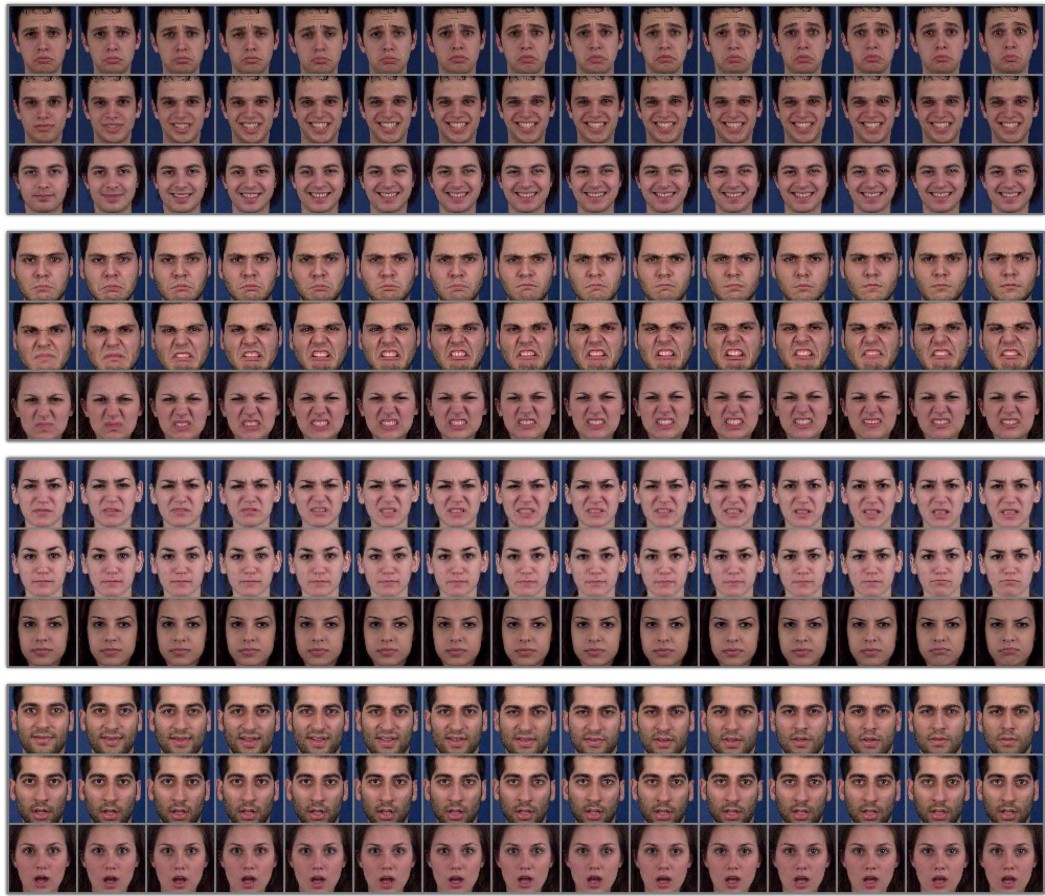

Figure 20: MUG unconditional swapping. The middle row represents the original video (real), the row above shows a dynamic swap (dynamics), and the row below shows a static swap (static).

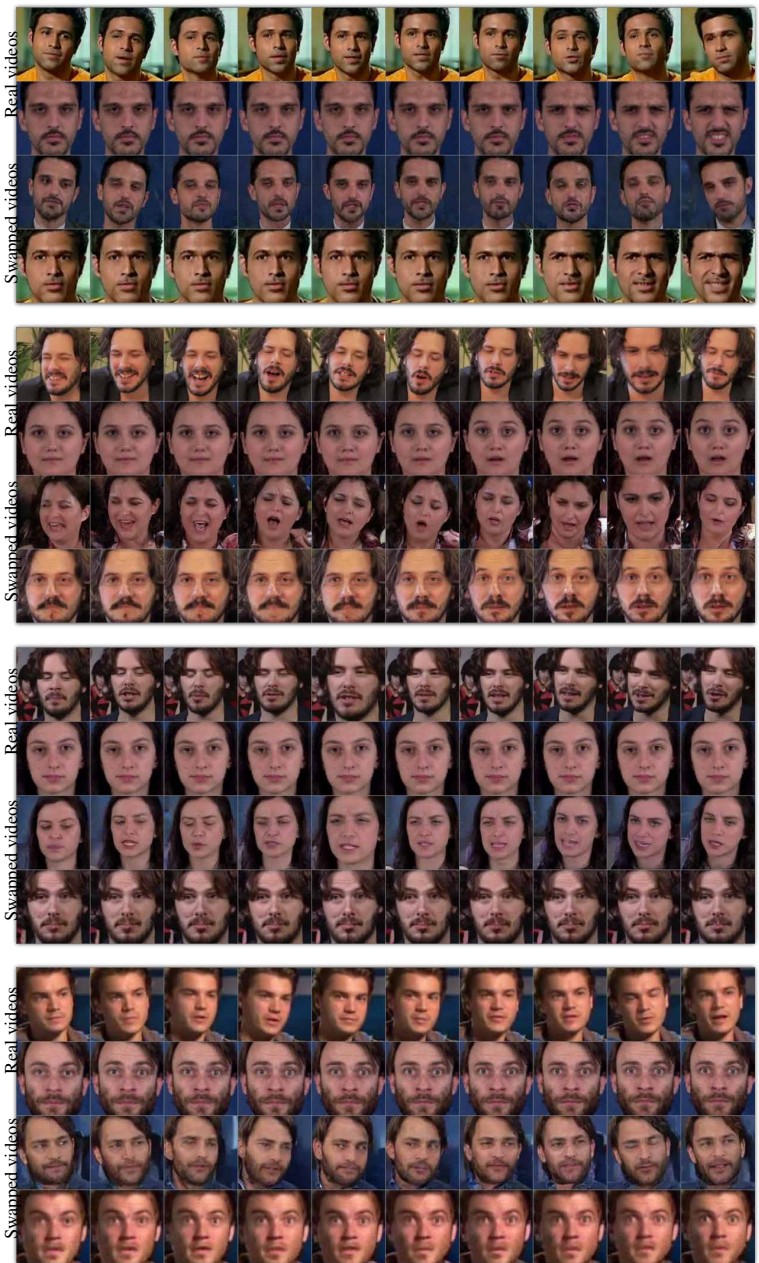

Figure 21: Each panel contains in its first and second rows a pair of real videos from VoxCeleb and MUG, respectively. We perform conditional swapping using a model that was trained on VoxCeleb, but we zero-shot swap the dynamic and static factors of a MUG example (Swapped videos).

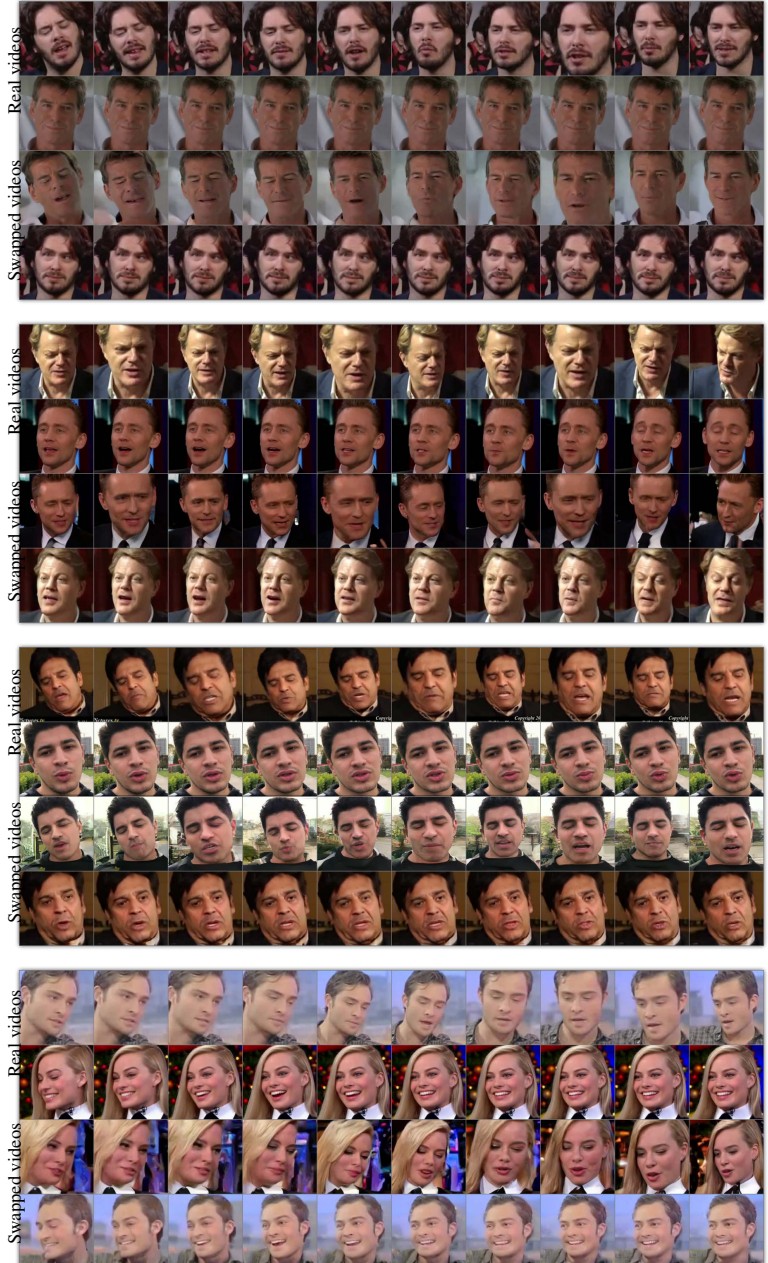

Figure 22: Each panel contains in its first and second rows a pair of real videos from VoxCeleb and CelebV-HQ. We perform conditional swapping using a model that was trained on VoxCeleb, but we zero-shot swap the dynamic and static factors of a CelebV-HQ example (Swapped videos).

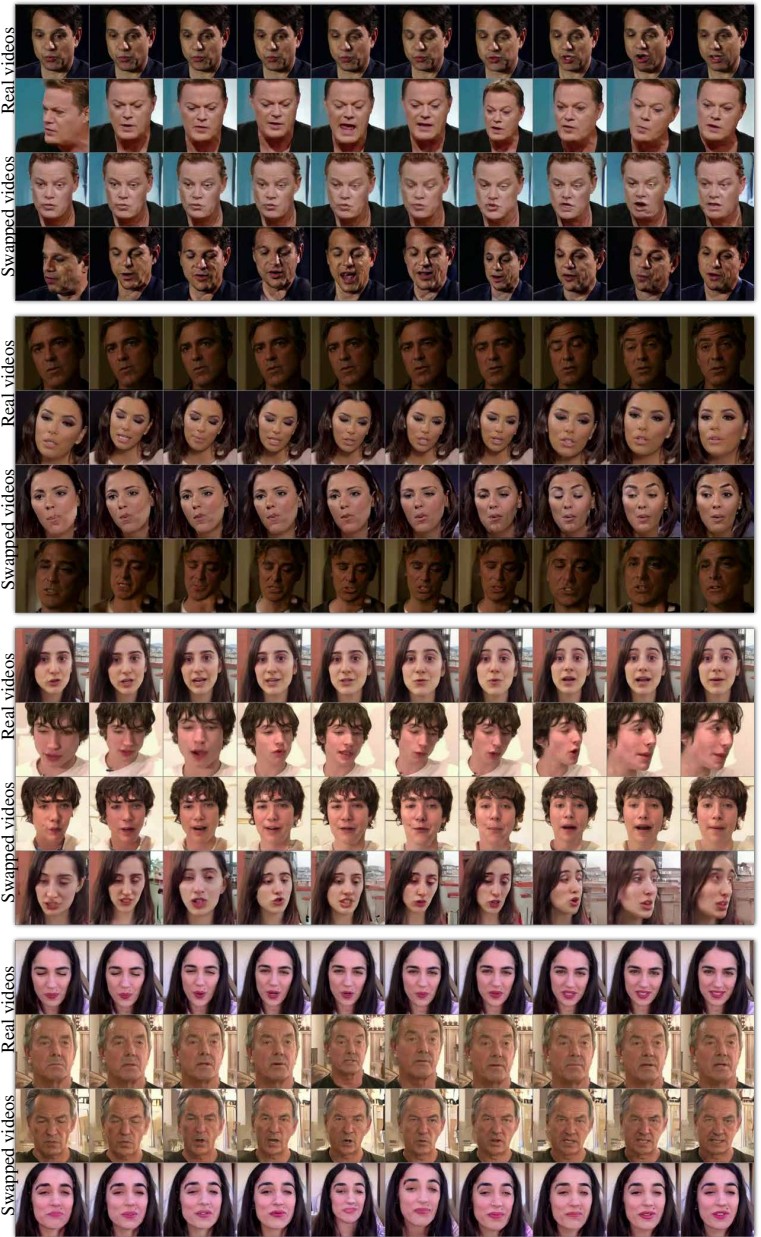

Figure 23: Each panel contains in its first and second rows a pair of real videos from CelebV-HQ and VoxCeleb. We perform conditional swapping using a model that was trained on CelebV-HQ, but we zero-shot swap the dynamic and static factors of a VoxCeleb example (Swapped videos).

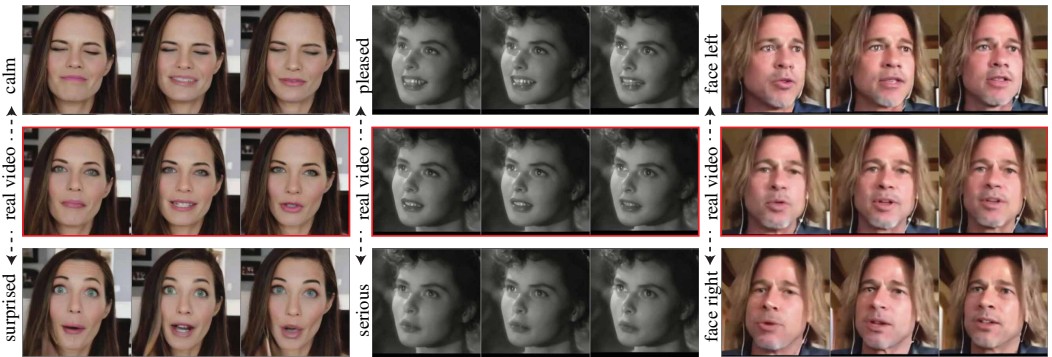

Figure 24: Traversing the latent space of DiffSDA via PCA reveals multiple dynamic variations on CelebV-HQ, including surprised and serious expressions, and different head orientations.

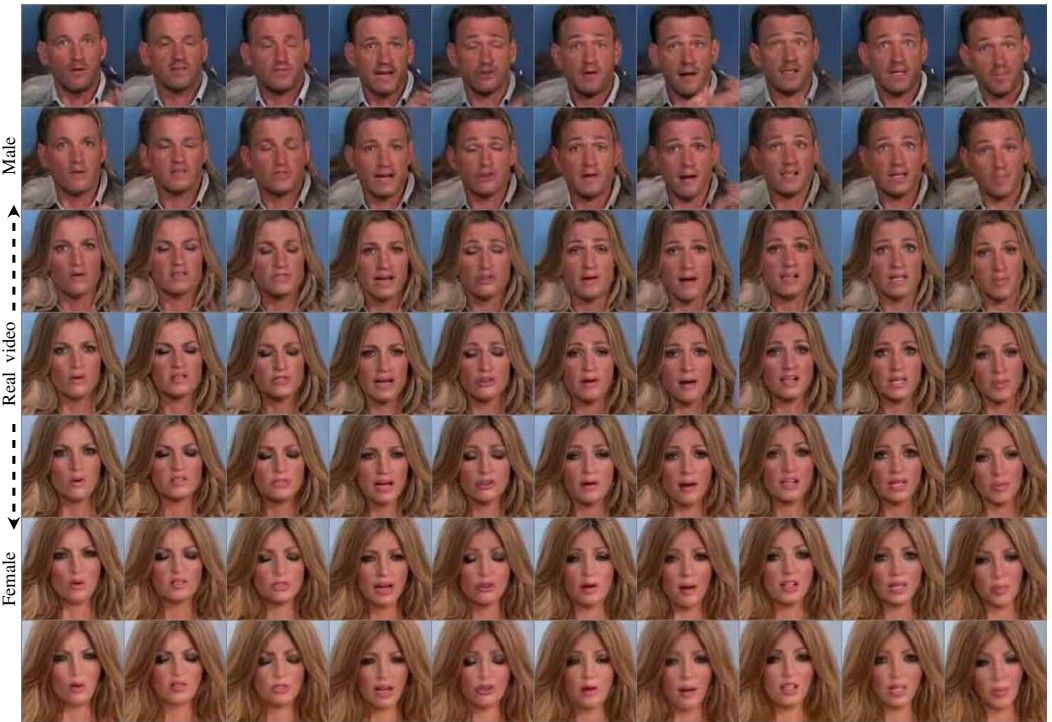

Figure 25: Traversing between Male appearances and Female appearances.

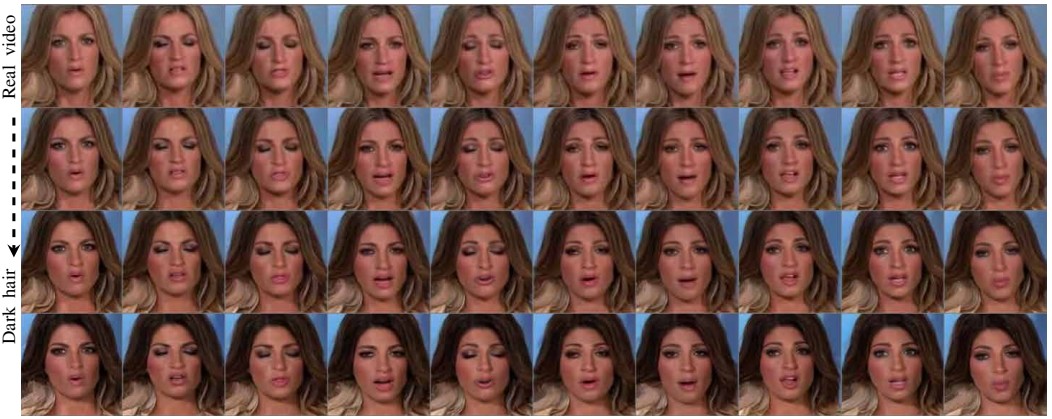

Figure 26: Traversing over a darker hair factor.

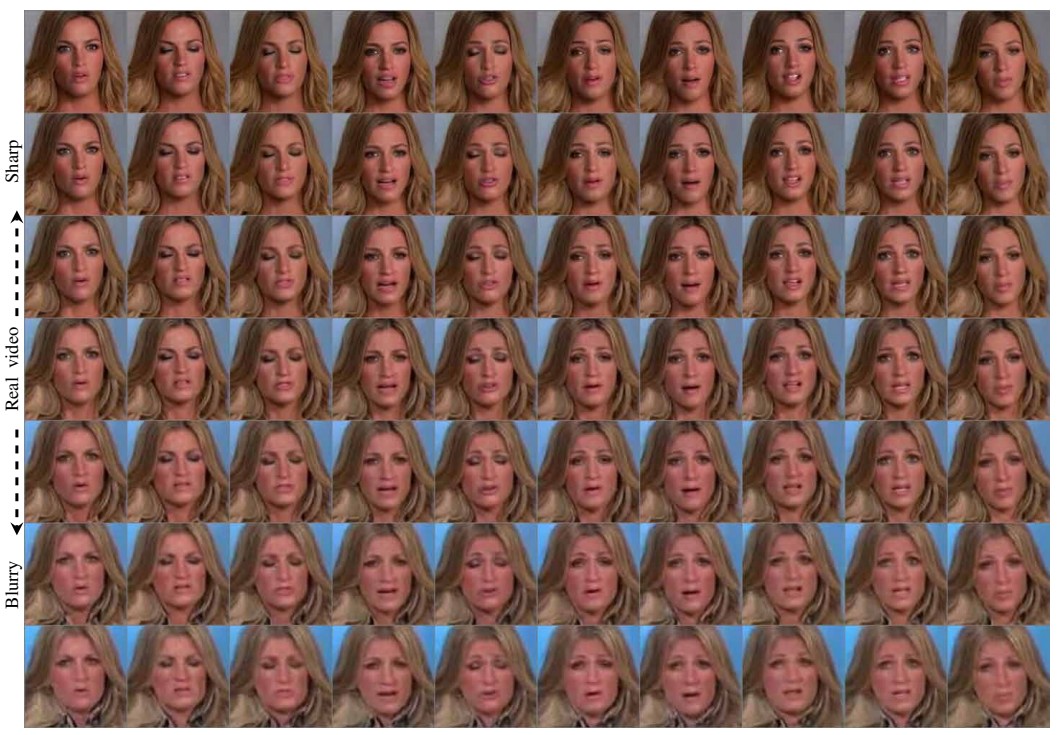

Figure 27: Traversing between sharper and blurry videos.

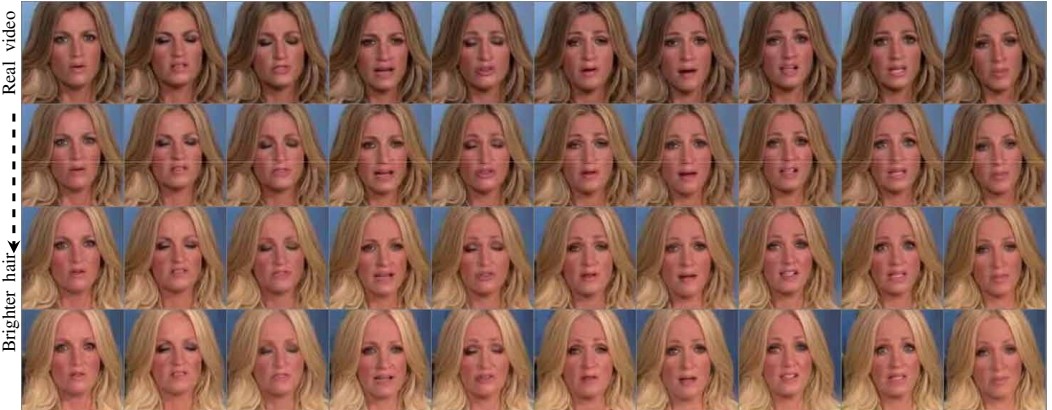

Figure 28: Traversing over a brighter hair factor.

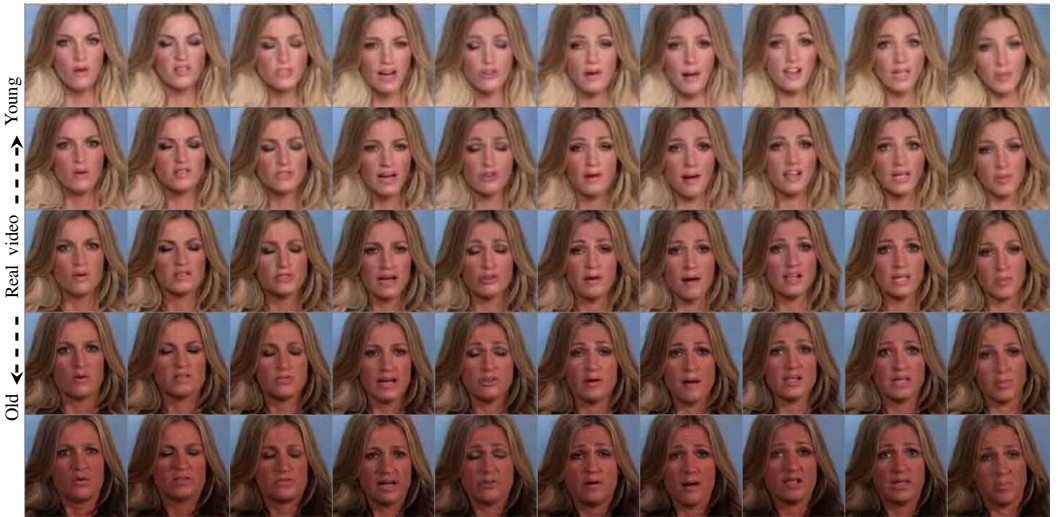

Figure 29: Traversing between younger and older appearances.

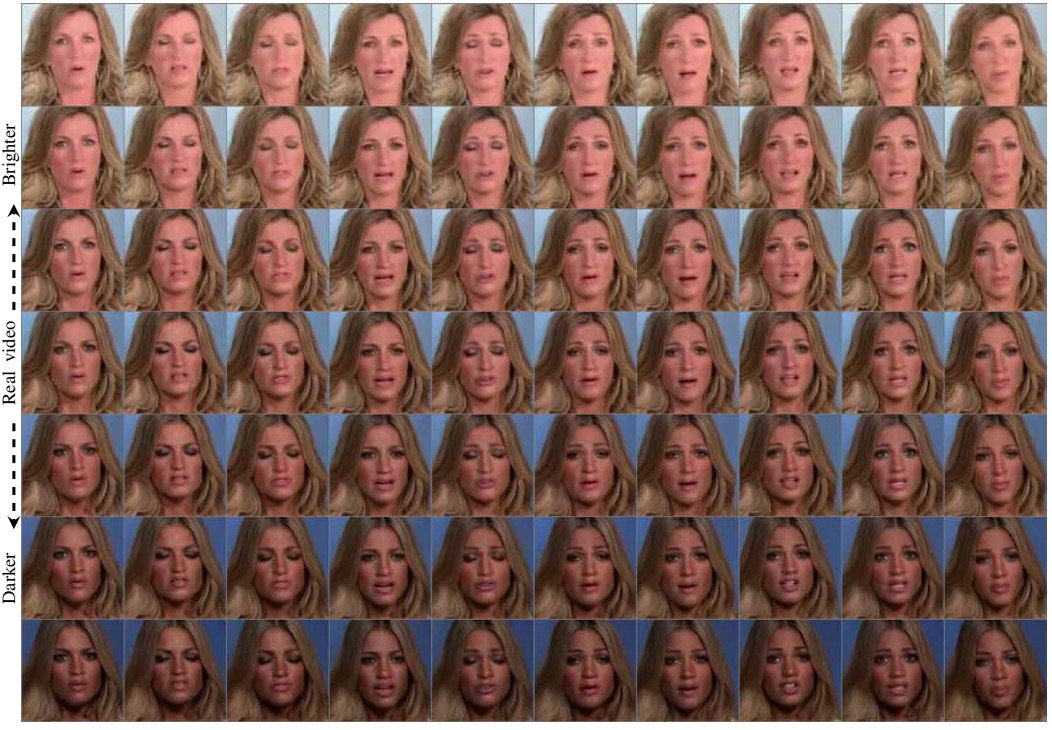

Figure 30: Traversing over skin color variations.

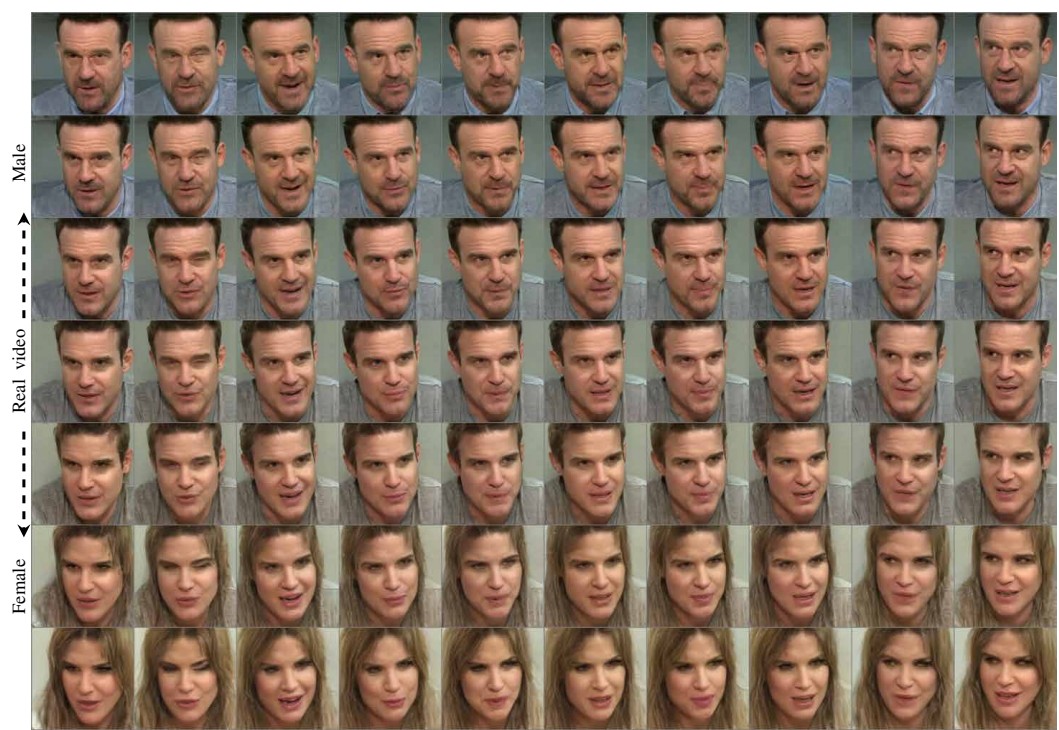

Figure 31: Traversing between Male appearances and Female appearances.

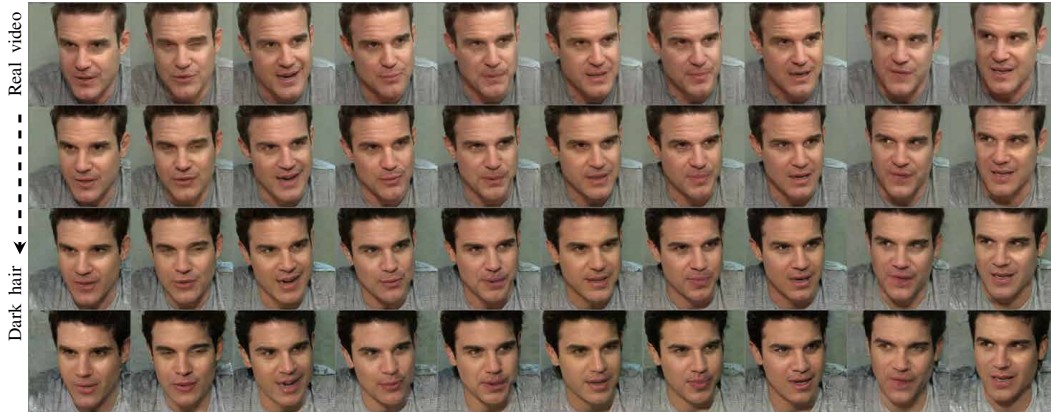

Figure 32: Traversing over a darker hair factor.

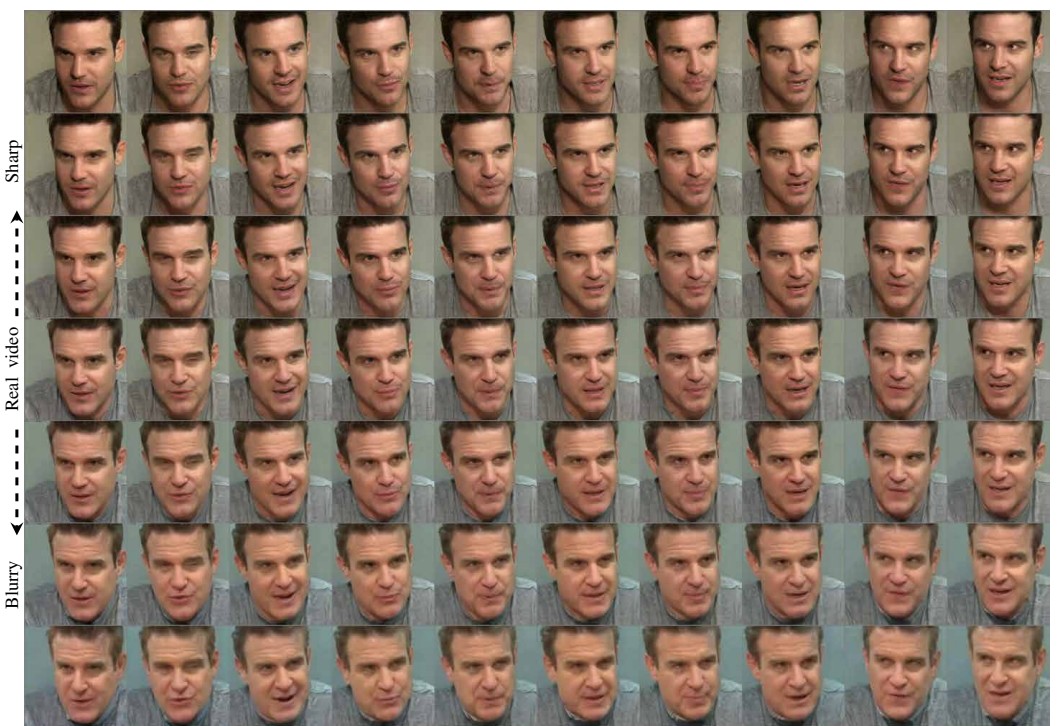

Figure 33: Traversing between sharper and blurry videos.

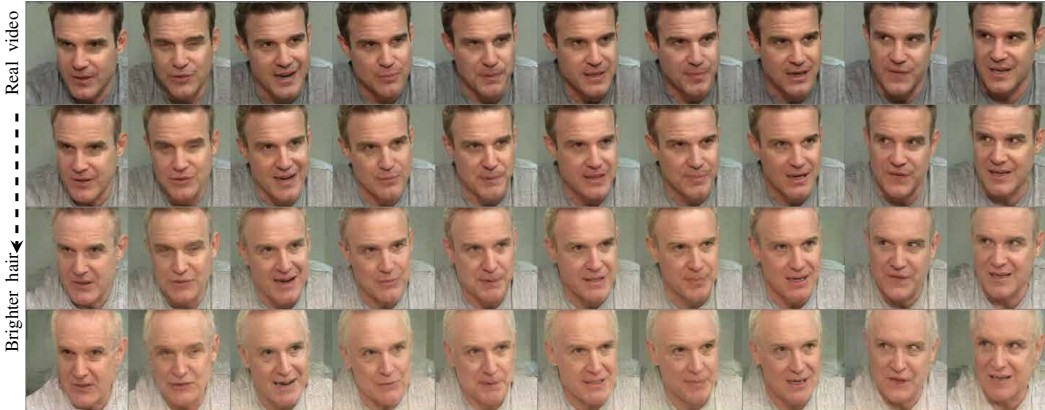

Figure 34: Traversing over a brighter hair factor.

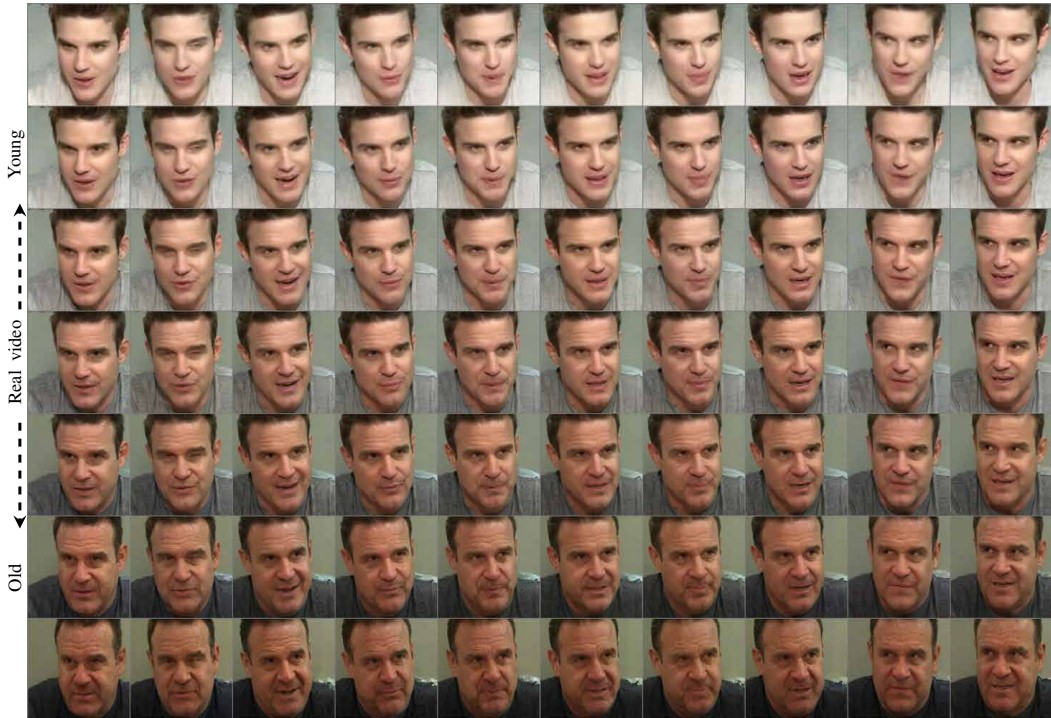

Figure 35: Traversing between younger and older appearances.

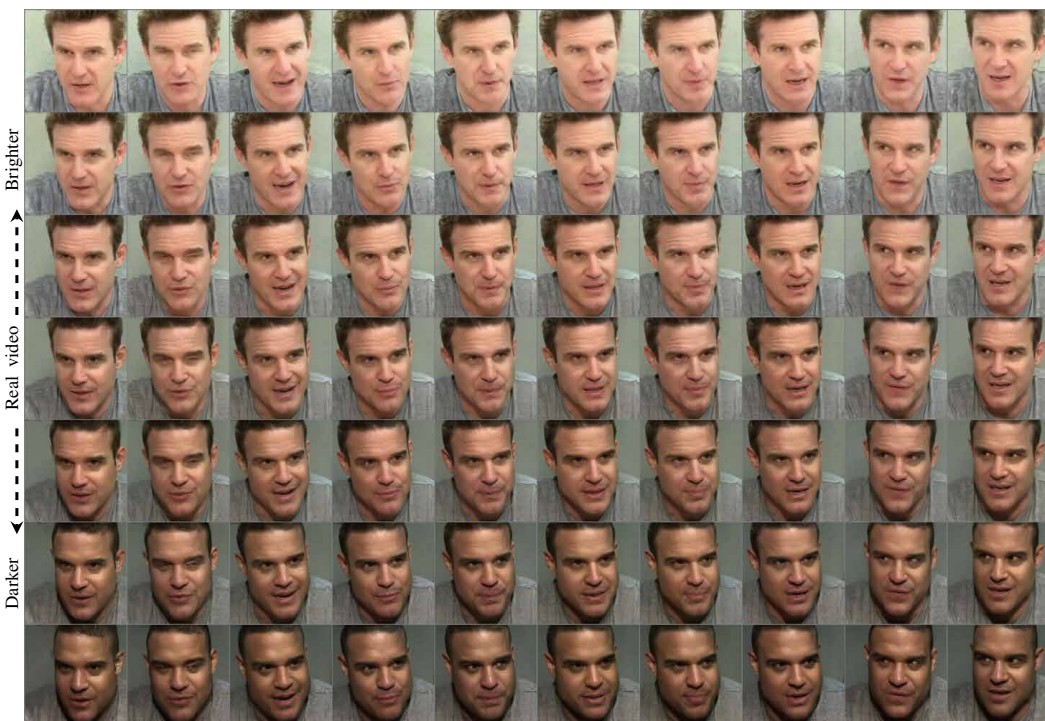

Figure 36: Traversing over skin color variations.

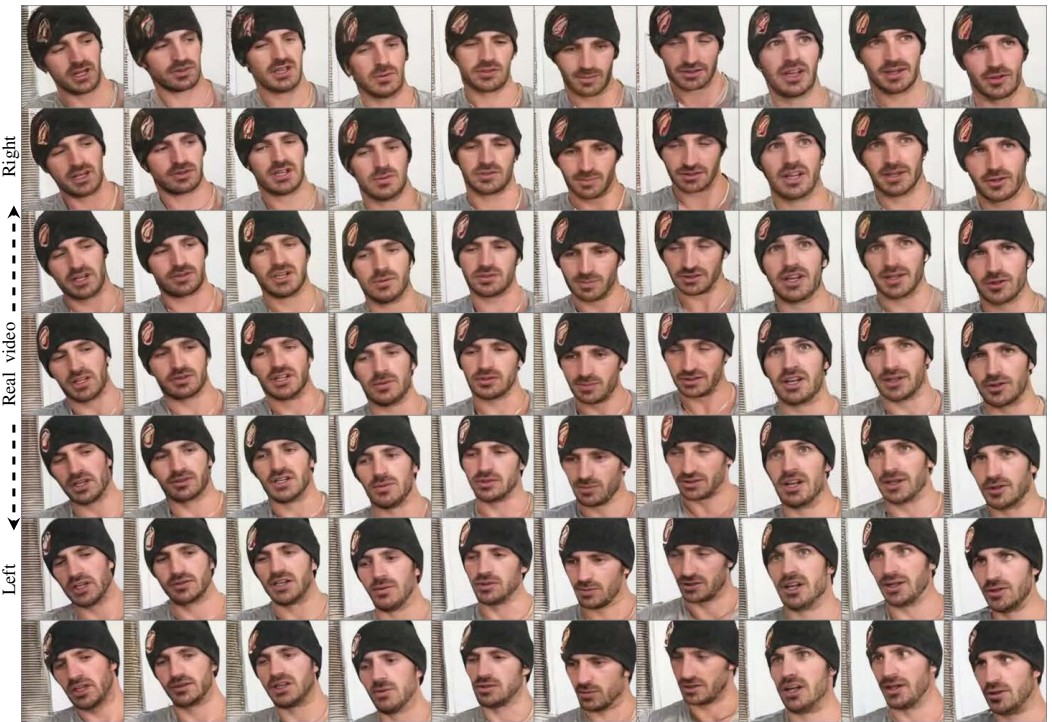

Figure 37: Traversing a head rotation factor.

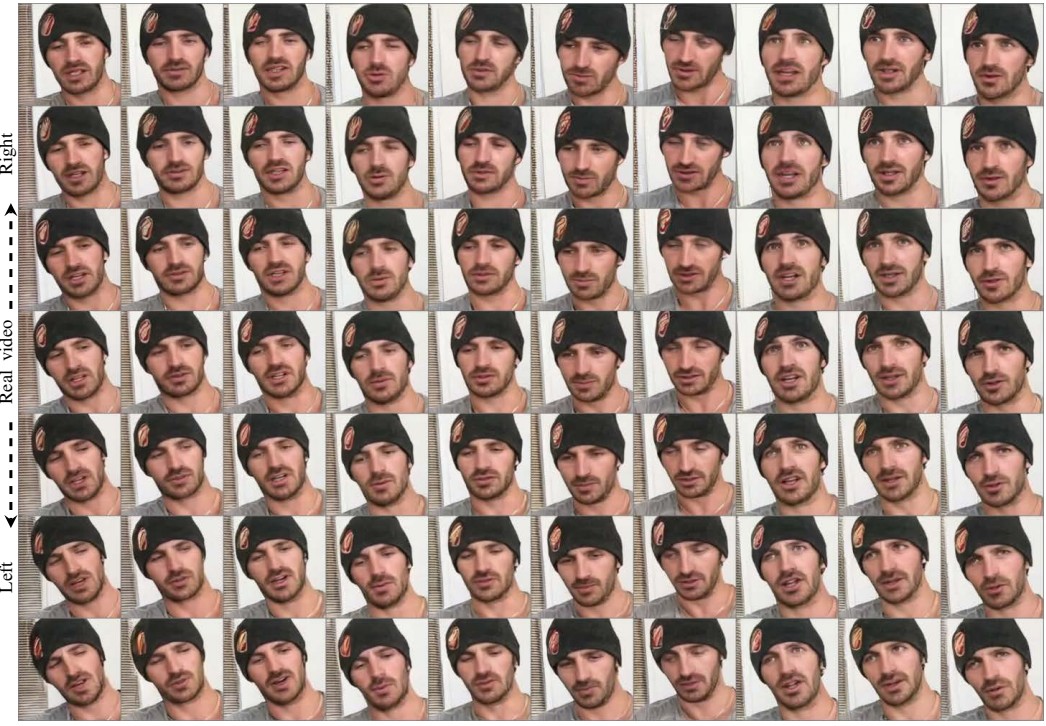

Figure 38: Traversing over head angles.

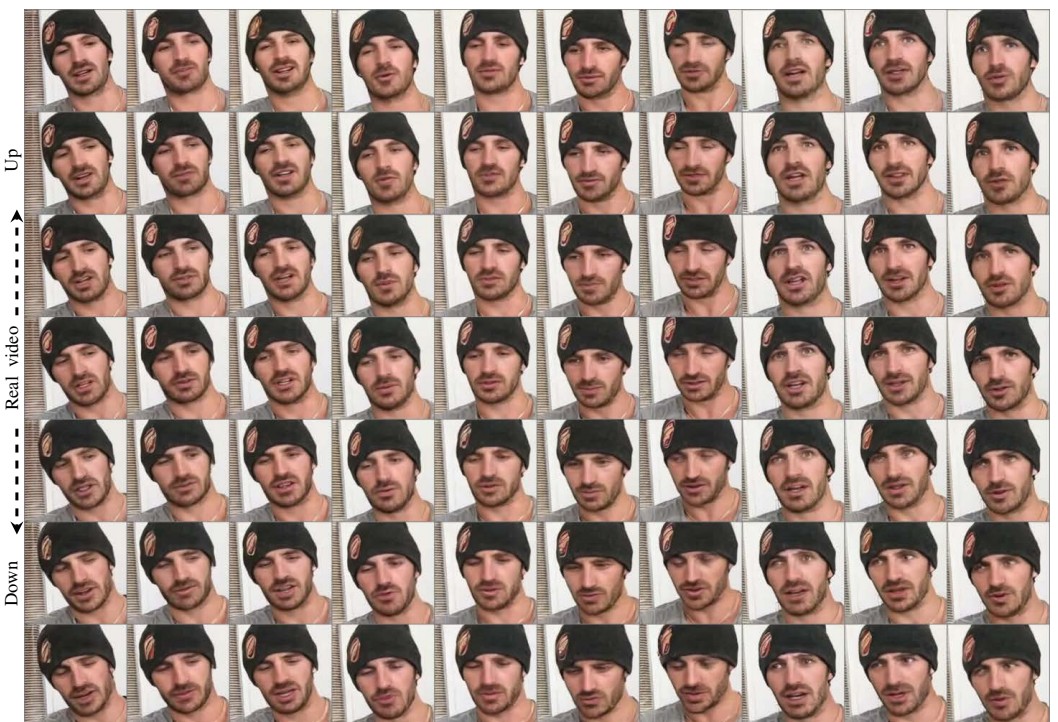

Figure 39: Traversing over up and down rotations.

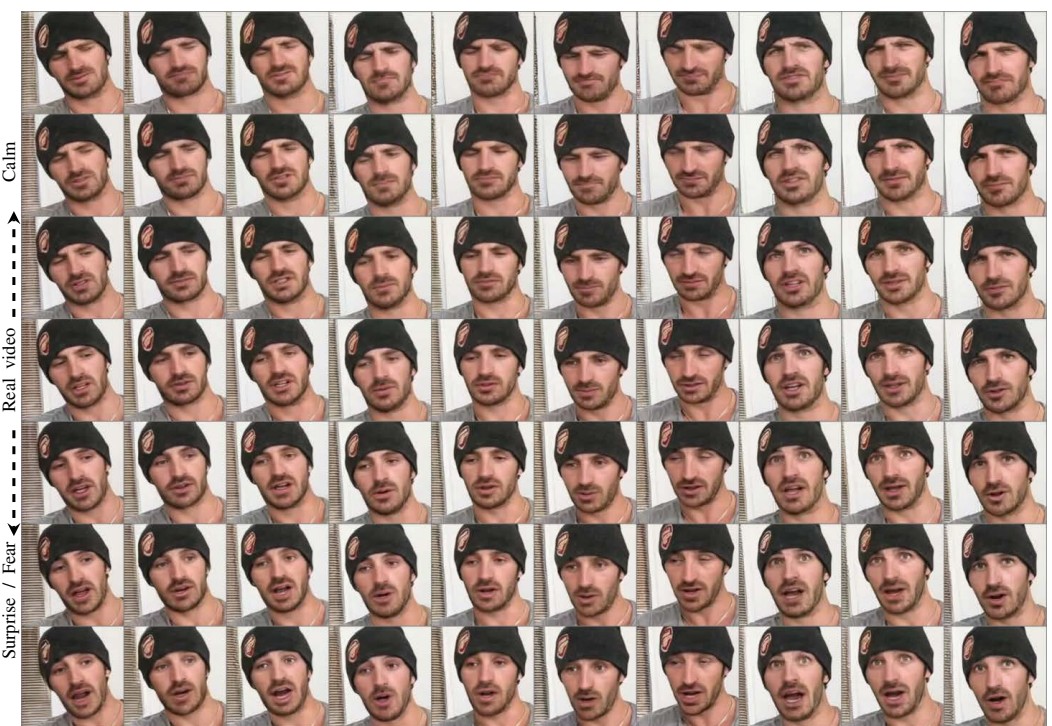

Figure 40: Traversing over facial expressions.

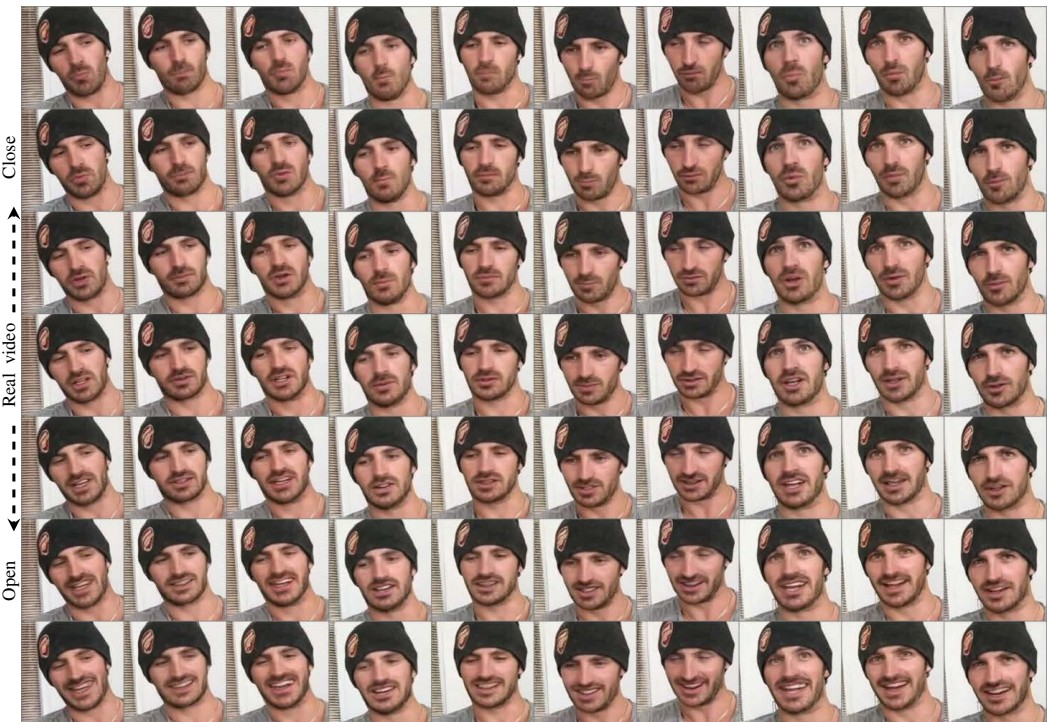

Figure 41: Traversing over mouth openness factor.

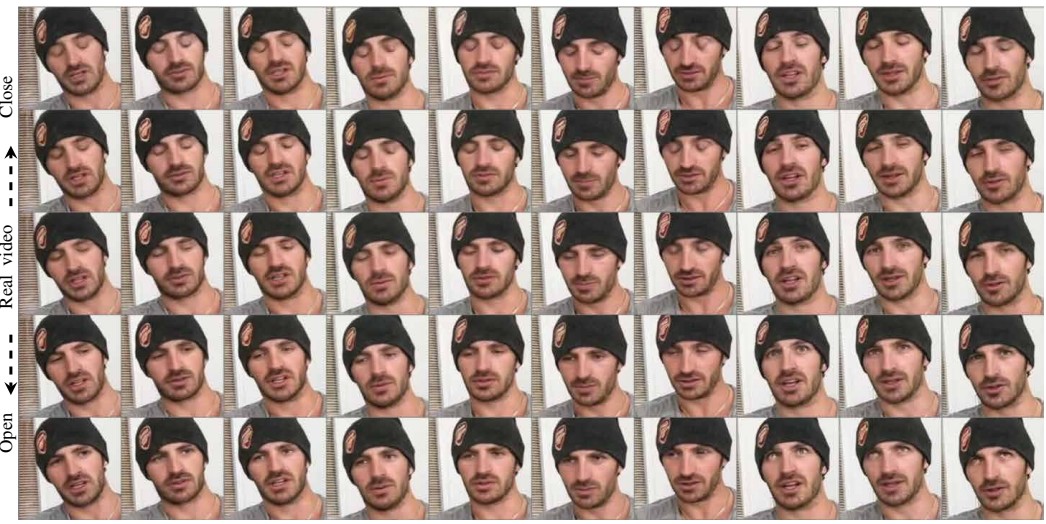

Figure 42: Traversing over eyes openness factor.

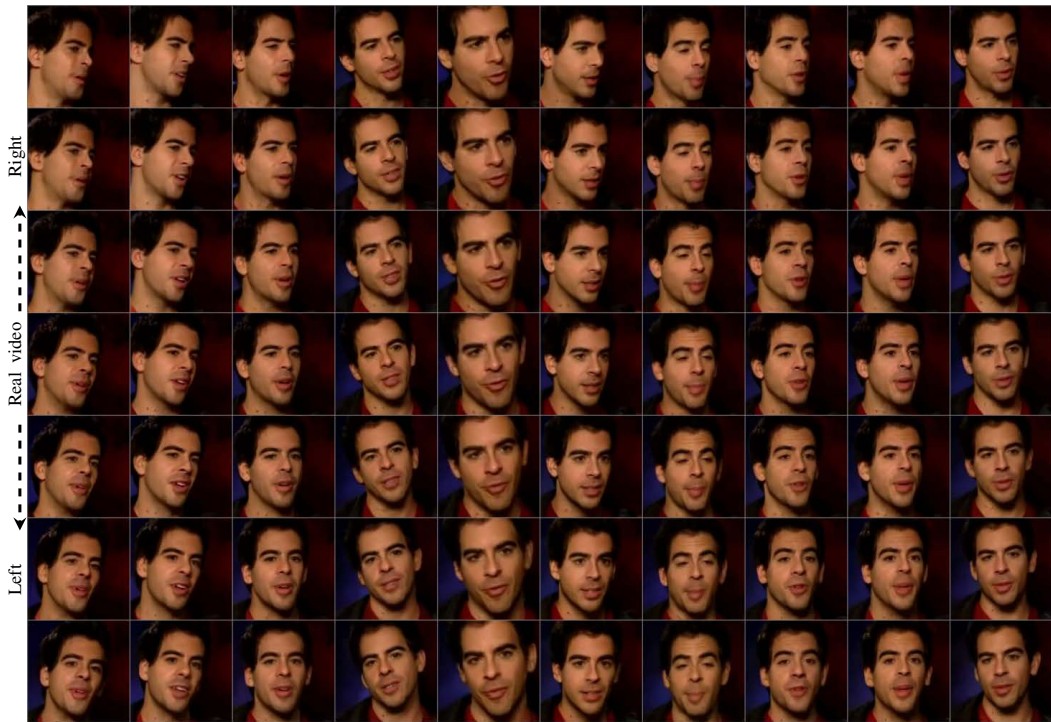

Figure 43: Traversing over a head rotation factor.

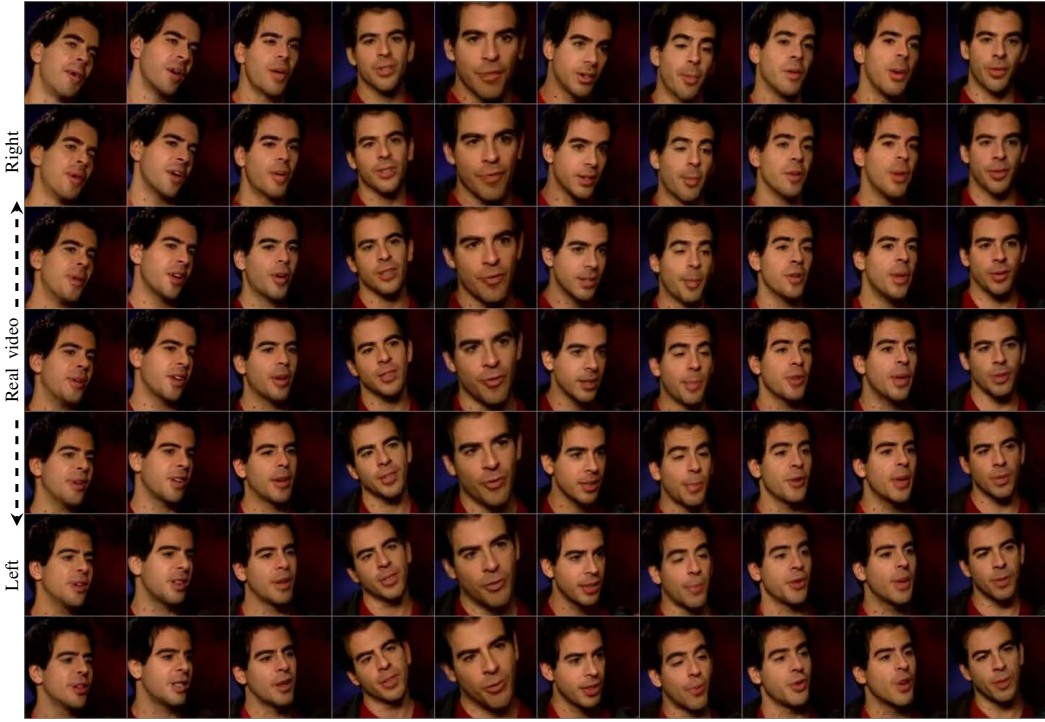

Figure 44: Traversing over various head angles.

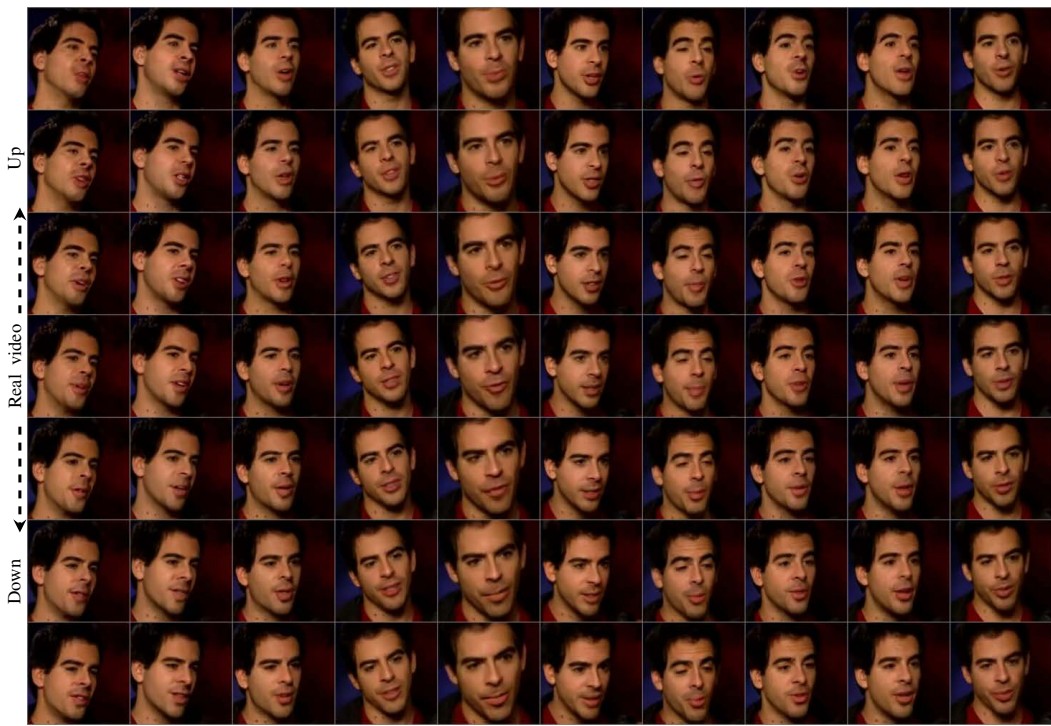

Figure 45: Traversing over up and down head rotations.

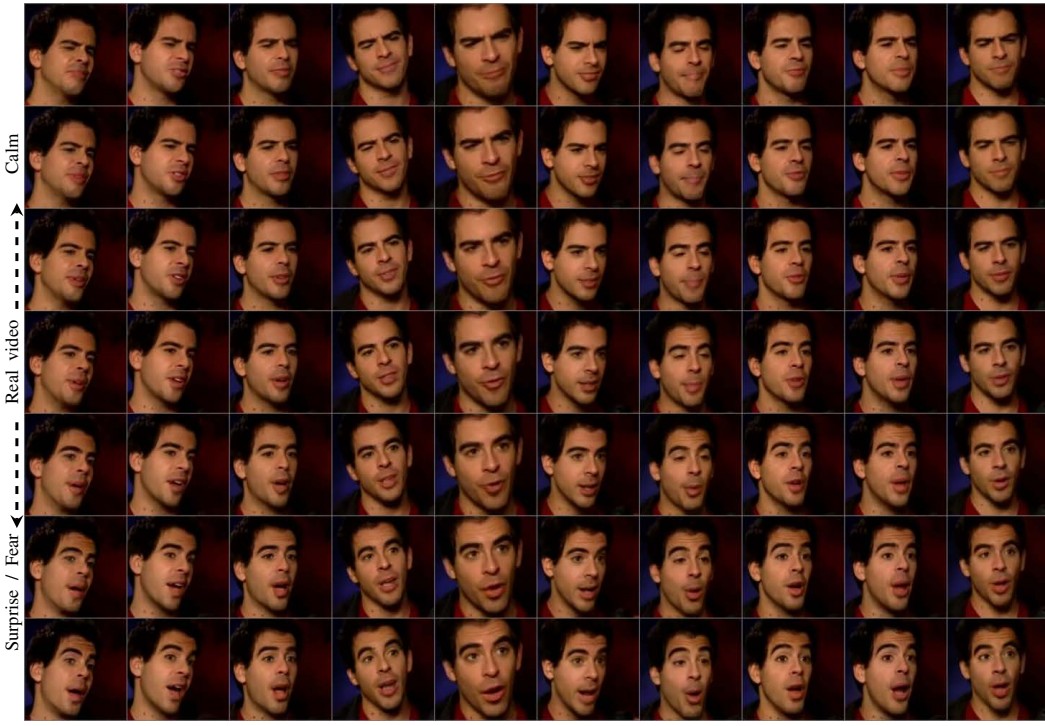

Figure 46: Traversing over facial expressions.

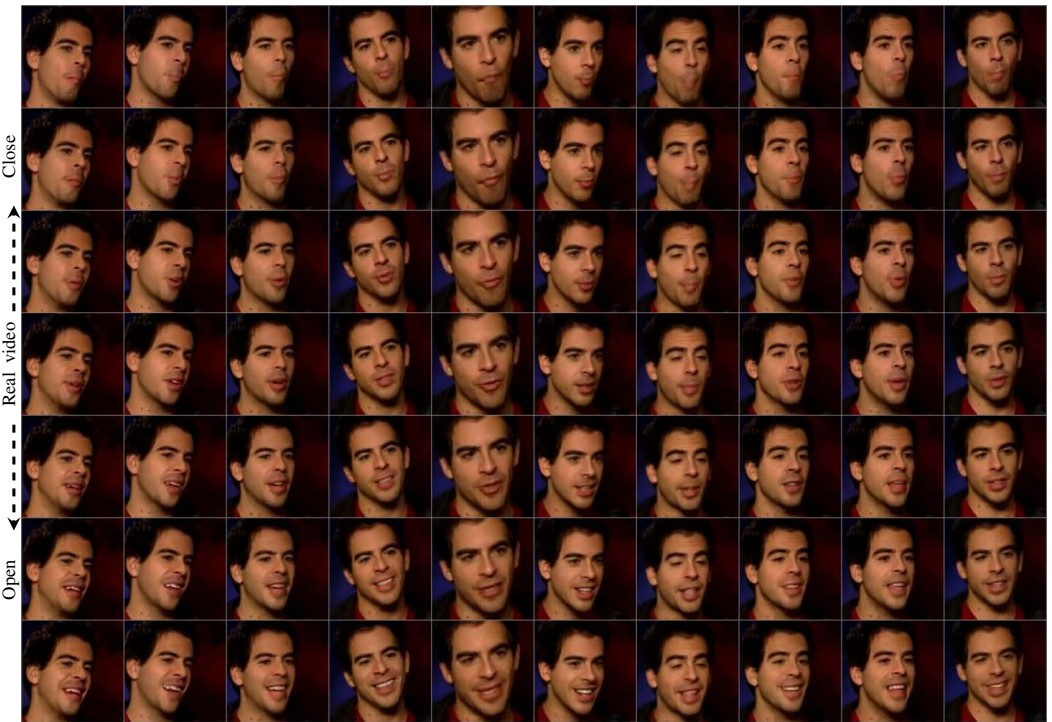

Figure 47: Traversing over mouth openness factor.

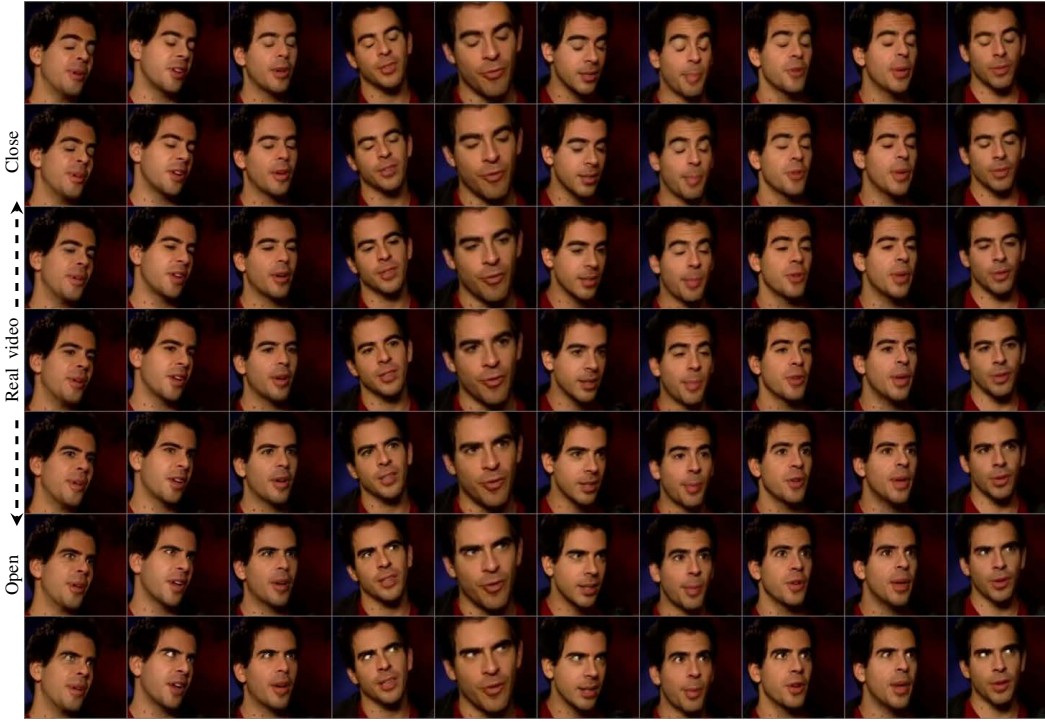

Figure 48: Traversing over eyes openness factor.

