# OpenReview forum: "DiffSDA: Unsupervised Diffusion Sequential Disentanglement Across Modalities"
_ICLR.cc/2026/Conference — ICLR 2026 Poster_

### Official Review · Reviewer_nupF · 2025-10-21

**Soundness:** 3
**Presentation:** 3
**Contribution:** 3
**Rating:** 6
**Confidence:** 4

**Summary:**

This work presents a new systematic approach to latent factor disentanglement of sequential data. It is based on a combination of several models 1) to extract static and dynamic components of latent factors, 2) to stochastically encode sequential data, 3) to conditionally decode sequential data based on latent factors. Additional components to deal with high-dimensional data (such as video) might be required as well, such as VQ-VAE.

The proposed method is described using the language of probabilities, akin to what we are used to when dealing with diffusion models.

The authors perform a thorough and extensive validation campaign, using an appropriate experimental protocol. They define both qualitative and quantitative metrics for the numerous tasks they address and show that the proposed method is superior to the state of the art.

**Strengths:**

* The experimental protocol used to assess DiffSDA and compare it to alternatives is extensive and globally well done. The authors have produced convincing qualitative and quantitative results indicating that the proposed method outperforms current SOTA approaches.

* Measuring quantitatively the effectiveness of disentanglement is a hard problem. The authors worked really hard on this issue and 1) tested an external judge (a classifier), and 2) proposed an unsupervised swapping metric to sidestep the limitations of 1).

* This work is self contained: as a reader, I greatly appreciate the quality and depth of the material in the appendix, especially the details on the metrics, as well as the additional results and ablations.

**Weaknesses:**

* It's minor, but I would argue that the narrative used in the introduction about the requirement for "labels" for generative models is a bit misleading. Text-to-image generation indeed requires pairs of image and prompts, but unconditional models have flourished before those work, and they do not require labels. In general, I find the problem of learning disentangled representation sufficiently recognized as an important issue to address in the community, that does not need additional motivations.

* I am a bit worried about the claim of "simplicity" of the proposed approach. I view the proposed method as a complex system, involving several models (some of which pretrained, others that require separate training) combined in a coherent design, which is far from being simple. Also, the lack of apparent hyper-parameters (a critic the authors advance for other methods on disentanglement from the state of the art), is in my humble opinion, another illustration of downplaying the complications of the proposed system. In practice, diffusion models are not so easy to train, as demonstrated by the choice of the authors to rely on an enormous amount of work that has been done to come up with "recipies" to elucidate the design space of diffusion models.

**Questions:**

* From the introduction, the requirement for "labeled" data stems from, e.g., text-to-image generation. Original work from Ho et al., and from Song et al., are unconditional models which do not require labeled data.

* Check your claims: when the authors say they propose a novel probabilistic approach, they are actually using standard practice in diffusion modeling. The main difference is that they combine two diffusion processes, one for learning (and generating) disentangled latents for static and dynamic components, and one for learning (and generating) the sequential data distribution, conditioned on the disentangled latents. Both models are learned concurrently using a single loss. While my remark does not want to diminish the technical contribution of the proposed method, I think that the claim for novelty is too strong.

* As stated in lines 212-213, eq. (3) does not require training for the latents $(s_0, d_0^{1:V})$ generative model, thus this optimization is separate. Then, the loss in eq. (3) is not the only loss you consider in your overall training of DiffSDA, right? Then, I would like to ask the authors to clarify why in lines 221-222 they claim that only a single loss term is required.

* In line 227, the last hidden representation $h^V$ is fed to a linear layer to produce $s_0$, but this is not reflected in Figure 1. This is just nitpicking, but for clarity, the figure could be easily amended to be coherent with the textual description of the architecture.

* In lines 265-269, the authors mention that for high-dimensional data such as video, they abuse the notation (for ease of reading), and use symbols that denote ambient space, to indicate vector-quantized latent variables. This is clear. The question is: when considering other modalities, such as audio or general timeseries, there is no need for the VQ-VAE embeddings right?

---

> ### Author Response · Authors · 2025-11-20
> **Part 1**
>
> ***W1: It's minor, but I would argue that the narrative used in the introduction about the requirement for "labels" for generative models is a bit misleading. Text-to-image generation indeed requires pairs of image and prompts, but unconditional models have flourished before those work, and they do not require labels. In general, I find the problem of learning disentangled representation sufficiently recognized as an important issue to address in the community, that does not need additional motivations.***
>
> We thank the reviewer for the clarification. We acknowledge that the phrasing in the introduction may come across as ambiguous: our intention was not to imply that generative models require labels, but rather to motivate why learning meaningful, disentangled structure without supervision is valuable. We will revise the introduction to make this point clearer and avoid any potential misunderstanding. The revised version will more accurately emphasize that unsupervised representation learning and disentanglement remain important and actively studied problems, independent of whether generative models are conditioned on labels or not.
>
>
> ***W2: I am a bit worried about the claim of "simplicity" of the proposed approach. I view the proposed method as a complex system, involving several models (some of which pretrained, others that require separate training) combined in a coherent design, which is far from being simple. Also, the lack of apparent hyper-parameters (a critic the authors advance for other methods on disentanglement from the state of the art), is in my humble opinion, another illustration of downplaying the complications of the proposed system. In practice, diffusion models are not so easy to train, as demonstrated by the choice of the authors to rely on an enormous amount of work that has been done to come up with "recipies" to elucidate the design space of diffusion models.***
>
>
> We appreciate the reviewer’s concern and understand how our framework may appear complex, given that it includes several components, some of which are pretrained and others trained separately. Our claim of “simplicity” is perhaps better understood in terms of training complexity for disentanglement, rather than the number of modules in the overall system.
>
> Our VQ-VAE encoder is pre-trained. While this adds an external component, it does not introduce hyperparameter choices or training demands within our framework. Using pre-trained tokenizers like VQ-VAE is standard practice, popularized by Latent Diffusion Models [1], as it shifts the training process from high-dimensional pixel space to a compressed latent space, thereby reducing computational demands.
>
> Importantly, existing sequential disentanglement methods (e.g., C-DSVAE, SPYL) also incorporate modules for modeling the prior. The key distinction in DiffSDA is that we decouple prior learning from disentanglement training: the latent DDIM prior is extremely lightweight (≈15 minutes on a single Nvidia 4090 GPU), trained independently, and only needed when sampling is required. Thus, it does not contribute to the difficulty of the disentanglement stage.
>
> Where DiffSDA aims to be simple and where it differs substantially from previous approaches is in the training objective for disentanglement itself. The entire disentanglement stage is optimized with a single loss function, avoiding the extensive hyperparameter tuning required to balance several competing terms, often four or more, in state-of-the-art VAE-based sequential disentanglement methods. This also eliminates gradient conflicts by construction (see our response to Reviewer JQNL). This is the sense in which our approach is “simple”: not in architectural minimalism, but in the absence of delicate multi-term loss engineering and tuning, which has previously been a difficulty in this domain.
>
> Finally, regarding the reviewer’s remark on the difficulty of training diffusion models, we would like to clarify our motivation. While the broader diffusion literature indeed explores a rich design space, in our setting, the training procedure is both stable and straightforward: we train a lightweight latent-space diffusion model using a single objective, and in our experience, this process does not pose practical difficulties. Our choice to adopt EDM is not due to training instability, but rather because it offers well-documented benefits in both training and inference efficiency, most notably, reducing the number of function evaluations required for sampling (from $\approx$1000 in standard DDPMs to $\approx$60). Thus, EDM allows us to retain simplicity while improving computational efficiency, without adding complexity to the disentanglement stage or the overall framework.
>
>
> [1] "High-Resolution Image Synthesis with Latent Diffusion Models", Rombach et al., 2021

---

> > ### Author Response · Authors · 2025-11-20
> > **Part 2**
> >
> > ***Q1: From the introduction, the requirement for "labeled" data stems from, e.g., text-to-image generation. Original work from Ho et al., and from Song et al., are unconditional models which do not require labeled data.***
> >
> > See our response to W1.
> >
> > ***Q2: Check your claims: when the authors say they propose a novel probabilistic approach, they are actually using standard practice in diffusion modeling. The main difference is that they combine two diffusion processes, one for learning (and generating) disentangled latents for static and dynamic components, and one for learning (and generating) the sequential data distribution, conditioned on the disentangled latents. Both models are learned concurrently using a single loss. While my remark does not want to diminish the technical contribution of the proposed method, I think that the claim for novelty is too strong.***
> >
> >
> > We respectfully disagree with the reviewer’s assessment. While it is true that our method builds on standard diffusion modeling principles, the probabilistic formulation we introduce for disentangling sequential data into static and dynamic components is, to the best of our knowledge, novel. Prior diffusion-based works do not provide a principled probabilistic mechanism for separating temporal information into complementary factors, nor do they define a joint model that enables both latent disentanglement and downstream conditional generation in a unified framework.
> >
> > In our approach, the disentanglement is not an ad-hoc architectural choice or heuristic, but arises directly from the probabilistic structure we define, specifically, the factorization of latents and the associated optimization given by Eq. (3). This probabilistic formulation is fundamentally different from existing diffusion architectures, and we are not aware of any prior diffusion-based methods that suggest such a decomposition of sequential information.
> >
> > ***Q3: As stated in lines 212-213, eq. (3) does not require training for the latents generative model, thus this optimization is separate. Then, the loss in eq. (3) is not the only loss you consider in your overall training of DiffSDA, right? Then, I would like to ask the authors to clarify why in lines 221-222 they claim that only a single loss term is required.***
> >
> >
> > We thank the reviewer for this thoughtful question and the opportunity to clarify this point. To clarify:
> >
> > a) Disentanglement itself is achieved solely by optimizing Eq. (3). The latent DDIM generative model is trained only if sampling (i.e., generation) is desired. Importantly, DiffSDA does not require a trained latent diffusion model in order to perform disentanglement: given an input sequence, we can recover its static and dynamic factors directly from Eq. (3) alone. Thus, lines 221–222 are accurate in stating that only a single loss term is required for disentanglement.
> >
> > b) Training the latent DDIM model is a completely separate and optional process, used only for generation. Its objective is independent from Eq. (3), and therefore does not introduce the coupled multi-term losses or hyperparameter tuning that characterize prior sequential disentanglement methods. This decoupling is a key advantage of DiffSDA: disentanglement relies on a single, self-contained optimization, while generative modeling, if needed, can be trained simply, independently, without affecting the disentanglement procedure and with no conflicting gradients.
> >
> >
> > ***Q4: In line 227, the last hidden representation is fed to a linear layer to produce , but this is not reflected in Figure 1. This is just nitpicking, but for clarity, the figure could be easily amended to be coherent with the textual description of the architecture.***
> >
> > We thank the reviewer for the helpful suggestion. We have updated Fig. 1 in the revised version to more accurately reflect our architecture, following the reviewer’s comment.
> >
> >
> > ***Q5: In lines 265-269, the authors mention that for high-dimensional data such as video, they abuse the notation (for ease of reading), and use symbols that denote ambient space, to indicate vector-quantized latent variables. This is clear. The question is: when considering other modalities, such as audio or general timeseries, there is no need for the VQ-VAE embeddings right?***
> >
> > Yes, that is correct. Our use of VQ-VAE notation is specific to the high-dimensional video setting, where vector-quantized latents are the practical choice. For lower-dimensional modalities such as audio or general time-series, VQ-VAE embeddings are not required. In those cases, the method operates directly on the native representation.

---

> > > ### Comment · Reviewer_nupF · 2025-11-24
> > > **Thank you for the rebuttal, no further questions on my side**
> > >
> > > Dear authors, this is a message to acknowledge and thank you for the rebuttal. I have read it and it positively addresses my main concerns.
> > >
> > > I have read the other reviews and your rebuttal for them. I now will wait for the discussion phase to begin.
> > >
> > > Thank you for your work!

---

### Official Review · Reviewer_jQnL · 2025-10-29

**Soundness:** 3
**Presentation:** 3
**Contribution:** 3
**Rating:** 6
**Confidence:** 4

**Summary:**

This paper proposes DiffSDA, a novel framework for unsupervised sequential disentanglement that leverages diffusion models to separate static and dynamic factors of variation across diverse modalities such as video, audio, and time-series data.
The key claimed contributions are:A novel, modal-agnostic probabilistic framework for sequential disentanglement based on diffusion processes, which models static and dynamic factors as interdependent.An efficient design that achieves disentanglement using a single, unified score estimation loss, simplifying optimization compared to methods requiring multiple loss terms.Demonstrated effectiveness on high-dimensional, real-world data, with capabilities for zero-shot disentanglement and further factorization into multiple interpretable factors via post-hoc PCA.The introduction of a new evaluation protocol for video-based disentanglement, including high-resolution datasets and unsupervised metrics (AED, AKD).

**Strengths:**

1. The work successfully bridges powerful diffusion models with the challenging problem of sequential disentanglement, a domain previously dominated by VAE/GAN-based approaches (e.g., SPYL, DBSE). The modal-agnostic claim is well-supported by experiments.
2. The use of a single diffusion loss term is a significant advantage, effectively circumventing the complex hyperparameter tuning and multi-term loss balancing required in prior works (e.g., C-DSVAE uses five loss weights). This makes the method more accessible and robust.
3. The paper provides extensive experiments. Qualitatively, the conditional/zero-shot swap and multifactor traversal results (e.g., Figs. 2, 3, 4, 23-47) are visually compelling and demonstrate clear disentanglement.
4. Quantitatively, the model consistently outperforms strong baselines (SPYL, DBSE) across multiple metrics (AED, AKD, etc.) and datasets.

**Weaknesses:**

1. Why is the dependent modeling of static and dynamic factors theoretically justified or preferable for disentanglement? The empirical result (13% FVD improvement) is convincing, but an intuitive or formal explanation is lacking.
2. A more detailed analysis of why the single diffusion loss naturally leads to disentanglement, beyond the empirical observations that the static factor is shared and the dynamic factors are low-dimensional, would be highly valuable.
3. The paper highlights the efficiency of the EDM sampler (63 NFEs) but does not provide a comparative analysis of training or inference costs against the baselines. Given that diffusion models are generally more computationally intensive, a discussion of this trade-off would be helpful.

**Questions:**

1.What key property of the diffusion model or its learned representations enables such strong cross-dataset, zero-shot disentanglement?
2.Are the semantic factors discovered via PCA (e.g., gender, age) consistent and stable across different random seeds and model initializations?
3.From an optimization standpoint, please justify the efficiency of the single loss formulation over a multi-term objective, for instance, by analyzing its mitigation of gradient conflict.
4.Please provide a quantitative comparison of training/inference time and GPU memory footprint against key baselines like SPYL and DBSE.

---

> ### Author Response · Authors · 2025-11-20
> **Part 1**
>
> ***W1: Why is the dependent modeling of static and dynamic factors theoretically justified or preferable for disentanglement? The empirical result (13% FVD improvement) is convincing, but an intuitive or formal explanation is lacking.***
>
> We thank the reviewer for this valuable question. In our description of the probabilistic model, we briefly mentioned that this modeling enables causality. That is, our model has the ability to learn intricate relationships between the static and dynamic factors, if needed.
>
> The theoretical justification lies in a crucial distinction between our prior (for generation) and our posterior (for inference/disentanglement). A dependent prior is theoretically preferable because it allows the model to learn the true generative distribution of the world, where static ($s$) and dynamic ($d$) factors are often highly correlated. An independent prior would be an incorrect model of reality, as it assumes any dynamic action is equally plausible for any static entity. An independent prior would treat "a baby driving a car" as just as likely as "a baby crawling." Our dependent prior can learn that the static factor "baby" strongly influences the likely dynamic factors ("crawling," "crying") and makes "driving" highly unlikely. This ability to capture intricate relationships leads to far more coherent and realistic generations, preventing unrealistic combinations. The 13% FVD improvement is the direct empirical validation of this superior modeling.
>
> Given an input sequence, and assuming its static component remains constant, there is no longer any need to model dependencies between the static and dynamic factors. Thus, disentanglement is achieved by enforcing independence in the posterior (as discussed in Eq. 2). This provides the necessary inductive bias to force the encoder to separate the static and dynamic factors during inference.
>
>
> ***W2: A more detailed analysis of why the single diffusion loss naturally leads to disentanglement, beyond the empirical observations that the static factor is shared and the dynamic factors are low-dimensional, would be highly valuable.***
>
> We appreciate the opportunity to clarify how unsupervised disentanglement is achieved in our method by learning with a single, simple loss objective.
>
> As outlined in the main paper, disentanglement emerges from two key inductive biases built into our model:
>
> Shared static vector across time: Using the same static vector for all time steps inherently prevents it from modeling temporal variation. As a result, temporal changes must be captured by the dynamic vector. If the static vector encoded time-varying features, it would produce identical outputs across frames, collapsing the temporal dimension and impairing both reconstruction and disentanglement.
>
> Low-dimensional dynamic vector: By constraining the dynamic vector’s dimensionality, we limit its capacity to store detailed or identity-specific information. This bottleneck encourages separation between dynamic content (e.g., motion) and static identity features.
>
> We provide empirical validation of these assumptions in App. G.2. Ablation studies show that removing the shared static vector significantly degrades disentanglement. Similarly, increasing the dimensionality of the dynamic vector leads it to capture static features, weakening the separation between the two factors.

---

> > ### Author Response · Authors · 2025-11-20
> > **Part 2**
> >
> > ***W3: The paper highlights the efficiency of the EDM sampler (63 NFEs) but does not provide a comparative analysis of training or inference costs against the baselines. Given that diffusion models are generally more computationally intensive, a discussion of this trade-off would be helpful.***
> >
> > We thank the reviewer for raising the question regarding the computational efficiency of the EDM sampler. Our focus on the EDM sampler is primarily in comparison to other diffusion-based autoencoders such as InfoDiff and DiffAE, which rely on significantly less efficient sampling strategies. EDM provides a substantially improved trade-off in sampling quality versus NFE budget, which is why it is central to our design.
> >
> > To further address concerns about computational cost, we added two tables comparing memory usage and runtime during both training and inference against state-of-the-art disentanglement models SPYL and DBSE. These comparisons allow us to verify that our method is not disproportionately resource-intensive relative to current approaches.
> >
> > As expected, VAE-based methods remain faster and lighter overall. However, we observe that our training costs are on par with methods such as SPYL and DBSE, and our inference costs, while higher, remain within a reasonable computational envelope for diffusion-based frameworks. Please refer to Tables 1 and 2 below for full results.
> >
> > ### **Table 1 - Training Time and Memory Usage**
> >
> > | **Method** | **Dataset** | **VRAM** | **Epochs** | **Time per Epoch (min)** | **Total Time (h)** |
> > |-----------|-------------|----------|------------|---------------------------|---------------------|
> > | **Ours**  | Vox1   | 17GB | 100 | 105 | 175 |
> > |           | CelebV | 17GB | 450 | 20  | 150 |
> > |           | Taichi | 20GB | 40  | 45  | 30  |
> > | **SPYL**  | Vox1   | 13.5GB | 150 | 55 | 137.5 |
> > |           | CelebV | 9GB    | 150 | 25 | 62.5  |
> > |           | Taichi | 12GB   | 600 | 2  | 20    |
> > | **DBSE**  | Vox1   | 10GB | 150 | 25 | 62.5 |
> > |           | CelebV | 10GB | 150 | 15 | 37.5 |
> > |           | Taichi | 3.5GB | 600 | 2  | 20   |
> >
> > **Table 1 caption:**
> > *Training times across methods and datasets. All models were trained on a single RTX 4090 for fair comparison. For experiments reported elsewhere in the paper, DiffSDA is typically trained on 3×RTX 4090 GPUs to reduce wall-clock time.*
> >
> >
> >
> > ### **Table 2 - Inference Time and Memory Usage**
> >
> > | **Method** | **Dataset** | **Total Time (s)** | **Peak VRAM** |
> > |------------|-------------|--------------------|----------------|
> > | **Ours**   | Vox1   | 3.746 ± 0.357 | 7GB |
> > |            | CelebV | 3.633 ± 0.011 | 7GB |
> > |            | Taichi | 3.097 ± 0.133 | 1.5GB |
> > | **SPYL**   | Vox1   | 0.021 ± 0.042 | 500MB |
> > |            | CelebV | 0.021 ± 0.041 | 500MB |
> > |            | Taichi | 0.016 ± 0.037 | 120MB |
> > | **DBSE**   | Vox1   | 0.021 ± 0.044 | 450MB |
> > |            | CelebV | 0.021 ± 0.044 | 450MB |
> > |            | Taichi | 0.017 ± 0.038 | 175MB |
> >
> > **Table 2 caption:**
> > *Inference performance (sampling + decoding) with VQ-VAE decoding included. Results averaged across 10 runs with batch size 8.*

---

> ### Author Response · Authors · 2025-11-20
> **Part 3**
>
> ***Q1: What key property of the diffusion model or its learned representations enables such strong cross-dataset, zero-shot disentanglement?***
>
>
> We appreciate the reviewer’s thoughtful question. One property that we suspect allows these strong results is the pre-trained VQ-VAE, which was trained on a diverse corpus of images and provides a unified latent representation space. This shared space facilitates the model’s ability to generalize across datasets, even when they differ in visual characteristics.
>
> To further investigate this hypothesis, we conducted an additional zero-shot experiment. Specifically, we trained DiffSDA on a downsampled (64×64) VoxCeleb dataset and tested its transferability to the MUG dataset. Importantly, in this setting, we did not use the VQ-VAE. Preliminary results from this setup indicate that zero-shot generalization degrades compared to the original pipeline that includes VQ-VAE. These findings support our belief that the VQ-VAE contributes to the model’s ability to generalize by encoding diverse images into a coherent and transferable latent space. We added the resulting images to the revised version of the paper to appendix G.5.
>
>
>
> ***Q2:  Are the semantic factors discovered via PCA (e.g., gender, age) consistent and stable across different random seeds and model initializations?***
>
> Thank you for the insightful question. To assess stability across seeds, we ran the following experiment. Using a VoxCeleb trained model, we sampled 8 batches with different random seeds, and computed PCA separately for the dynamic and static components as described in Section 4.3. We then calculated the cosine similarity between every pair of seeds and examined the average similarity for the first 20 static components and for all 12 dynamic components. The dynamic components showed consistently strong alignment, with cosine similarity above **0.99**. Likewise, the static components also exhibited high stability, with average cosine similarity above **0.92**.
>
>
> ***Q3: From an optimization standpoint, please justify the efficiency of the single loss formulation over a multi-term objective, for instance, by analyzing its mitigation of gradient conflict.***
>
>
> Thank you for highlighting the gradient-conflict issue. To quantify it, we measured the average number of conflicts per batch on the MUG dataset, defining a conflict when the angle between two task-gradients exceeds $\frac{\pi}{2}$ [1]. DBSE shows **2.8 conflicts** per batch (maximum 3), while SPYL shows **5.4 conflicts** (maximum 15), confirming that multi-term objectives
> in sequential disentanglement suffer from substantial gradient interference. In contrast, our single-loss formulation inherently avoids these conflicts (i.e., 0 conflicts), leading to more efficient and stable optimization.
>
> [1] "Gradient Surgery for Multi-Task Learning", Yu et al., 2020
>
> ***Q4: Please provide a quantitative comparison of training/inference time and GPU memory footprint against key baselines like SPYL and DBSE.***
>
> Please see our response to W3.

---

> ### Comment · Reviewer_jQnL · 2025-11-27
>
> Thank you for the detailed rebuttal. I have no further comments.

---

### Official Review · Reviewer_d96r · 2025-11-01

**Soundness:** 3
**Presentation:** 3
**Contribution:** 2
**Rating:** 6
**Confidence:** 2

**Summary:**

DiffSDA is a modal-agnostic diffusion sequential disentanglement autoencoder that factorizes static and dynamic factors using a single, unified score-estimation loss.

**Strengths:**

1. It allows static and dynamic factors to be interdependent, which makes the model more general.

2. It can be applied to video, audio and time-series data, and achieves sota performance.

3. It present a new evaluation protocol for high-quality visual sequential disentanglement.

**Weaknesses:**

1. I am curious to what extent this model can reduce hyperparameter tuning. The paper claims that other methods always rely on multiple loss terms, making the optimization process more complex, whereas this model only requires a single standard loss term. However, in Appendix B.2, it seems that the VQ-VAE is pre-trained with a perceptual loss and a patch-based adversarial objective, and in Appendix B.3, you train a latent DDIM prior. Could you elaborate on how this method is actually simpler than GAN- or VAE-based approaches that can also disentangle dynamic and static components?

2. In Figure 3, it seems that SPYL failed to reconstruct the image well. Since reconstruction is usually simpler than disentanglement, this might be due to bad hyperparameters. If such trivial failures like SPYL’s could be avoided, the comparison would be more meaningful.

**Questions:**

See Weaknesses.

---

> ### Author Response · Authors · 2025-11-20
>
> ***W1: I am curious to what extent this model can reduce hyperparameter tuning. The paper claims that other methods always rely on multiple loss terms, making the optimization process more complex, whereas this model only requires a single standard loss term. However, in Appendix B.2, it seems that the VQ-VAE is pre-trained with a perceptual loss and a patch-based adversarial objective, and in Appendix B.3, you train a latent DDIM prior. Could you elaborate on how this method is actually simpler than GAN- or VAE-based approaches that can also disentangle dynamic and static components?***
>
> We appreciate the reviewer’s thoughtful question. We understand that our framework may initially appear more complex because it includes several components, but the training required for disentanglement is substantially simpler than in prior work.
>
> As the reviewer noted, the VQ-VAE is pre-trained and therefore does not introduce additional hyperparameters or training demands. Using pre-trained tokenizers like VQ-VAE is standard practice, popularized by Latent Diffusion Models [1], as it shifts the training process from high-dimensional pixel space to a compressed latent space, thereby reducing computational demands. Regarding the prior, existing sequential disentanglement methods (e.g., C-DSVAE, SPYL) also relied on separate components for modeling the prior; the key distinction in DiffSDA is that we decouple prior learning from disentanglement itself. Our latent DDIM prior is extremely lightweight (≈15 minutes of training on a single Nvidia 4090 GPU), trained independently, and only required when generation is needed, so it does not add to the complexity of the disentanglement stage.
>
> Crucially, the disentanglement stage in DiffSDA optimizes a single loss function, avoiding the extensive hyperparameter tuning required to balance multiple competing terms, often four or more, in state-of-the-art VAE-based sequential disentanglement methods. It also inherently avoids gradient conflicts (as discussed in our response to Reviewer JQNL, where we show that prior methods exhibit significant optimization interference). This results in a substantially reduced tuning burden and more stable training. Finally, GAN-based disentanglement approaches are no longer competitive with current methods; for this reason, we do not include them in our comparisons.
>
> [1] "High-Resolution Image Synthesis with Latent Diffusion Models", Rombach et al., 2021
>
>
> ***W2: In Figure 3, it seems that SPYL failed to reconstruct the image well. Since reconstruction is usually simpler than disentanglement, this might be due to bad hyperparameters. If such trivial failures like SPYL’s could be avoided, the comparison would be more meaningful.***
>
> We appreciate the reviewer’s concern regarding SPYL’s reconstruction quality. To ensure a fair comparison, we conducted an extensive hyperparameter search on Taichi, spanning 144 configurations, but none produced satisfactory reconstructions or disentanglement performance. This aligns with observations reported in the original SPYL paper, where reconstruction quality can be unstable and highly sensitive to hyperparameters. To further validate this, we additionally performed experiments on Vox and CelebV, running 400 configurations for each dataset, chosen based on the best-performing settings we identified on MUG. This strategy allowed us to balance fairness with computational feasibility while still giving SPYL multiple opportunities to succeed. Despite this comprehensive search, SPYL consistently struggled to produce acceptable reconstructions, suggesting that the observed failures in Figure 3 are not due to trivial hyperparameter issues but rather reflect the method’s disadvantage in processing real-world data.

---

### Official Review · Reviewer_DzJi · 2025-11-01

**Soundness:** 2
**Presentation:** 3
**Contribution:** 2
**Rating:** 4
**Confidence:** 4

**Summary:**

This paper propose a unsupervised diffusion model that can disentangle factors in a time series data. The model is insprired on a principled probabilistic model, with the ability of both dynamic-static and conponent level disentanglement. The extensive experiments results verify the conclusion.

**Strengths:**

1. A probabilitic model is provided, which provide a theoritical guarantee for disentanglement in diffusion models.
2. Extensivce experiments are provided, which makes the paper more convincing.
3. The paper is clear and easy to follow.

**Weaknesses:**

### Major Concerns
1. This paper assumes $d$ and $s$ are independent. However, in many real world senarios, they cannot be independent, as sometimes the content of a video can decide the type of actions to be taken in a video. This limits the generalization of thie model.
2. The dataset is limited to simple scenarios like human face, where this assumption is reasonable. However, the performance in open-domain dataset is not verified.
3.  This paper claims the ability of disentanglement, but some traditional disentanglement metrics are not compared, i.e., SAP, modularity, e.t.c..
4. The baseline models in the paper are out-dated, especially for the video generation part.
### Minor Concerns
1. Typo: line 203 pr-evious

**Questions:**

1. The video is generated frame by frame, which may reduce the spatio-temporal consistency in the video generation. Is it possible to adapt this model to LVDM or similar models to strength its video generation ability?
2. Can the unsupervisely learned concepts be combined with other supervised condition signals? For example, how this method can contribute to the current T2V generation framework?

---

> ### Author Response · Authors · 2025-11-20
>
> ***W1: This paper assumes $s$ and $d$ are independent. However, in many real world senarios, they cannot be independent, as sometimes the content of a video can decide the type of actions to be taken in a video. This limits the generalization of thie model.***
>
> We will clarify this point as it highlights a crucial design choice in our model: the distinct treatment of the posterior and the prior. In Eq. (2), we assume independence between static and dynamic components in the posterior. In unsupervised sequential disentanglement, the fundamental assumption is that static features (whether in video, audio, or time-series) remain unchanged throughout the input sequence $\mathbf{x}$. Therefore, during inference, given $\mathbf{x}$, we can extract these features independently to achieve separation, even if they are effectively dependent a priori. To address the natural dependencies mentioned by the reviewer, we are the first to propose a dependence assumption in the prior distribution. When training the latent DDIM for generation, we do not enforce independence between factors. This allows the model to capture the intricate relationships between static and dynamic features (e.g., specific content dictating specific actions), enabling the generation of coherent and realistic samples. This is briefly mentioned in lines 190-191 of the main paper: "iii) causality-our model has the ability to learn intricate relationships between the static and dynamic factors, if needed.
>
>
> ***W2: The dataset is limited to simple scenarios like human face, where this assumption is reasonable. However, the performance in open-domain dataset is not verified.***
>
>
> We thank the reviewer for this point, as it allows us to clarify the breadth of our evaluation. We respectfully note that the reviewer's comment is imprecise. Our evaluation is not limited to human faces. A key contribution of our work is demonstrating our method's modality-agnostic capabilities on diverse data types, including challenging audio and time-series datasets, which are not "human face scenarios."
>
> Furthermore, regarding the benchmarks that are human-centric (like VoxCeleb and CelebV-HQ), we must respectfully disagree with the characterization that they are "simple scenarios." All the real-world video benchmarks we use, represent a significant step in complexity over prior work on unsupervised sequential disentanglement, which focused almost exclusively on "toy" datasets (e.g., MUG, Sprites). In this context, VoxCeleb and CelebV-HQ can be viewed as open-domain datasets with significant variability. State-of-the-art performance on them is far from saturated, confirming they remain challenging and unsolved testbeds.
>
>
> ***W3: This paper claims the ability of disentanglement, but some traditional disentanglement metrics are not compared, i.e., SAP, modularity, e.t.c..***
>
> We thank the reviewer for this important point, which highlights a key distinction. We agree that the metrics referenced are standard for static disentanglement (e.g., in image data). Our work, however, addresses sequential disentanglement, where the goal is to separate static factors ($\mathbf{s}$) from dynamic factors ($\mathbf{d}_{1:\tau}$). This problem structure requires an adaptation of the standard metric definitions:
>
> 1) Modularity: We adapt the principle from [1] to our static/dynamic split. In our context, modularity means static data factors should only influence $\mathbf{s}$, and dynamic factors should only influence $\mathbf{d}_{1:\tau}$. Our Judge metric, AKD, and AED are all designed to measure this specific property in the sequential setting.
>
> 2) Compactness (SAP): The standard SAP metric measures a very strict compactness concept, where one factor ideally maps to a single dimension [1]. To the best of our knowledge, this strict assumption is not standard in the sequential disentanglement community, which often models complex, high-dimensional dynamics.
>
>
> [1] "Measuring disentanglement: a review of metrics", Carbonneau et al., 2020

---

> > ### Author Response · Authors · 2025-11-20
> >
> > ***W4: The baseline models in the paper are out-dated, especially for the video generation part.***
> >
> > We would like to clarify that our framework is fully unsupervised and modality-agnostic. It is designed to disentangle static and dynamic factors in any sequential modality (video, audio, time series, etc.). Within this setting, the appropriate baselines are methods that likewise perform unsupervised disentanglement of sequential data, rather than supervised or domain-specialized video generators.
> >
> > Among this family of methods, we compare against the current state-of-the-art approaches, DBSE and SPYL, which are the most relevant and directly comparable baselines. We also included Table 1 in the submission to provide a high-level comparison to a broader spectrum of related approaches, including video-generation-oriented methods, to contextualize the relation between our method and prior work. We are happy to further extend and refine this comparison table in the revised version to more explicitly clarify the distinctions between unsupervised disentanglement baselines and video generators.
> >
> >
> > ***MC: Typo: line 203 pr-evious***
> >
> > We fixed it in the revised version.
> >
> >
> > ***Q1: The video is generated frame by frame, which may reduce the spatio-temporal consistency in the video generation. Is it possible to adapt this model to LVDM or similar models to strength its video generation ability?***
> >
> > Thank you for the insightful suggestion. Adapting our approach to LVDM or related video-generation frameworks is indeed an interesting direction for future work. Our current method is limited to per-frame generation because our disentanglement relies on a separate latent vector $d_i$ that varies across frames, as we discuss in Appendix G.2. This mechanism would not directly transfer to models where temporal dynamics are jointly modeled, and $s$ and $d_{0:V}$ cannot be fully separated during generation. Nonetheless, we agree that exploring such extensions is valuable, and we will outline this direction as a potential avenue for future research.
> >
> > ***Q2: Can the unsupervisedly learned concepts be combined with other supervised condition signals? For example, how this method can contribute to the current T2V generation framework?***
> >
> > We did not explore the integration of our learned concepts with supervised conditioning signals, such as those used in text-to-video (T2V) frameworks. We agree that combining disentangled latent representations with supervised modalities is a promising direction that could further enhance controllability in T2V generation, and we consider this an exciting avenue for future work.

---

> ### Comment · Reviewer_DzJi · 2025-11-20
> **Further questions**
>
> Thank you to the authors for their responses. I now better understand the distinction between the posterior and prior independence assumptions. However, I still have some concerns regarding the practical implications of this design choice. While the prior allows dependencies between static and dynamic components, it would be helpful to demonstrate whether the model can indeed capture such dependencies in practice. For instance, when the appearing objects differ, such as a guitar versus a pian, the distribution of actions is also likely to change. Evaluating the model on datasets like UCF-101, which naturally contain such interdependencies, would better illustrate the model’s ability to generalize beyond idealized independence assumptions.

---

> > ### Author Response · Authors · 2025-11-24
> >
> > We thank the reviewer for the thoughtful follow-up and for engaging with the distinction between prior and posterior independence in our model.
> >
> > As the reviewer notes, modeling static and dynamic features independently at the generative level can lead to unrealistic combinations such as “a person playing the piano while holding a guitar.” This is precisely the type of undesirable behavior our design aims to mitigate: although we assume a factorized posterior for tractable inference, the prior over (s,d) is allowed to capture dependencies between static and dynamic components. In other words, the generative model can assign low probability to incompatible pairs (s,d), such as guitar–piano hybrids, even if the inference network factorizes.
> >
> > To clarify our evaluation: our main experiments are based on posterior swaps, where we explicitly measure modularity using Judge, AKD, and AED metrics. In these setups, we follow the established evaluation protocol from prior work, in which swapping is performed between sequences where the counterfactual is semantically meaningful (e.g., changing the background or appearance while keeping the type of motion comparable). For example, in CelebA-HQ, the expression and face direction of a celebrity can be taken from another video, and in Taichi, the movement of one person can be transferred to another person performing Taichi. In contrast, on datasets like UCF-101, two videos such as “playing guitar” and “playing piano” typically differ both in the object (guitar vs. piano) and in the fine-grained motion patterns. In an unsupervised sequential disentanglement setting, and as we noted in our answer to W1, the standard assumption is that static features remain unchanged over the sequence, while dynamics encode how the static content evolves. Under this assumption, a swap between “guitar” and “piano” clips does not provide a meaningful test of modularity: the instrument identity would be encoded in the static factor $s$, while the action style (how the instrument is played) is captured in $d$, so swapping across entirely different action categories becomes ill-posed rather than informative. This breaks the standard assumption underlying unsupervised sequential disentanglement, and thus the request to evaluate via such swaps on UCF-101 deviates from the commonly used definition and benchmarking protocol for static–dynamic modularity.

---

### Meta-Review · Area_Chair_FPQF · 2026-01-06

**Summary:**

This paper develops a framework for unsupervised sequential disentanglement that leverages diffusion models to separate static and dynamic factors of variation across modalities such as video, audio, and time-series data. The reviewers generally appreciated the methodological contributions and extensive experiments. While there are initial concerns e.g. about empirical studies and certain design choices, most of them were addressed during the rebuttals, and I thereby recommend acceptance.

**Reviewer Concerns:**

Concerns addressed by rebuttal:
- Reviewers nupF questioned the claim about "simplicity" of the proposed approach, as it involves several modules combined. The authors explained that simplicity is understood as training complexity for disentanglement rather than number of modules.
- Reviewer DzJi raised concern about dataset being limited to "human face" scenarios, which the authors addressed by highlighting experiments on audio and time-series data.
- Reviewer jQnL requested a more detailed analysis of why the single diffusion loss naturally leads to disentanglement, which were further provided by the authors.
- Reviewer DzJi raised concern about the lack of comparison with traditional disentanglement metrics. The authors explained that their use of AKD and AED are more well-suited for their task of sequential disentanglement.
- Reviewer jQnL pointed out the issue of why the dependent modeling of static and dynamic factors is theoretically justified or preferable for disentanglement. The authors provided an explanation to address it.

Partially outstanding concerns:
- Reviewer Dzji was concerned about the model's performance on datasets like UCF-101, where static and dynamic components  are heavily interdependent. The authors argued that this violates the standard assumption of unsupervised sequential disentanglement.

**Reviewer Scores:**

Reviewer DzJi may have increased their score to 6 if they had been able to participate fully in the discussion.

---

### Decision · Program_Chairs · 2026-01-26

Accept (Poster)